# Midbrain extracellular matrix and microglia are associated with cognition in aging mice

**Daniel T. Gray** [1] ✉, **Abigail Gutierrez**[1], **Yasaman Jami-Alahmadi**[2], **Vijaya Pandey**[2], **Lin Pan**[3], **Ye Zhang**[3], **James A. Wohlschlegel** [2], **Ross A. McDevitt** [4] & **Lindsay M. De Biase** [1] ✉

Synapse dysfunction is tightly linked to cognitive changes during aging. Emerging evidence suggests that microglia and the extracellular matrix (ECM) can potently regulate synapse integrity and plasticity. Yet the brain ECM, and its relationship with microglia, synapses, and cognition during aging remains virtually unexplored. In this study we combine ECM-optimized proteomic workflows with histological analyses in aging mice and discover regional differences in ECM composition and aging-induced ECM remodeling across basal ganglia nuclei. Moreover, we combine two distinct behavioral classification strategies with fixed-tissue confocal imaging and proteomic analysis and identify relationships between the hyaluronan- and proteoglycan-rich ECM and cognitive aging phenotypes. Finally, we provide evidence that aging midbrain microglia lose capacity to interact with and regulate the ECM, and that these aging-associated microglial changes are accompanied by local ECM accumulation and worse behavioral performance. Together, these observations indicate that changing microglia-ECM-synapse interactions contribute to cognitive functioning during healthy aging.

Synapse loss and changes in synaptic plasticity are hallmark features of both normative brain aging and preclinical phases of neurodegenerative disease[1–3]. These synaptic changes have critical consequences, as preserved synapse status is linked to better cognitive outcomes in healthy aged rodents and nonhuman primates[1,4]. Growing evidence suggests the extracellular matrix (ECM) is a robust regulator of synaptic physiology during development and in early adulthood[5,6]. The ECM is not a static structure, but rather a dynamic network of proteins and carbohydrates remodeled to support numerous brain functions, including synaptic plasticity and tissue repair[5–7]. Histochemical staining of ECM components suggests that ECM abundance[8] and composition[9,10] vary across brain regions and that ECM remodeling can occur during aging[7,11,12]. This raises the possibility that ECM status helps determine vulnerability of specific brain regions to aging-related synaptic decline.

Compared to other bodily tissues, the brain ECM is relatively depleted in fibrous ECM proteins like collagens and elastins and enriched in glycosylated proteins like proteoglycans[5,13,14]. Ongoing regulation of the brain ECM is a cooperative effort amongst different neuronal, vascular, and glial cell subtypes involved in ECM protein and carbohydrate synthesis, assembly, and degradation[5,6,15]. Microglia, the brain's innate immune cells, express numerous ECM-relevant degradative enzymes as well as protease inhibitors[16–18], positioning them as potent modifiers of ECM structure. Microglia also regulate synapses[19–24], including through a recently discovered mechanism involving targeted degradation of ECM proteoglycans[7,25,26]. Moreover, microglial properties are substantially altered during aging, and changes in microglial density in the cerebral cortex are associated with ECM alterations in mouse models of Alzheimer's disease and aging

[1]Department of Physiology, David Geffen School of Medicine, University of California, Los Angeles, Los Angeles, CA, USA. [2]Department of Biological Chemistry, David Geffen School of Medicine, University of California, Los Angeles, Los Angeles, CA, USA. [3]Department of Psychiatry and Biobehavioral Sciences, Semel Institute for Neuroscience and Human Behavior, David Geffen School of Medicine, University of California, Los Angeles, Los Angeles, CA, USA. [4]Comparative Medicine Section, National Institute on Aging - Intramural Research Program, Baltimore, MD, USA. ✉e-mail: dtgray@mednet.ucla.edu; ldebiase@mednet.ucla.edu

macaques[27,28]. Together, these observations suggest that changes in microglia-ECM interactions may play central roles in shaping synapse dynamics during aging, and consequently patterns of age-associated cognitive decline. Yet, this microglia-ECM-synapse triad remains virtually unexamined in most brain regions in both young adulthood and aging.

Numerous cognitive deficits associated with normative aging have been linked with dysregulation in midbrain dopaminergic circuits[29,30]. Microglia near midbrain dopamine neurons exhibit robust aging-related phenotypes characterized by increases in proliferation and inflammatory factor production that are evident by middle age in mice[31]. Whether these premature microglia aging phenotypes coincide with changes in ECM composition, and whether microglial-ECM responses to aging impact cognitive phenotypes, remains unexamined.

In this study, we combine high-resolution imaging, quantitative tissue proteomics, and multiple behavioral characterization strategies to comprehensively map how microglial-ECM responses to aging in the basal ganglia align with individual differences in synapse status and cognition. Our results indicate that aging results in greater midbrain ECM and microglial abundances that arise alongside relatively stable synapse protein abundances. While both ECM and microglial aging phenotypes are associated with poorer cognitive functioning, we provide evidence that they impact cognition through somewhat independent mechanisms in aging mice.

## Results

### ECM-optimized proteomics reveals associations between ECM and synapse abundance in the aging basal ganglia

ECM proteins are highly glycosylated and comparatively insoluble, making it difficult to leverage traditional proteomic approaches to comprehensively map the brain ECM (i.e., matrisome). We compared two tissue processing workflows that have been used for ECM enrichment and analysis in peripheral bodily tissues: solubility-based subcellular fractionation[32] and chaotropic extraction and digestion[33]. While the two approaches identified similar numbers of structural ECM proteins (core matrisome) in brain tissue, subcellular fractionation, on average, was more efficient at enriching ECM proteoglycans, while chaotropic digestion better enriched more fibrous ECM proteins like collagens (Fig. S1). Given the relative enrichment of proteoglycans in brain tissue[13,14], we used the solubility-based workflow to create a comprehensive proteomic mapping of the aging ECM proteome in the midbrain and striatum, two key basal ganglia nuclei, from young-adult (3 months) and aged (20+ months) wild-type (WT) mice (Fig. 1a).

Volcano plots of the abundance of all proteins in the tissue showed a relatively even split between proteins that were up- and down-regulated during aging (Fig. 1b). In contrast, most core matrisome proteins increased in abundance during aging. Beyond protein abundance, changes to the structure and assembly of the matrix can occur and will be reflected in alterations to the solubility of specific ECM components. As expected, in both brain regions, ECM proteins were primarily detected in more insoluble tissue fractions (Fig. 1c). In the midbrain, individual ECM proteins exhibiting significant aging-related changes in abundance were relatively equally distributed across solubility fractions (Fig. 1d). In the striatum, however, the majority of individual matrisome proteins that changed in abundance with age were found in the most insoluble fraction (Fig. 1d). These regional differences were also observed at the level of protein abundances such that midbrain ECM protein abundances showed similar aging-related increases across subcellular fractions (Fig. 1e) whereas striatum ECM protein abundances disproportionately increased within insoluble fractions during aging (Fig. 1e). To probe this data further, we determined the number of ECM proteins showing significant changes in solubility during aging. This analysis revealed that just 6 percent of midbrain ECM proteins became more soluble in older animals and

none more insoluble, whereas no striatal ECM protein became more soluble and 18 percent became more insoluble (Fig. S1). These results indicate there are regional differences in both the abundance and solubility of ECM proteins at different points of the lifespan.

Core matrisome proteins fall into three subclasses: glycoproteins, proteoglycans, and collagens. Comparable numbers of glycoproteins were detected in the midbrain and striatum, and several were significantly upregulated in both regions with aging (e.g., VWA 1, TARP). Laminin glycoproteins were specifically upregulated in the aging striatum (Fig. 1f). More proteoglycans were detected in the midbrain compared to striatum, and nearly half of midbrain proteoglycans showed significant increases in abundance with age compared to just 1 in the striatum (Fig. 1f). Collagen abundances also varied prominently between regions, with eight distinct collagens detected in the midbrain compared to just two in striatum (Fig. 1f). Finally, several ECM regulatory proteins, which include metalloproteinases and protease inhibitors, were significantly more abundant in the aged midbrain (HTRA1, ADAM10, Cystatin C), whereas none changed in abundance with aging in the striatum (Fig. 1f). Together, these observations demonstrate that regional heterogeneity in ECM composition and modulation during aging extends to all three subclasses of ECM proteins and their regulators. An important cautionary note is that the proteomic data search strategy used did not include ECM-relevant post-translational modifications, which very likely resulted in an underestimation of certain highly cross-linked and modified ECM proteins such as collagens. It will be critical for future studies to determine the extent to which this limitation affects the detection and quantification of all classes of brain ECM proteins.

To relate matrisome status with other features of the whole tissue proteome, Weighted Gene Coexpression Network Analysis[34] was leveraged for the unbiased identification of protein co-expression patterns (Fig. 2a). This analysis identified 12 modules of covarying proteins. Most core matrisome (~82%) and synapse proteins (~63%) were members of yellow, brown, or tan modules (Fig. 2a), highlighting these modules as warranting further analysis. Innate immune proteins, which can play key roles in synapse remodeling[23], were also found within yellow (15%) and brown modules (12%), in addition to prominent presence in the turquoise (33%) module. Examination of module eigengenes revealed stark regional differences for the brown, yellow, and tan modules, but not the turquoise module (Fig. 2b). Pathway analysis indicated that brown module proteins were associated with numerous metabolic processes, yellow with chemical synaptic transmission, turquoise with RNA metabolism and translation, and tan with a mix of biological processes including ECM organization (Fig. 2c).

To further evaluate relationships between these modules and ECM, synapse, and immune proteins, we treated individual protein abundances as "traits" and examined their correlation with module eigengenes across samples (Fig. 2d). Relatively prominent associations were observed between ECM proteoglycans and brown/yellow/tan module eigengenes, with the hyaluronan linker proteins HAPLN1-4 and aggrecan showing significant correlations. Although many immune signaling proteins also showed significant associations with brown/yellow/tan module eigengenes (Fig. S2), complement proteins C1qA and C1qB, which are known to tag synapses for microglial engulfment[23], were only significantly correlated with turquoise module eigengenes. Finally, numerous synaptic proteins showed relationships with brown/yellow/tan module eigengenes, whereas very few correlated significantly with the turquoise module. Collectively, these observations suggest that synaptic and ECM protein abundances are linked in the basal ganglia, and that complement signaling may not be involved in targeted ECM remodeling in these brain regions.

Overall abundance of most synaptic proteins did not significantly differ between young-adult and aged mice, indicating that substantial synapse loss is likely not occurring in the basal ganglia during healthy aging (Fig. 2e). This aligns with previous reports showing that, while

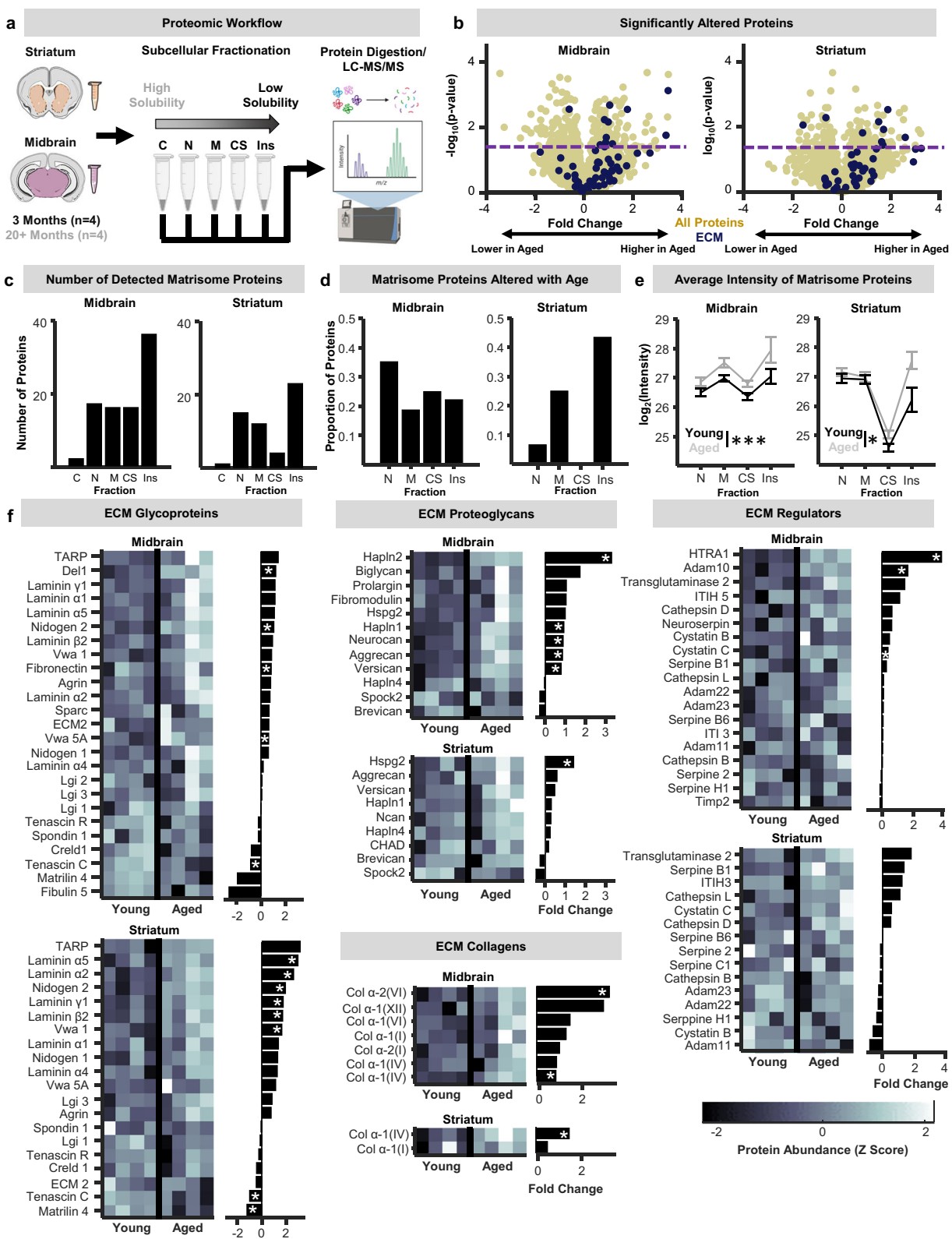

**a** Proteomic Workflow

**b** Significantly Altered Proteins

**c** Number of Detected Matrisome Proteins

**d** Matrisome Proteins Altered with Age

**e** Average Intensity of Matrisome Proteins

**f** ECM Glycoproteins | ECM Proteoglycans | ECM Regulators | ECM Collagens

synapse loss does occur in vulnerable brain regions during healthy aging, synapse numbers remain stable in many others[1,35]. To further explore relationships between age-associated changes in ECM composition (Fig. 1) and synapse status, we carried out additional pathway analysis focusing only on module proteins that were significantly altered during aging. Visualizing identified pathways as network plots revealed interconnected clusters of brown/yellow/tan module

proteins associated with ECM organization and ECM-receptor interactions in the midbrain. This cluster was directly connected to clusters associated with synapse organization (bolded lines; Fig. 2f), meaning that functional annotation predicts that proteins within these pathway "nodes" influence one another. While similar ECM clusters were observed in the striatum, they were not directly connected to pathway nodes associated with synapse organization (Fig. S2), suggesting that

**Fig. 1 | Quantitative proteomic analysis of the aging basal ganglia core matrisome. a** Schematic of the proteomic workflow. Schematic created in BioRender. Gray, D. (2025): https://BioRender.com/y6dza6g. The midbrain and striatum of young-adult (3 months) and aged (20–24 months) wild-type mice were extracted. Tissue underwent a solubility-based subcellular fractionation protocol that yielded multiple samples of proteins with decreasing solubility that correspond to cytoplasmic, nuclear, membrane, and cytoskeletal subcellular localizations, and a final insoluble pellet. All samples were then sent for liquid chromatography with tandem mass spectrometry (LC-MS/MS) analysis. **b** Volcano plots of all proteins pooled across solubility fractions for the midbrain and striatum. Dark blue dots denote individual structural extracellular matrix (ECM) proteins (e.g., core matrisome) detected in each region. Fold-changes were calculated with respect to aged mice (positive values indicate greater with age). Purple line represents $p < 0.05$ (unpaired ttest; two-sided) **c** Bar plots of the number of core matrisome proteins detected

across the different subcellular fractions in the midbrain and striatum (Fisher's exact test; Benjamini-Hochberg correction). **d** Bar plots depicting the proportion of structural ECM proteins that exhibited significant increases or decreases in abundance with age ($p < 0.05$; unpaired t-test; two-sided). **e** Average log2-transformed intensity values of core matrisome proteins across the nuclear, membrane, cytoskeletal, and insoluble fractions for young-adult (black) and aged (gray) mice in the midbrain and striatum (n-way ANOVA; two-sided; Midbrain(Age): p = 0.0012; Striatum(Age): $p = 0.67$; $n = 4$ mice per age per fraction). Data are presented as mean values ± SEM. **f** Heat maps of protein abundances (z scored) and bar plots of corresponding fold-changes with age for ECM glycoproteins, proteoglycans, collagens, and ECM regulators (* represents p < 0.05; unpaired t-test; two-sided). Source data are provided in the file Source Data - Fig. 1 and all statistics are provided in Supplementary Data 6. * represents p < 0.05, *** represents p < 0.001.

---

age-associated ECM remodeling has region-specific relationships with synapse status. Together, this proteomic mapping of matrisome, synapse, and immune proteins indicates that, in a normative aging context, regional variation in ECM status plays roles in establishing and/or maintaining regional basal ganglia synapse profiles.

## Mesolimbic ECM networks differ across region and age and are positioned for synapse interactions

Although proteomic approaches provide unbiased and comprehensive quantification of ECM protein abundance, they cannot reveal the morphology and spatial distribution of ECM components. For independent analysis of the brain ECM during aging, we histologically examined the ventral tegmental area (VTA, midbrain) and nucleus accumbens (NAc, ventral striatum), first using Wisteria floribunda agglutinin (WFA), a lectin that preferentially labels N-acetylgalactosamine residues found on glycosylated ECM proteins[36–39]. High-resolution (63x) confocal images were acquired from young-adult (4 months) and late-middle-aged (18 months), WT C57Bl6 mice, and ECM field-of-view coverage was calculated. In both the VTA and NAc, WFA label was distributed relatively evenly across fields of view and perineuronal net accumulations were only occasionally observed, indicating presence of a prominent interstitial matrix in both regions (Fig. 3b). WFA tissue coverage measures incorporating all WFA signal regardless of its association with the interstitial matrix or perineuronal nets were similar between the VTA and NAc of young-adult mice, and substantially greater in the VTA of middle-aged mice compared to young (Fig. 3c). To evaluate whether the aging-related increase in WFA within the VTA was driven primarily by increases in the abundance of the interstitial matrix or larger accumulations around neurons and/or vasculature, we implemented a size filter (150 pixels) to separate smaller WFA puncta from larger accumulations. This analysis indicated that increases in WFA within the VTA of older mice arise both from greater abundance of the interstitial ECM and the size or abundance of larger perineuronal/perivascular WFA accumulations (Fig. S3).

Chondroitin sulfate proteoglycans are sometimes referred to as 'hyalectans' due to their ability to interact with the ubiquitous ECM scaffold hyaluronan via several different linker proteins (hyaluronan and proteoglycan link proteins (HAPLNs))[40]. Because proteomic mapping suggested that numerous hyalectans and HAPLNs were significantly upregulated in the aged midbrain (Fig. 1), and because hyaluronan is positioned to play central roles in determining overall ECM tissue topology[26,41], we also histologically examined hyaluronan in young-adult and late-middle-aged mice. Hyaluronan tissue coverage was greater in the VTA compared to NAc, and on average higher in the VTA during aging (Fig. 3e). Together, these histological findings are consistent with proteomic detection of greater midbrain ECM protein levels during aging (Fig. 1f) and indicate that a feature of VTA aging is an accumulation of glycosylated ECM proteins and hyaluronan scaffolds.

Proteoglycans anchored to hyaluronan can impact synapses via multiple mechanisms, including limiting structural remodeling and regulating lateral diffusion of neurotransmitter receptors[7,42]. To relate ECM structure to local synapse status, densities of excitatory pre- and post-synaptic proteins (VGlut1 and Homer2, respectively) were quantified (Fig. 3f). Both VGlut1 and Homer2 densities were greater in the NAc compared to VTA but not altered by aging in either region (Fig. S4), consistent with proteomic data suggesting minimal synapse loss in the aging midbrain and striatum. The abundance of colocalized VGlut1-Homer2 puncta, which may better represent functional synapses, also did not differ with age in either region (Fig. 3g). To probe spatial relationships between ECM and synapses, hyaluronan fibrils were reconstructed and dilated by 0.5 μm, to estimate the density of Homer2 puncta within the potential territory of proteoglycans anchored to this scaffold (Fig. 3h). In both regions, hyaluronan-homer2 spatial associations were greater than would be expected by chance (associations detected when rotating one fluorescence channel by 90 degrees, Fig. 3i), supporting the idea of functional associations between local ECM and synapses. The density and proportion of Homer2 within 0.5 μm of hyaluronan were similar in young-adult and late-middle-aged mice and not different across regions, although greater variability was observed in the NAc. Together, these histological findings validate regional ECM heterogeneity revealed by proteomics, confirm regional differences in age-associated ECM remodeling, and suggest that ECM-synapse spatial associations occur throughout life.

## Microglia-ECM relationships during normative aging

VTA microglia exhibit robust aging-related phenotypes characterized by changes in proliferation, morphology, and inflammatory factor production[31], raising the possibility that microglia contribute to ECM accumulations in the aging VTA. Furthermore, aging-related increases in ECM protein abundance at the level of tissue proteomics (Fig. 1) arose alongside increases in the abundance of most detected microglia-enriched proteins with advanced age (Fig. 4a). To examine whether changes in microglial and ECM abundance align in the aging VTA, we histologically co-labeled the ECM (hyaluronan) and microglia (IBA1) in young-adult (4 months) and late-middle-aged (18 months) WT mice and examined relationships between the two. VTA microglia densities were greater in middle-aged mice compared to young-adults (Fig. 4b; see scatter plot). Microglia densities were significantly positively correlated with hyaluronan tissue coverage in young-adult mice, and this significant relationship was lost in the older mice (Fig. 4b). Next, microglial morphology was evaluated using a 2-dimensional Sholl analysis and, as shown previously[31], aged VTA microglia exhibited fewer Sholl intersections compared to young, indicative of a less complex morphology (Fig. S5). As with microglia densities, microglial morphological complexity was significantly positively correlated with hyaluronan deposition in young-adult mice and this relationship was lost in the older animals (Fig. 4c). To determine whether VTA microglia

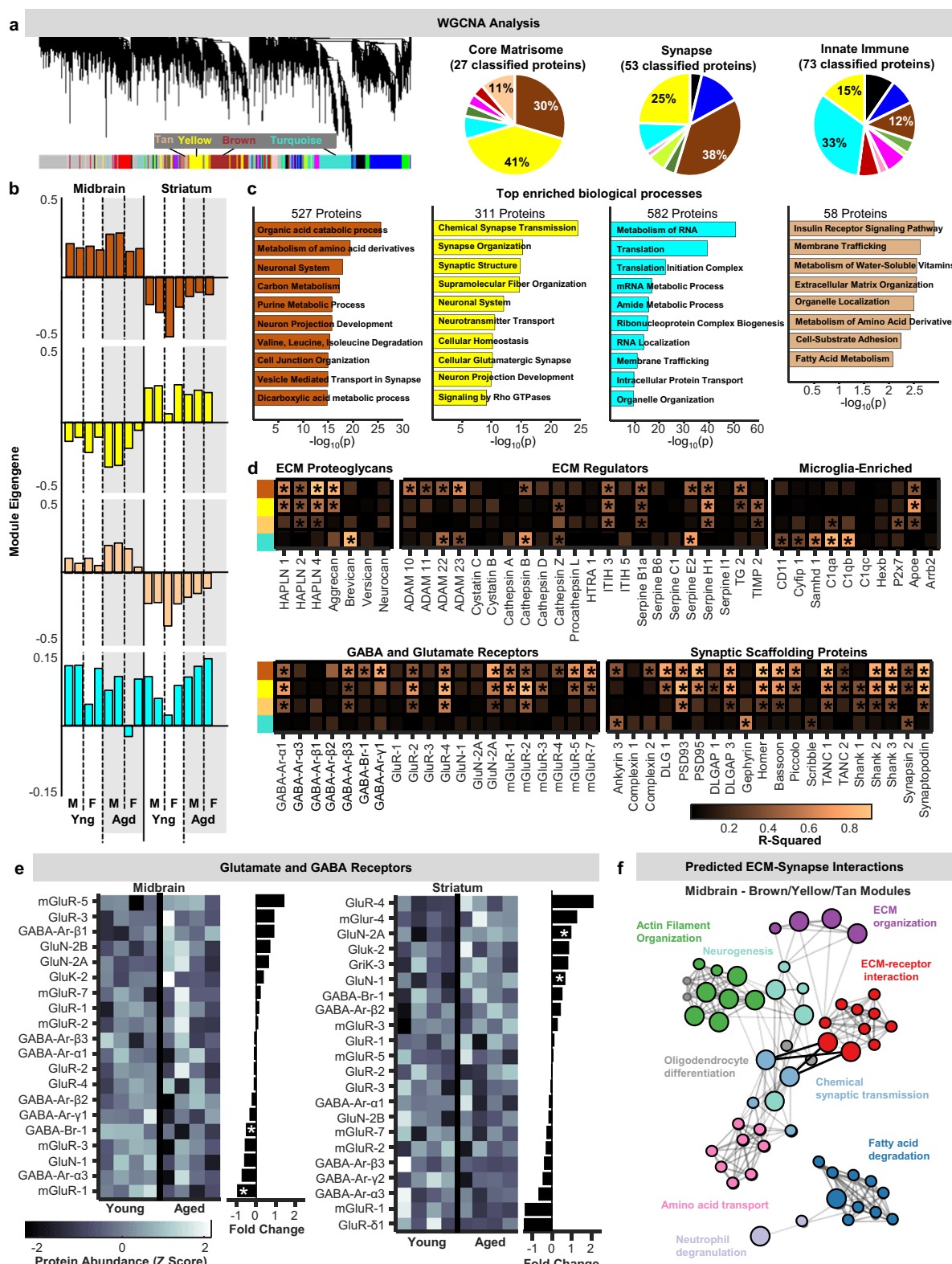

make direct contact with the hyaluronan matrix, reconstruction of microglia and hyaluronan was carried out in Imaris using tissue from young-adult (3–4 months), middle-aged (12–17 months) and aged (18–22 months) Cx3Cr1^EGFP/+ mice, which enable precise visualization of microglial morphology. VTA microglia in young mice made relatively regular putative contacts with hyaluronan fragments, both along their processes and proximal to microglial somas. Quantification of the density of hyaluronan contacts normalized to GFP signal revealed an age-associated decrease in microglia-hyaluronan contacts in the VTA (Fig. 4d). This raises the possibility that loss of the ability of VTA microglia to interact directly with hyaluronan networks is related in some way to greater hyaluronan deposition within the tissue. Importantly, an emerging body of evidence indicates that Cx3Cr1^EGFP/+ microglia exhibit some phenotypic differences from WT microglia, including expression of some ECM-relevant genes (Fig. S5)[43–45]. Thus, future work using distinct microglia reporter lines, immunostaining

**Fig. 2 | Relationships between extracellular matrix and synapse protein abundances in tissue proteomes. a** Left: dendrogram generated by Weighted Gene Coexpression Network Analysis (WGCNA) of all midbrain and striatal proteomic samples. Right: pie charts of classified core matrisome, synapse, and innate immune protein distributions across modules. **b** Module eigengenes for the brown, yellow, tan, and turquoise modules for each sample. One aged female was classified as an outlier by the WGCNA and was removed from subsequent analyses. **c** Enriched biological terms associated with brown, yellow, tan, and turquoise module proteins. **d** Heatmaps representing correlations (r² values; linear regression) between brown, yellow, tan, and turquoise module eigengenes and abundances of specific

extracellular matrix proteoglycans, complement proteins, GABAergic receptors, glutamatergic receptors, and synaptic scaffolding proteins. **e** Heat maps of z-scored protein abundances and bar plots of corresponding fold-changes with age for all detected glutamate and GABA receptors (* represents $p < 0.05$; unpaired t-test; two-sided). **f** Network plots of process and pathway enrichment terms associated with all midbrain tan/yellow/brown module proteins whose abundances were modulated by age ($p < 0.05$; Fisher's exact test; one sided). Source data are provided in the file Source Data - Fig. 2 and all statistics are provided in Supplementary Data 6. * represents $p < 0.05$.

approaches in WT mice that faithfully label microglial fine distal processes, and super-resolution microscopy will be needed to expand these analyses of microglial-ECM contact.

## Young-adult VTA microglia attenuate ECM deposition

To probe observed correlations between ECM deposition and microglial properties in the VTA, we examined tissue from young-adult (3-4 months) microglia-deficient mice (Csf1r$^{\Delta FIRE/\Delta FIRE}$ mice; Fig. 4e)[46]. Compared to control mice, Csf1r$^{\Delta FIRE/\Delta FIRE}$ mice exhibited higher WFA deposition within the VTA, but no difference in Homer2 densities (Fig. 4f, g), as observed in aging WT mice (Fig. 3). Together with findings that the ECM accumulates with age in the VTA (Fig. 3), these observations suggest that young-adult microglia attenuate ECM deposition in the VTA and that this microglial function is diminished with normative aging. While elevated WFA deposition was not associated with altered numbers of postsynaptic structures in either aging WT or microglia-deficient mice, these changes likely impact proteoglycan deposition around synapses and modify capacity for synapse plasticity[5].

Next, we sought to examine the ECM in a context where microglia are present but undergoing distinct aging trajectories. Cx3Cr1 deficiency has been suggested to alter microglial aging phenotypes[31,43], impact synaptic plasticity[47], and regulate ECM composition in other bodily tissues[48,49]. Indeed, when we mined a published RNAseq dataset[43] of microglia from young-adult (2 months) and middle-aged (12 months) WT and Cx3Cr1-deficient (KO) mice, we found that over 50 matrisome-related genes (both structural ECM proteins and ECM regulatory proteins) were differentially expressed in 2mo KO microglia compared to 2mo WT microglia (Fig. S5). Critically, more ECM-relevant genes were altered during aging in WT microglia compared to the number that were altered during aging in KO microglia, indicating that Cx3Cr1-deficiency perturbs ECM-related aspects of microglial aging. The most robust difference between genotypes was an age-related upregulation of genes associated with negative regulation of peptidase and hydrolase activity in WT microglia that was absent in KO microglia (Figs. 4h, S5). Altogether, these results indicate that Cx3Cr1-deficiency is a suitable manipulation to probe how altered microglial aging trajectories impact the ECM.

Via immunostaining, we found that VTA microglia from KO (Cx3Cr1$^{EGFP/EGFP}$; KO) mice exhibited reduced morphological complexity compared to Cx3Cr1-heterozygous (Cx3Cr1$^{EGFP/+}$, HET) mice, confirming that this manipulation enhances features of VTA microglial aging that we have reported previously (Fig. S5)[31]. We then examined ECM (WFA) and synapse abundance (Homer2) in the VTA of young-adult (3-4 months) and middle-aged (12–18 months) WT and KO mice. WFA tissue coverage was significantly greater in KO mice compared to WT both in young-adulthood and middle-age (Fig. 4i), and this same effect was also observed in the NAc (Fig. S6). Moreover, age-related ECM accumulations observed in WT mice were absent in KO mice (Figs. 3, 4i), further supporting the hypothesis that Cx3Cr1-defiency alters ECM-related aspects of microglial aging phenotypes. Hyaluronan abundance was not significantly different between WT and KO mice at any age (Fig. S6), suggesting that effects of Cx3Cr1-deficiency on ECM regulation preferentially impact some ECM components over

others. KO mice exhibited significant reductions in Homer2 density by middle age, unlike WT mice where synapse numbers were stable into late middle age (Figs. 3, 4j). Analysis of colocalization between microglia, hyaluronan, and homer2 in KO mice indicated that putative microglial hyaluronan engulfment was not different with age (Fig. S6). Putative microglial synapse engulfment was also not different in young-adult and middle-aged KO mice (Fig. S6), indicating that age-associated reductions in synapse density in these mice do not arise from greater microglial synapse engulfment. Together, these results indicate that WT microglia have an attenuating effect on ECM deposition relative to contexts where microglia are absent or altered, and that during normative aging, this microglial function is lost to some degree. It will be critical to replicate these observations using conditional microglial manipulations, as some findings may be influenced by developmental compensations that arise in constitutive models.

## Hyaluronan and synapse remodeling in the VTA aligns with reward-based memory in middle-aged mice

Healthy brain aging is an active process that engages endogenous mechanisms of plasticity to protect circuit function as aging-related challenges emerge[50]. To begin understanding whether ECM accumulations in the aging VTA represent vulnerabilities or adaptive responses that support continued circuit function, we developed a behavioral paradigm that engages reward circuitry and probes aspects of dopamine-relevant cognition known to be altered with aging, including reward memory and cognitive flexibility (Fig. 5a)[29,51]. In this task, mice learn to explore a large arena and forage for palatable food rewards that change location daily. During testing, mice encode a rewarded location and, after 2 or 24 h, reenter the arena with 4 unrewarded feeders to measure their memory of the rewarded location (probe trials; Fig. 5a). Mice then immediately re-enter the arena and are allowed to consume food reward in the previously rewarded location to minimize extinction of training (Fig. 5a). In total, mice undergo 5 weeks of training/testing (5 days/week), which allows for robust assessment of cognitive status of individual mice and minimizes behavioral variables like novelty and stress at the time when tissue is collected for histological examination.

Both young-adult (4 months) and late-middle-aged (18 months) non-food-restricted mice learned the task and exhibited consistent foraging at similar points of the experiment timeline (Figs. 5b, S7). Average walking speeds, start box exit latencies, and proportion of time on the arena perimeter did not differ between age groups, indicating similar levels of task engagement and absence of prominent age-related differences in anxiety-like behavior (Fig. 5b). During probe sessions, young-adult mice showed similar performance that was better than chance following both 2- and 24 h delays. Middle-aged mice performed comparably to young-adults with a 2 h delay but made fewer correct feeder visits and more errors following 24 h delays (Fig. 5c). Critically, we observed higher variability in middle-aged mice during 24 h probe trials compared to young, which is a hallmark of cognitive aging[52]. Hence, this behavioral paradigm establishes a strategy to probe links between cellular/molecular features of mesolimbic

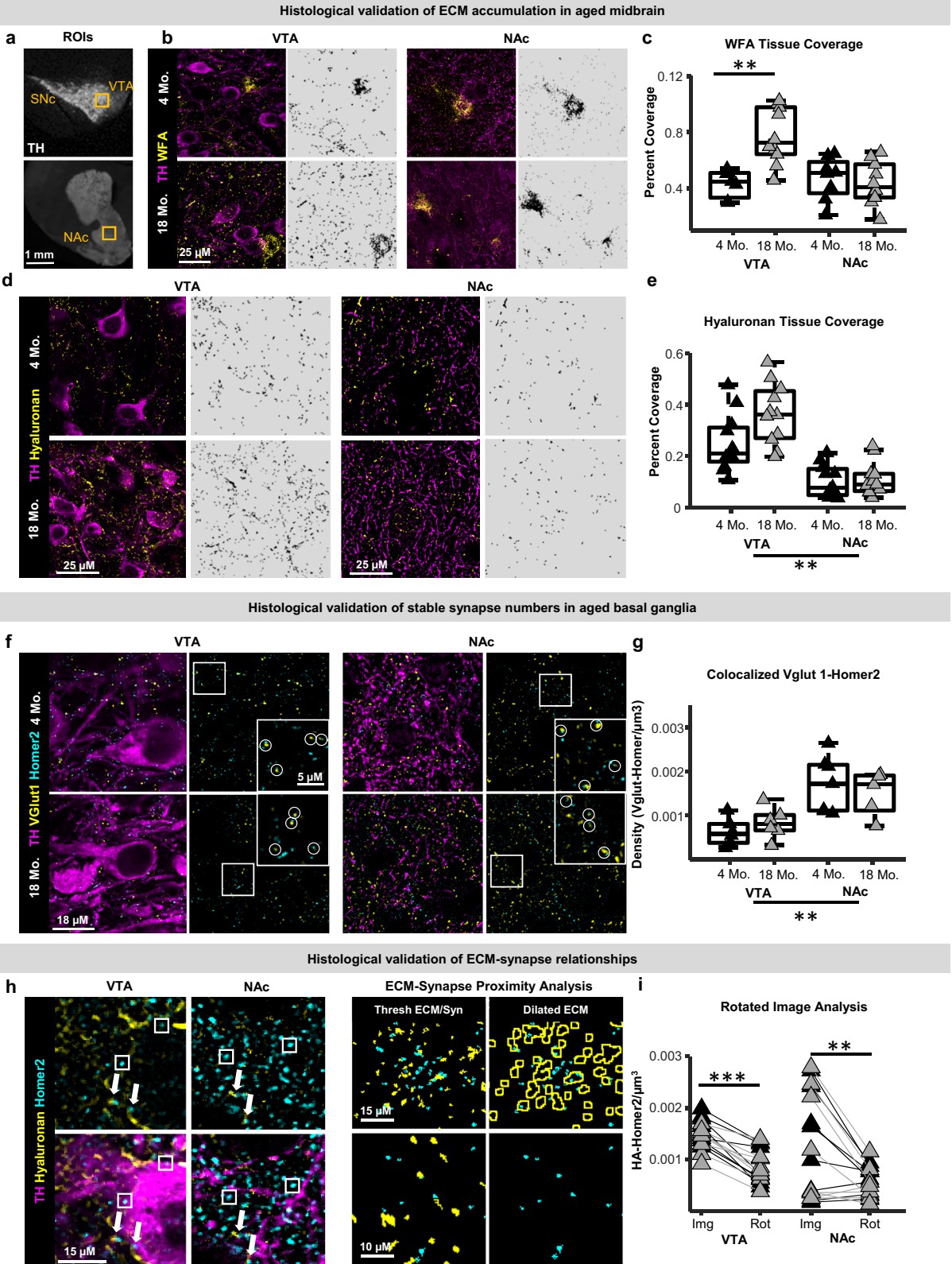

**Histological validation of ECM accumulation in aged midbrain**

a ROIs | b VTA | NAc | c WFA Tissue Coverage

d VTA | NAc | e Hyaluronan Tissue Coverage

**Histological validation of stable synapse numbers in aged basal ganglia**

f VTA | NAc | g Colocalized Vglut 1-Homer2

**Histological validation of ECM-synapse relationships**

h VTA | NAc | ECM-Synapse Proximity Analysis | i Rotated Image Analysis

dopaminergic circuits and age-associated changes in reward-based cognition.

To enable assessment of the impact of the 5-week behavioral training itself, each cohort of mice included sedentary controls housed in the same vivarium as behaving mice for the duration of the experiment (Fig. 5e). Compared to sedentary mice, behavior-trained mice exhibited significantly less hyaluronan within the VTA (Fig. 5f, g),

particularly at middle-age. Hence, the accumulation of VTA hyaluronan observed in aging sedentary animals appeared to be mitigated by engaging in behavioral training. Furthermore, middle-aged mice with lower hyaluronan densities showed better task performance (Fig. 5h), indicating that this remodeling is beneficial. Behavior-induced hyaluronan remodeling was not observed in multiple other brain regions examined, including the NAc, mPFC, and retrosplenial cortex (Fig. S7).

**Fig. 3 | Histological analysis of extracellular matrix and synapse proteins in the aging mesolimbic dopamine system. a** Areas within the ventral tegmental area (VTA) and nucleus accumbens (NAc) histologically examined. Experiments were performed once on a single cohort of mice. **b** Photomicrographs of tyrosine hydroxilase (TH) and Wisteria floribunda agglutinin (WFA) from a young-adult (4 months) and late-middle-aged (18 months) mouse (left). Right: binarized images used for quantification. **c** Boxplots depicting VTA and NAc WFA field-of-view coverage (%) in young-adult (black; $n = 8$ mice) and late-middle-aged mice (gray; $n = 10$ mice; n-way ANOVA; post-hoc Tukey-Kramer; two-sided; Age: p = 0.026). **d** Photomicrographs of hyaluronan and TH in young-adult and late-middle-aged mice (left). Right: binarized images used for quantification. **e** Boxplots depicting VTA and NAc hyaluronan filed-of-view coverage in young-adult (black; $n = 12$ mice) and late-middle-aged (gray; $n = 12$) mice (n-way ANOVA; post-hoc Tukey-Kramer; two-sided; Age: $p = 0.23$). **f** Photomicrographs of Homer2, VGlut1, and TH in young-adult and middle-aged mice. Insets depict the field of view outlined by the white squares. Circles delineate synapses with presynaptic and postsynaptic elements. **g** Boxplots of VTA and NAc homer2 densities (n-way ANOVA; post-hoc Tukey-Kramer; two-sided; Age: $p = 0.98$; $n = 7$ young and $n = 6$ aged mice). **h** Photomicrographs of hyaluronan, homer2, and TH in the VTA and NAc of a young-adult mouse. Bottom panels are equivalent to top panels with the addition of the TH channel. Arrows highlight putative hyaluronan-homer2 colocalized puncta and squares depict homer2 puncta not associated with hyaluronan. Right: schematic ECM-synapse proximity analysis. Experiments were performed once on a single cohort of mice. **i** Densities of homer2 within $0.5 \mu m$ of hyaluronan compared to when rotating 1 channel 90 degrees (paired t-test; two-sided; VTA: $p = 0$; NAc: $p = 0.003$; $n = 8$ young and $n = 8$ aged mice). In all boxplots, boxes represent the interquartile range (IQR; 25–75 percentiles), middle lines the median, and whiskers extend $\pm 1.5 \cdot IQR$. Source data are provided in the file Source Data - Fig. 3 and all statistics are provided in Supplementary Data 6. * represents $p < 0.05$, ** represents $p < 0.01$, *** represents $p < 0.001$ for each statistical test.

Behavior-associated hyaluronan reductions were observed in the substantia nigra pars compacta and hippocampus, suggesting that hyaluronan remodeling occurs only in specific circuits when animals repeatedly engage with reward-based spatial memory tasks.

To link behavior-induced VTA hyaluronan remodeling with synapse status, Homer2 was also analyzed in these mice. Compared to sedentary mice, behavior-trained mice had elevated Homer2 puncta densities, indicating that behavioral training/testing had net synaptogenic effects (Fig. 5i, j). Surprisingly, however, behavior-trained mice with fewer synapses showed better task performance (Fig. 5k), suggesting that synapse refinement that impacts performance may also occur across this 5-week paradigm. In sedentary mice, hyaluronan tissue coverage and Homer2 puncta densities were not correlated in either young-adult or late-middle-aged mice. Importantly, however, a significant positive correlation between hyaluronan tissue coverage and homer2 densities was observed in middle-aged mice that underwent behavioral training, but not in the young adults (Fig. 5l). Furthermore, behavior-trained middle-aged mice with more hyaluronan tissue coverage had a higher density of Homer2 puncta within $0.5 \mu m$ of hyaluronan (see ECM-synapse proximity analysis; Fig. 3h), and again this relationship was not seen in young-adult mice or in sedentary mice at either age (Fig. 5m). Together, these experiments suggest that greater VTA hyaluronan abundance may stabilize excitatory synapse numbers but limit synaptic refinements that optimize reward-driven behavior in aging mice.

## Proteomic signatures of cognitive phenotypes in middle-aged mice

Because histochemistry cannot provide comprehensive quantitative information on large families of proteins, we sought to use tissue proteomics to identify matrisome and synapse protein expression patterns associated with cognitive function in aging mice. To this end, young-adult (4 months) and late-middle-aged (18 months) WT mice were tested on 3 standard mouse behavioral paradigms: an open field test of anxiety-like behavior, a novel object recognition (NOR) test of non-spatial recognition memory, and a T-maze test of spontaneous alternation behavior (Fig. 6a). These 3 behaviors were selected due to their relatively high-throughput nature, allowing for a more rapid assessment of cognitive status while minimizing remodeling effects of extended behavioral training, and because this behavioral battery will be more easily replicated across laboratories within different experimental contexts.

Middle-aged mice on average exhibited lower discrimination and alternation on the NOR and T-maze tasks, respectively, both of which are indicative of poorer performance. As in the foraging paradigm (Fig. 5), middle-aged mice also exhibited higher variability in performance across tasks. Leveraging this variability, we fed estimates of anxiety-like behavior from the open field test, discrimination indices from the NOR test, and alternation indices from the T-maze into an

unbiased hierarchical clustering algorithm to cognitively classify all mice tested in this pipeline. Importantly, while anxiety-like behavior does not directly reflect cognitive abilities per se, these measures were included to account for potential confounding effects of anxiety on exploration-based cognitive tasks such as the NOR and T-maze tests[53,54]. This analysis resulted in two parent clusters, one containing 75% of the young-adult mice and roughly half (46%) of the middle-aged mice, and another cluster containing the remaining animals. Using these clusters, we categorized the mice into 3 groups: young average, middle-aged unimpaired, and middle-aged impaired (Fig. 6a). Midbrains were harvested from mice in each age and cognitive group and subjected to solubility-based subcellular fractionation (Fig. 1), and membrane, cytoskeletal, and insoluble fractions were analyzed via mass spectrometry.

Principal component analysis (PCA) of entire proteomes from each fraction revealed the emergence of group separations between middle-aged impaired and unimpaired mice (Fig. S8), suggesting links between the overall midbrain proteome and the cognitive status of middle-aged mice. When restricting PCA analysis only to matrisome proteins, more prominent separations with respect to age and cognitive status were apparent across subcellular fractions (Fig. 6b), supporting the idea that midbrain ECM status shapes cognition during aging. Among ECM subfamilies, proteoglycans showed more robust group separations in membrane and cytoskeletal fractions, while glycoproteins showed the greatest separation in insoluble fractions (Fig. S8). These observations suggest that both the abundance and subcellular localization/solubility of proteoglycans play roles in maintaining cognitive function with advanced age.

Proteoglycan abundance in membrane and cytoskeletal fractions was substantially higher in middle-aged mice compared to young (Fig. 6c; Fig. S8). In membrane fractions, proteoglycan abundance was also significantly greater in middle-aged impaired mice compared to unimpaired (Fig. 6c), whereas this separation in proteoglycan abundance with respect to cognitive status was not observed in either the cytoskeletal or insoluble fractions (Fig. S8). To evaluate what specific proteins drive these relationships, regression analyses between the abundances of individual proteoglycans, age, and cognitive status were performed. This approach revealed that the predominant proteoglycans driving separations between middle-aged impaired and unimpaired mice were multiple HAPLNs (proteins that anchor proteoglycans to hyaluronan) and chondroitin sulfate proteoglycans (e.g., aggrecan, brevican, etc.; Fig. 6c). Consistent with our prior proteomic analyses, many of these proteins increased in abundance during aging and showed significant positive correlations with age. Moreover, supporting the idea that this accumulation is not beneficial cognitively, many of these same proteins exhibited significant negative correlations with behavioral performance. As observed in our prior proteomic dataset, the abundance of most synaptic proteins in these tissues did not change with age (Figs. 6d, S9), although multiple neurotransmitter

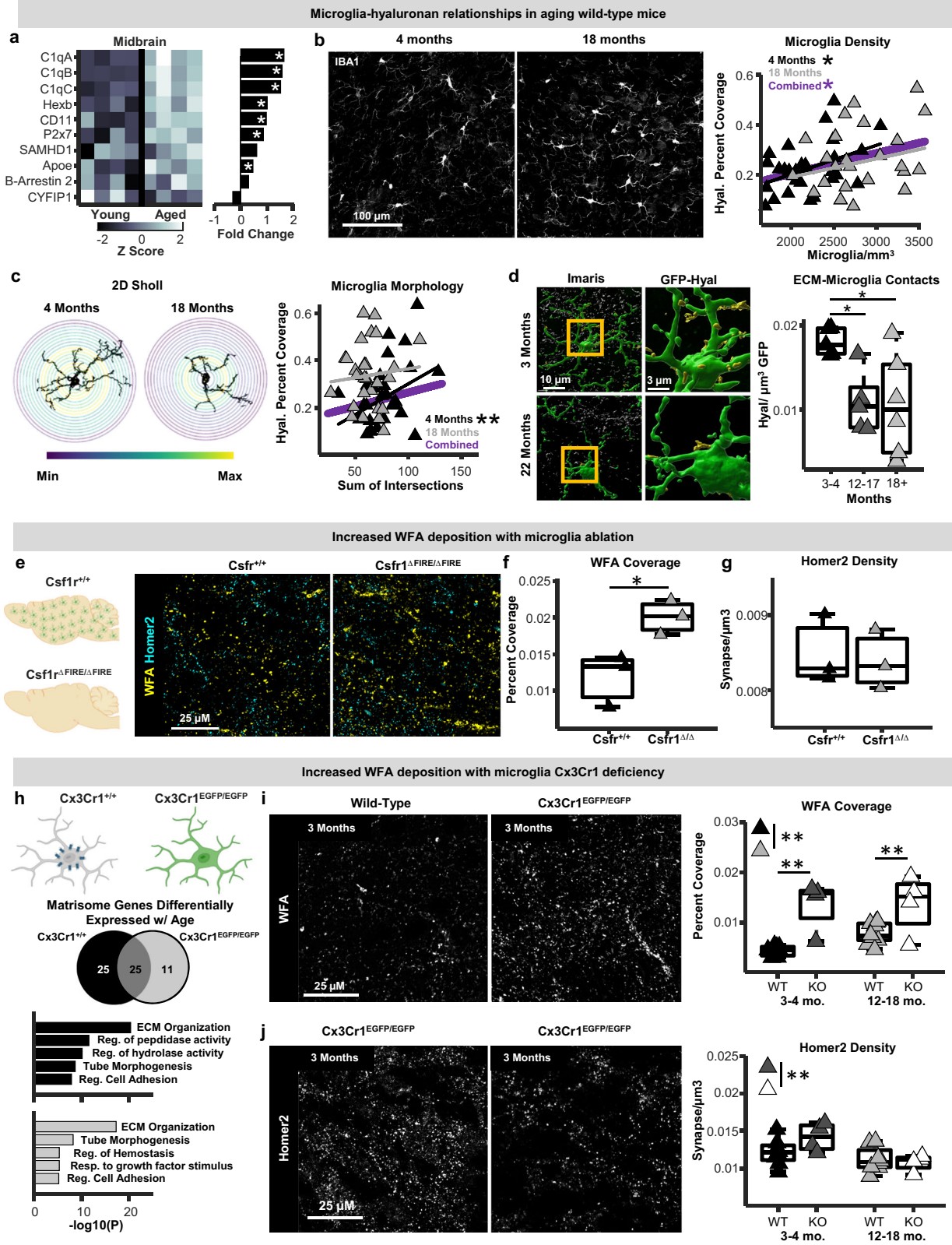

receptors were negatively correlated with behavioral performance, indicating that middle-aged mice with less synaptic protein performed better on the cognitive battery (Figs. 6d, S9). This data provides an independent verification that having lower ECM proteoglycan and excitatory synapse abundances in the midbrain is cognitively beneficial for middle-aged mice and identifies the hyaluronan-proteoglycan matrix as a modulator of cognitive aging phenotypes.

## Synapse and ECM abundance on dopamine neurons map onto cognitive aging phenotypes

Proteomic mapping in cognitively classified mice indicated that hyalectan abundances align with cognitive phenotypes in middle-aged mice (Fig. 6). In the subset of cognitively characterized mice (Fig. 6a) that did not undergo proteomic analysis, we sought to independently validate this finding via immunohistochemistry for HAPLN1, an ECM

**Fig. 4 | Microglia-extracellular matrix relationships in aging wildtype and microglial mutant mice. a** Z-scored protein abundances and fold-changes with age for microglia-enriched proteins within the midbrain proteomic dataset (* denotes $p < 0.05$; unpaired t-test; two-sided). **b** Left: photomicrographs of IBA1-positive microglia in the ventral tegmental area (VTA) of young-adult (4 months) and late-middle-aged (18 months) mice. Experiments were performed once on a single cohort. Right: scatter plots of relationships between VTA microglia densities and hyaluronan coverage (robust regression; two-sided; Young: $p = 0.027$; Middle-aged: $p = 0.26$). **c** Left: binarized VTA microglia and Sholl radii. Right: scatter plots of relationships between microglia complexities and hyaluronan coverage (robust regression; two-sided; Young: $p = 0.00028$; Middle-aged: $p = 0.51$). **d** Left: reconstructions of putative microglia-hyaluronan contacts. Right: Boxplots of microglia-hyaluronan contact densities (n-way ANOVA; post-hoc Tukey-Kramer; two-sided; $p = 0.011$; $n = 5$ young; $n = 5$ middle-aged; $n = 6$ aged mice). **e** Photomicrographs of wisteria floribunda agglutinin (WFA) and Homer2 from 3-month Csf1r$^{\Delta FIRE/\Delta FIRE}$ and Csf1r$^{+/+}$ mice. Schematic created in BioRender. Gray, D. (2025) https://BioRender.com/tr8bq45. **f** VTA WFA coverage and **g** Homer2 densities in Csf1r$^{+/+}$ (black; n = 3 mice) and Csf1r$^{\Delta FIRE/\Delta FIRE}$ mice (gray; $n = 3$ mice; unpaired t-test; two-sided; WFA:

$p = 0.029$; Homer2: $p = 0.53$). **h** Top: ECM and synapses were examined in young-adult (3-4 months) and early-middle-aged (12-15 months) Cx3Cr1-knockout (Cx3Cr1$^{EGFP/EGFP}$) mice. Schematic created in BioRender. Gray, D. (2025) https://BioRender.com/6vjywib. Middle: pie chart of numbers of differentially expressed matrisome-related genes with age (Gyoneva et al., 2019). Bottom: enriched biological pathways associated with differentially expressed matrisome genes (Fisher's exact test). **i** Left: photomicrographs of VTA WFA from wild-type and Cx3Cr1-knockout mice. Right: Boxplots of WFA coverage in wild type (black; light gray) and Cx3Cr1-knockout mice (dark gray; white; n = 4 mice per age per genotype; n-way ANOVA; two-sided; Genotype: p = 0). **j** Left: photomicrographs of Homer2 in 3-month and 12-month-old Cx3Cr1-knockout mice. Right: boxplots of Homer2 densities in wild-type and Cx3Cr1-knockout mice ($n = 4$ mice per age group per genotype; n-way ANOVA; two-sided; Genotype: $p = 0.53$). In all boxplots, boxes represent interquartile ranges (IQR; 25-75 percentiles), middle lines the median, and whiskers extend +/- 1.5*IQR. Source data are provided in the file Source Data - Fig. 4 and all statistics are provided in Supplementary Data 6. * represents p < 0.05, ** represents p < 0.01.

link protein that directly interacts with hyaluronan, and aggrecan, an abundant chondroitin sulfate proteoglycan in the brain. HAPLN1-aggrecan complexes were found throughout the VTA, primarily near the surfaces of dopamine neurons and other neuronal or glial cells (Fig. 7a), suggesting that HAPLN1 likely helps organize hyaluronan-proteoglycan interactions within the VTA. While the field of view (FOV) HAPLN1 abundance did not differ across age, middle-aged impaired mice exhibited significantly higher HAPLN1 abundances compared to middle-aged unimpaired mice (Fig. 7b). Aggrecan abundance did not differ with age or cognitive status (Fig. 7b), although there were trends toward increased aggrecan abundance in middle-aged impaired mice. When analysis was restricted to zones within ~0.5 μm of dopamine neuron surfaces, we observed that HAPLN1 abundance was greater in middle-aged mice, and trending towards being significantly higher in impaired mice compared to unimpaired mice (Fig. 7c). Aggrecan abundance on dopamine neurons did not differ with age or cognitive status. However, the abundance of aggrecan-HAPLN1 complexes on dopamine neuron surfaces was greater in middle-aged mice and also trending towards being significantly higher in impaired mice compared to unimpaired mice ($p = 0.07$; Figs. 7c, S10). These observations suggest that local ECM proteoglycan deposition on and around dopamine neurons informs overall cognitive performance during late middle age.

To further explore the role of dopamine-neuron-localized ECM in cognitive aging, we carried out similar analyses in tissue from mice trained in our 5-week food reward foraging paradigm (Fig. 5), as well as sedentary controls (Fig. 7d). Synapse abundance, as assessed via Homer2, on dopamine neurons did not differ between sedentary and behaving mice or across age. However, middle-aged animals with fewer dopamine-neuron-localized Homer2 puncta exhibited better task performance (Fig. 7e, f), consistent with FOV analyses indicating that fewer VTA synapses correlated with better performance (Fig. 5k). Hyaluronan abundance on dopamine neurons was lower in behavior-trained mice compared to sedentary mice (Fig. S10), agreeing with FOV findings that behavior training reduces hyaluronan density (Fig. 5g). Importantly, both young and aging behavior-trained mice with more hyaluronan on dopamine neurons had significantly more Homer2 on dopamine neurons (Fig. 7g). This correlation was completely absent in sedentary mice, suggesting that behavioral training engages hyaluronan-synapse remodeling around VTA dopamine neurons, and that failure to generate and refine these complexes is associated with cognitive decline in aging. Collectively, these histological observations identify hyaluronan and proteoglycan link proteins as promising targets for future mechanistic studies of cognitive resilience vs. decline.

## Microglia and ECM abundances independently correlate with cognitive aging phenotypes

Our findings in the context of microglial depletion (Csfr1$^{\Delta FIRE/\Delta FIRE}$) and altered microglial aging trajectory (Cx3cr1$^{EGFP/EGFP}$, Cx3cr1 KO) indicate that microglia may be poised to regulate abundance of VTA hyaluronan and proteoglycans (Fig. 4). Moreover, in the context of normative aging, VTA microglial density was correlated with abundance of hyaluronan, and our results suggest that reduced microglial-ECM contact may contribute to ECM accumulations. To further probe potential links between microglial aging in mesolimbic circuits and cognition, we quantified VTA and NAc microglial densities in tissue from mice trained in the 5-week food reward foraging paradigm. As we showed previously, microglia densities were higher in the NAc compared to VTA and increased with aging only in the VTA (Fig. 8a, b)[31]. Middle-aged mice with more VTA microglia exhibited worse task performance (Fig. 8c), highlighting associations between this feature of microglial aging and cognition. Surprisingly, NAc microglia densities showed the opposite relationship, where mice with greater microglial densities exhibited better performance (Fig. 8c). This suggests that regional microglial specializations and aging phenotypes uniquely impact the neuronal circuits in which they reside.

Proteomic mapping also revealed increased abundance of multiple microglia-enriched proteins in the aged midbrain, and many of these increases were associated with age-related cognitive impairment (Fig. 8d). The strong alignment between ECM abundance and cognitive aging phenotypes (Fig. 4) prompts the question of whether microglia-ECM interactions influence cognitive aging, or whether their associations with cognition are independent of one another. To glean insights into this question, we first performed regression analyses between individual ECM proteoglycan abundances from cognitively characterized aging mice (Fig. 6), and total microglial and synapse receptor abundances in tissue proteomes from the same mice (Fig. 8e). This analysis suggested that microglial protein abundances were not strongly aligned with most ECM proteoglycan abundances, although a significant negative correlation was observed with HAPLN2. Conversely, synapse receptor abundances showed strong positive relationships with HAPLN1-4 and all chondroitin-sulfate proteoglycans detected in this tissue (Fig. 8e). To further evaluate these relationships, correlation network plots that integrate cognitive performance with microglial, ECM, and synapse abundances (measured via proteomics and histology) were created. In these plots, nodes represent individual features, and line thicknesses represent r values of pairwise correlations between traits. In young-adult mice, microglia abundances were strongly correlated to both ECM and synapse abundances, both of which showed relatively pronounced correlations with cognition

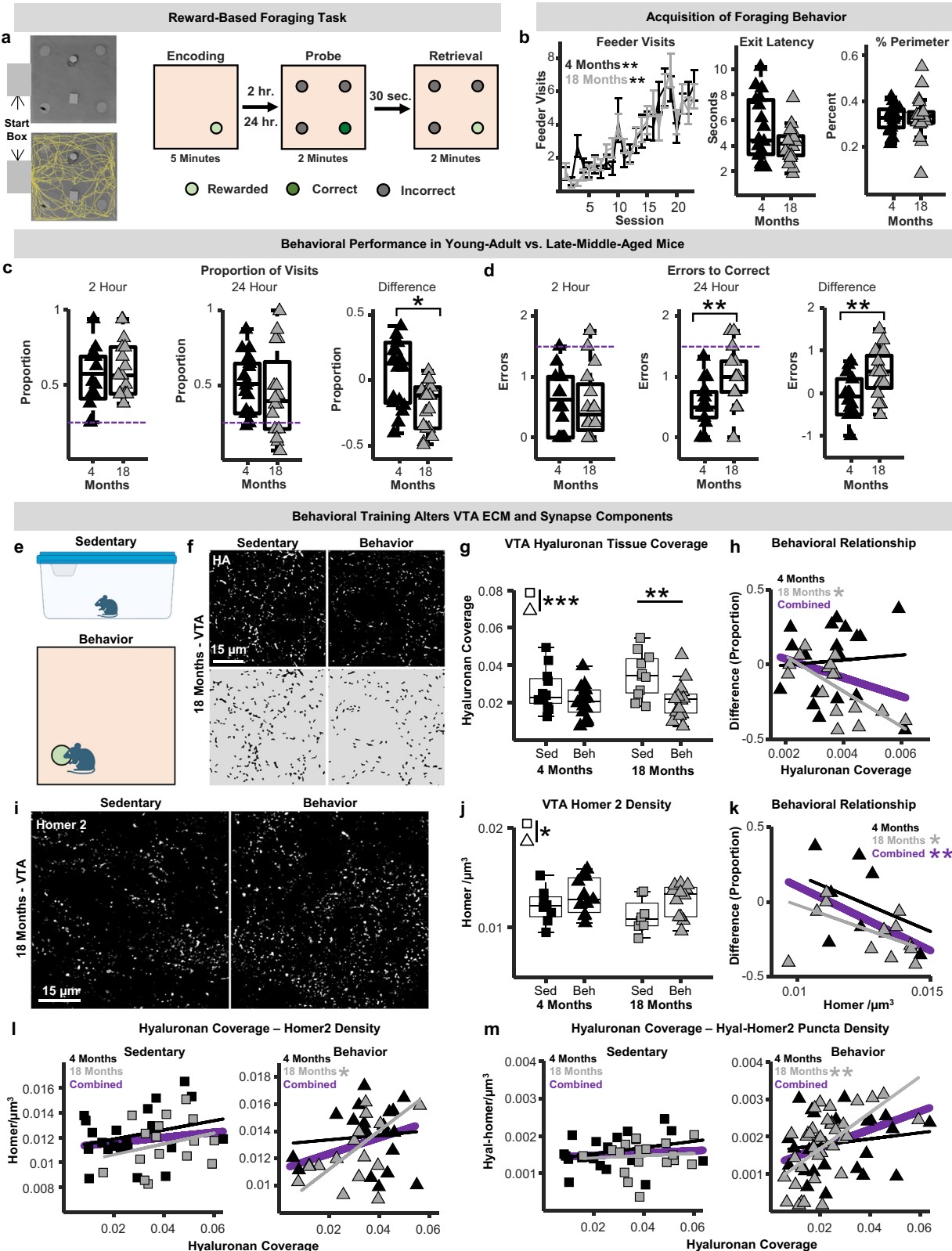

(Fig. 8f). In middle-aged mice, however, microglial relationships with both ECM and synapse abundances were reduced, and ECM-synapse and ECM-cognition relationships were higher (Fig. 8f). Taken together, these analyses suggest that microglia and the ECM somewhat independently influence cognitive aging trajectories, and that ECM-synapse dynamics become more central to cognitive processing during normative aging.

## Discussion

As the predominant structure occupying the extracellular space, the ECM is positioned to play central roles in almost all neurological processes. Yet, brain ECM research remains in its early stages, and optimal strategies for observing and measuring the ECM's complexity are still being defined. Revealing how the ECM impacts discrete brain structures—such as synapses—is a challenge for the field. ECM near synapses

**Fig. 5 | Behavior-induced extracellular matrix and synapse remodeling in the ventral tegmental area. a** Schematic of foraging arena and behavioral paradigm. **b** Right: number of feeder visits across sessions in young-adult (4 months; black; $n = 18$ mice) and late-middle-aged (18 months; gray; n = 18 mice) mice (repeated-measures ANOVA; two-sided; Age: p = 0.28). Data presented as mean +/- SEM. Middle: latencies to exit start box. Right: proportion of time on arena perimeter. **c** Proportion of correct feeder visits during probe trials (left and middle), and within-subject difference measures (24 h-2 h; n = 18 young and 18 middle-aged mice; n-way ANOVA; two-sided; $p = 0.028$). **d** Number of errors to correct during probe trials (left and middle), and within-subject difference measures ($n = 18$ young and 18 middle-aged mice; n-way ANOVA; two-sided; $p = 0.0067$). **e** VTA hyaluronan and Homer2 were assessed in young-adult and late-middle-aged behavior-trained and sedentary mice. Schematic created in BioRender. Gray, D. (2025) https://BioRender.com/kpedggq. **f** Top: photomicrographs of hyaluronan from sedentary and behavior-trained 18-month old mice. Bottom: binarized images used for quantification. **g** Boxplots of hyaluronan coverage in sedentary (squares) and behavior-trained mice (triangles; n = 12 young-sedentary, $n = 10$ young-behavior, $n = 12$ middle-aged-sedentary, n = 13 middle-aged-behavior mice; n-way ANOVA; two-sided; Behavior: p = 0.00094). **h** Relationship between hyaluronan coverage and foraging task performance (robust regression; two-sided; Young: p = 0.57; Middle-aged: p = 0.015). **i** Photomicrographs of Homer2 from sedentary and behavior-trained 18-month-old mice. **j**) Boxplots of VTA Homer2 densities (n = 8 young-sedentary, $n = 10$ young-behavior, 8 middle-aged-sedentary, 12 middle-aged-behavior mice; n-way ANOVA; two-sided; Behavior: $p = 0.029$). **k** Relationship between Homer2 densities and foraging task performance (robust regression; two-sided; Young: $p = 0.15$; Middle-aged: $p = 0.047$). **l** Relationships between hyaluronan coverage and Homer2 densities in sedentary and behavior-trained mice (robust regression; two-sided; Young Sedentary: p = 0.25; Middle-aged Sedentary: $p = 0.38$; Young Behavior: p = 0.075; Middle-aged Behavior: p = 0.015). **m** Relationships between hyaluronan coverage and densities of Homer2 within 0.5 μm of hyaluronan (robust regression; two-sided; Young Sedentary: p = 0.25; Middle-aged Sedentary: p = 0.74; Young Behavior: p = 0.41; Middle-aged Behavior: p = 0.0004). In all boxplots, boxes represent interquartile ranges (IQR; 25–75 percentiles), middle lines represent medians, and whiskers extend ± 1.5*IQR. Source data are provided in the file Source Data - Fig. 5 and all statistics are provided in Supplementary Data 6. * represents p < 0.05, ** represents p < 0.01, *** represents p < 0.001.

can regulate AMPA receptor diffusion, positioning of neuronal pentraxins[42,55], extracellular ion concentrations, and access of phagocytic cells to synaptic elements[26,56]. Moreover, the "sweet spot" of optimal ECM abundance near synapses depends on context; appropriate ECM deposition may protect against synapse loss, but targeted ECM degradation is also essential for structural plasticity in support of learning and memory[7]. Advancing knowledge in this area is likely to reshape our understanding of synapse regulation in the aging brain and illuminate novel approaches to manipulate the brain ECM in support of healthy circuit function.

This report presents a comprehensive proteomic mapping of the brain ECM during normative aging and reveals regional heterogeneity in ECM composition and age-associated remodeling across different basal ganglia nuclei. This argues that findings about the ECM in one brain region cannot be generalized to other brain regions and that similar ECM mapping of additional brain regions in a wide variety of contexts (CNS development, brain injury, brain cancer, neurodegeneration, etc.) is needed. In general, basal ganglia matrisome protein abundance increased during aging, and histological examination of ECM proteoglycans and glycosaminoglycans confirmed this increase, aligning with previous biochemical and gene expression studies[57,58]. In the same tissue, we found no evidence of substantial aging-related synapse loss in the midbrain or striatum across multiple histological and proteomic datasets. This differs from previous reports that have shown age-related synapse loss in forebrain regions, including the prefrontal cortex and hippocampus[1,59,60]. These discrepancies could either reflect true brain region differences or that the present study primarily focused on middle-aged rather than geriatric mice. In terms of regional heterogeneity, the NAc had significantly more excitatory synapses and less ECM deposition compared to the VTA. This raises the possibility that NAc networks are more malleable, as the ECM is thought to restrict plasticity since, for example, the closing of developmental critical periods aligns with drastic increases in ECM abundance[61]. Furthermore, it may suggest that, although excitatory VTA synapse numbers remain stable with age, those synapses may also become increasingly rigid as VTA ECM deposition increases during aging.

Indeed, our data reveals multiple lines of evidence implicating the ECM as a regulator of basal ganglia synapse function even in the absence of any changes to synapse number. For example, unbiased computational analyses of our proteomic data revealed that abundance of specific synaptic proteins aligned with WGCNA modules containing ECM proteins, arguing that even in the absence of changes in synapse number, there is a critical role for the ECM in shaping synapse structure and composition. Histological analyses also revealed

close anatomical proximity between postsynaptic markers (homer2) and hyaluronan, consistent with a structural and potentially functional relationship between ECM components and synaptic elements. Surprisingly, this analysis also revealed that synaptic protein abundances were not aligned with WGCNA modules containing complement proteins. Thus, while microglial synapse engulfment through complement tagging has been implicated in numerous disease contexts[23,62,63], our data argues that local ECM status plays more prominent roles in regulating synapses during healthy brain aging. An important future direction for this research will be to examine relationships between basal ganglia ECM, immune, and synapse protein status using isolated synaptosomes, which allow for a more targeted quantification of synapse-associated proteomes[64]. Additionally, it will be crucial to directly measure synaptic activity and capacity for synaptic plasticity, and relate these measures to aging-related changes in ECM abundance and composition.

Our observation that middle-aged mice with fewer excitatory synapses exhibited better cognitive performance across both naturalistic foraging behaviors (paired with histology) and high-throughput behavioral assays (OF, NOR, T-maze - paired with proteomics) was somewhat unexpected. These observations challenge prevailing assumptions that more synapses automatically equates to better cognition[4,35] and suggest that positive cognitive aging outcomes depend on the ability to refine and remodel synaptic networks. In support of this idea, we observed that repeated engagement in naturalistic foraging-based behavior had a net synaptogenic effect for both young and late-middle-aged mice. However, mice that performed best on the foraging tasks had fewer excitatory postsynaptic puncta, implying that the capacity to remodel rather than retain synapses is critical for optimal circuit function. In both behavioral experiments, greater synaptic protein levels and poorer cognitive performance were also associated with greater ECM abundance. These findings support a model in which excessive ECM accumulation may constrain the synapse remodeling and pruning required for optimal cognition[6,7,65,66], and extend this framework into the context of normative brain aging. Moreover, these findings carry important implications for pathological aging contexts, including presymptomatic neurodegeneration or recovery from brain injury, where compensatory synaptogenic responses have been observed[67–69]. Our data argue that if these newly formed synapses are not appropriately refined, they may, at best, fail to facilitate appropriate circuit activity, and, at worst, exacerbate network dysfunction and cognitive decline through the formation/retention of aberrant or inefficient synaptic connections. Nonetheless, an important limitation of our study is the lack of information about the source and identity of synaptic inputs into the VTA and midbrain.

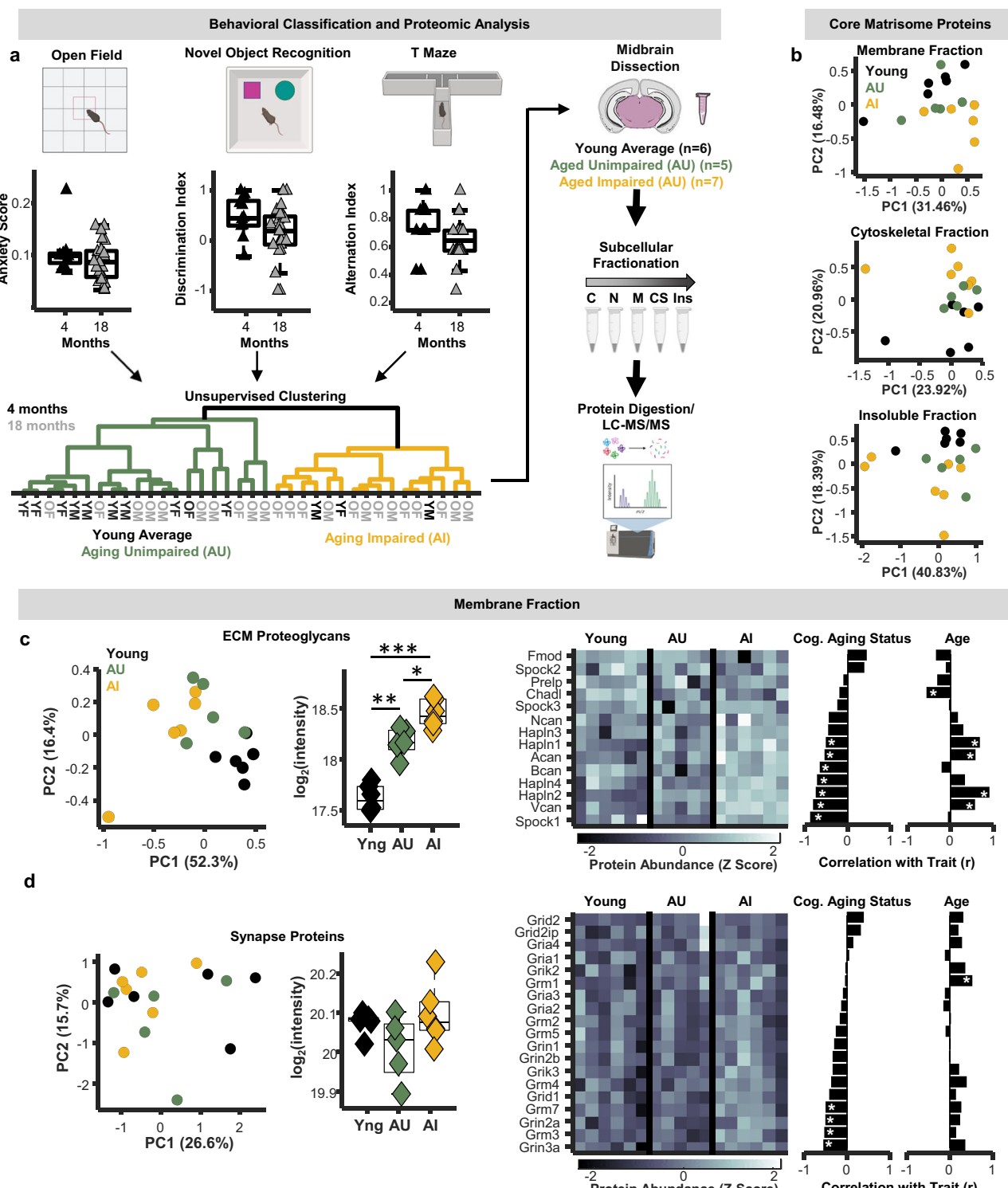

Future studies leveraging connectivity mapping and in vivo functional imaging will be essential to fully understand how ECM-driven synaptic dynamics contribute to circuit-level adaptations and cognitive resilience in aging.

One unique feature of the VTA is that microglia within this region exhibit accelerated aging phenotypes, characterized by increases in proliferation and inflammatory factor production, compared to microglia in other basal ganglia nuclei[31]. Here, we provide replication of these VTA microglial aging patterns and build on this work by identifying an additional feature of midbrain microglial aging - a loss of capacity to contact and regulate the ECM. Several key findings support

this hypothesis. First, while microglia densities and morphologies were significantly correlated with ECM deposition in young-adult mice, these relationships were lost by late-middle age. Next, constitutive microglial depletion phenocopied normative age-associated increases in ECM abundance, arguing that microglia typically restrict ECM deposition and that this ability is lost during aging. Finally, perturbing microglial aging trajectories via deletion of Cx3Cr1 resulted in a failure to upregulate ECM-regulatory genes that are upregulated in aging WT microglia[43], resulting in greater ECM deposition compared to WT mice, and a loss of excitatory postsynaptic puncta by early middle age. These findings align with recent work showing that microglia regulate ECM

**Fig. 6 | Proteomic signatures of cognitive aging phenotypes. a** Schematic of behavioral classification and proteomic analysis. Young-adult (4 months; $n = 12$; black triangles) and late-middle-aged (18 months; $n = 24$; gray triangles) male and female mice were tested on open field, novel object recognition, and T Maze tasks. Schematic created in in BioRender. Gray, D. (2025) https://BioRender.com/h5muudq. Data from all 3 behaviors were used to perform hierarchical clustering analyses to delineate young-average (black), aging-unimpaired (AU; green), and aging-impaired (AI; yellow) mice. Midbrain tissue from behaviorally-classified mice then underwent subcellular fractionation followed by proteomic analysis. **b** PCA plots of all core matrisome proteins from membrane, cytoskeletal, and insoluble fractions. **c** PCA plot of extracellular matrix proteoglycans from the membrane fraction (left). Average protein intensities of extracellular matrix proteoglycans in the membrane fraction separated by age and cognitive status (middle; $n = 6$ young, $n = 5$ middle-aged unimpaired, $n = 7$ middle-aged impaired mice). Aging mice exhibited higher proteoglycan abundances (n-Way ANOVA; two-sided; $p = 0$), and

aging-impaired mice showed higher proteoglycan abundances compared to aging-unimpaired mice ($p = 0.021$; post-hoc Tukey-Kramer). Right: heat plot of extracellular matrix proteoglycan abundances in the membrane fraction and bar plots of their relationship with cognitive status and age (*$p < 0.05$; linear probability model). **d** PCA plot of synapse proteins from the membrane fraction (left). Average protein intensities of synapse proteins in the membrane fraction separated by age and cognitive status (n-Way ANOVA; two-sided; $p = 0.62$; $n = 6$ young, 5 middle-aged unimpaired, 7 middle-aged impaired mice). Right: heat plot of glutamate receptor abundances and bar plots of their relationship with cognitive status and age (* $p < 0.05$; linear probability model). In all boxplots, boxes represent interquartile ranges (IQR; 25–75 percentiles), middle lines represent medians, and whiskers extend $\pm 1.5*IQR$. Source data are provided in the file Source Data - Fig. 6 and all statistics are provided in Supplementary Data 6. * represents $p < 0.05$, ** represents $p < 0.01$, *** represents $p < 0.001$ for each statistical test.

structure to support synaptic plasticity in young-adult brains[7,25], and argue that a loss of microglial regulation of the ECM during normative aging may contribute to excess ECM accumulation patterns we observe. An important future direction is to causally test how microglia modify ECM-synapse interactions in genetic mouse models with conditional manipulations to key microglial ECM-sensing proteins and ECM-degradative enzymes[70].

Our data also indicated that several features of microglial aging may impact cognition independently of microglial interactions with the ECM. For example, we found that the abundance of complement proteins C1qA, C1qB, and C1qC was elevated with advanced age, consistent with previous work[71]. Via proteomic mapping of cognitive aging phenotypes, we also found that late-middle-aged mice with better cognitive performance had lower levels of complement proteins. Complement proteins could be influencing cognition via tagging synapses for phagocytic removal or via complement-ECM interactions that shape accessibility of synapses for pruning[72]. However, our proteomic analysis indicates that complement proteins do not strongly align with ECM- and synapse protein proteomic profiles during healthy aging. Moreover, recent findings indicate that C1q interacts with neuronal ribonucleoprotein complexes in an age-dependent manner, perturbing neuronal protein synthesis in a way that impacts cognition[73]. Together, these observations point to both ECM-dependent and ECM-independent molecular mechanisms by which microglia and immune molecules can potentially shape neural circuit function and cognitive resilience during aging.

This study lays a foundation for future research on glial-matrix biology in the context of cognitive aging. One of the more important elements of our study design was the inclusion of two unique behavioral strategies that enabled us to link distinct cellular- and molecular-level observations about midbrain ECM and synapses to the cognitive status of individual young and aging mice. For example, by including sedentary control mice in experiments where young-adult and middle-aged mice underwent reward-based behavioral training, we were able to detect behavior-induced hyaluronan matrix remodeling characterized by less ECM deposition around dopamine neurons that was aligned with synaptic phenotypes associated with better reward-based memory. This remodeling may reflect the engagement of somato-dendritic dopamine release within the VTA in mice engaged in this task, as D1/D5 receptor activation has been linked to downstream protease release and ECM degradation[74]. More broadly, this observation demonstrates that ECM structure in the aging brain is not passively shaped by chronological age alone, but rather actively shaped by behavioral experience. This opens up exciting possibilities for non-invasive interventions (e.g., cognitive training or environmental enrichment)[75,76] to modulate ECM states in support of cognitive function during aging.

The finding that protein expression patterns in tissue proteomes of late-middle-aged mice segregate solely based on unsupervised

classification of their performance on canonical, high-throughput behavioral phenotyping tasks provides a strategy for linking cognitive heterogeneity in aged mice to underlying molecular states. This approach enabled us to identify the hyalectans as a specific family of ECM proteins strongly aligned with cognition in late-middle-aged mice. This behavioral classification strategy parallels work in the rat hippocampus linking histological and electrophysiological signatures of excitatory/inhibitory imbalance to poor cognitive aging[77]. More broadly, our work provides a scalable and unbiased experimental framework for appropriately sampling cognitive variability in aging mice in support of downstream experiments aimed at mechanistically dissecting cognitive resilience vs. decline. Importantly, while our analysis presents one strategy for behavioral classification of aging mice, different classification strategies may highlight distinct aspects of behavioral heterogeneity across the lifespan. It will be important for the field to apply alternative approaches to evaluate how different behavioral analysis strategies influence the relationships between behavior and distinct molecular states.

While this work establishes pipelines for aligning ECM composition with cognitive performance, the immense complexity of the ECM leaves much to be uncovered. For example, our regional proteomic mapping experiments revealed aging-related changes in numerous perivascular ECM molecules (e.g., collagens and laminins) that have been linked to microglial activation patterns and remodeling of the neuronal ECM[78]. Given the important links between neurovascular health and cognitive function during aging[79,80], it will be critical for future studies to focus on aging-related changes in the basement membrane-associated ECM in the context of cognitive aging. Moreover, our current methods provide limited insight into ECM assembly and spatial architecture beyond colocalization patterns. While we did not detect major shifts in solubility of most midbrain ECM proteins, we did observe aging-related differences in hyaluronan fragment size and filament length (Fig. S4). Hyaluronan size has been shown to influence membrane excitability, diffusion properties in the extracellular space, and receptor signaling[26,81,82]. It will be important for future studies to understand how these subtler changes in ECM composition influence such physiological processes in the healthy aged brain. In this regard, there is evidence that unique post-translational modifications on ECM proteoglycans (i.e., hydroxylation, sulfation, and glycosylation) can alter aspects of ECM physiology and its regulation of neuronal function during aging[10,11]. While our primary findings are based on relative comparisons within consistent experimental and analytical frameworks, our proteomic database searches did not include assessment of these ECM-relevant post-translational modifications, meaning that we likely underestimated matrisome protein abundance[83–85] and cannot provide insights into more subtle changes in matrisome composition. A key future direction will be to implement proteomic pipelines and enrichment strategies that incorporate ECM-relevant post-

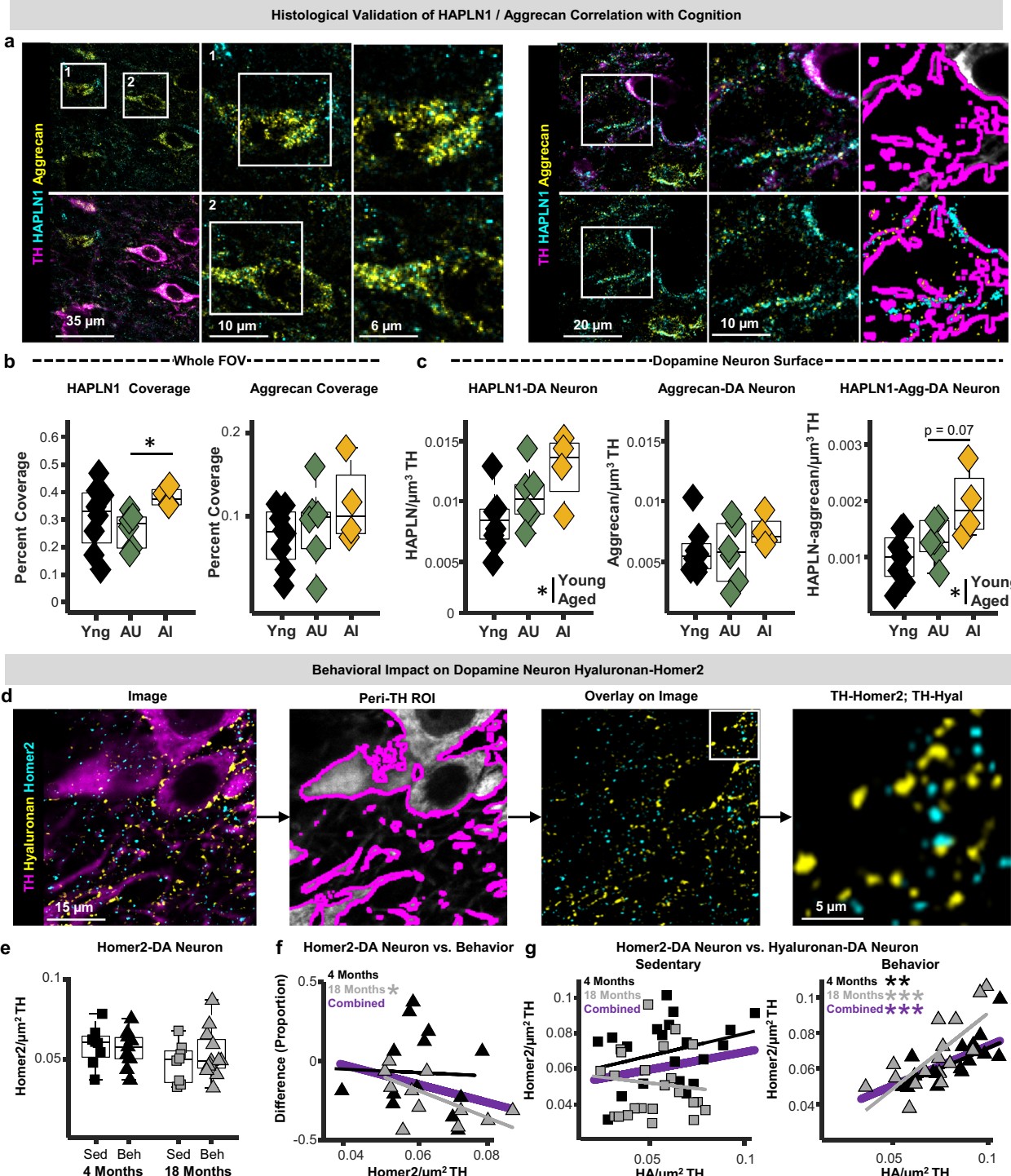

**Fig. (a)** Histological Validation of HAPLN1 / Aggrecan Correlation with Cognition

**b** Whole FOV
**c** Dopamine Neuron Surface
**d** Behavioral Impact on Dopamine Neuron Hyaluronan-Homer2

translational modifications to capture the full complexity of the aging brain matrisome and its relationship with cognition.

Furthermore, there remains a pressing need for more precise and physiologically relevant ECM manipulation strategies in neuroscience. Pharmacological approaches using compounds like 4-methylumbelliferone, a hyaluronan synthesis inhibitor, show inconsistent efficacy in the brain, require long treatment timelines, and provide limited temporal and spatial control[86,87]. Enzymatic ECM-degradation strategies have also been widely implemented in brain research; however, these treatments require invasive intracranial surgeries that induce mechanical injuries and the associated inflammatory and glial scarring responses[88,89]. This presents a major confound when examining

normative aging phenotypes of neuron-extrinsic factors like the ECM and microglia[90,91]. These considerations underscore the need for new tools to manipulate specific ECM targets through viral approaches using retro-orbital delivery[92] or similar minimally invasive methods.

Finally, while this study identifies numerous aging-related ECM and microglial phenotypes associated with cognitive impairment, brain aging is not a passive process and there are numerous examples of adaptive aging-related neurobiological changes[50]. One potential example from the present study was that, while VTA microglial aging phenotypes were associated with worse behavioral performance, aging mice with more NAc microglia exhibited better goal-directed behavior. Hence, while some aspects of microglial aging represent

**Fig. 7 | Histological analysis of extracellular matrix deposition on dopamine neurons in behaviorally characterized aging mice. a** Left: photomicrographs of immunohistochemically labeled hyaluronan and proteoglycan link protein 1 (HAPLN1), aggrecan, and tyrosine hydroxylase (TH) from the ventral tegmental area (VTA) of a late-middle-aged mouse. Numbered images in middle panel correspond to numbered fields of view within white squares in the left panels. Right panels depict the fields of view delineated by white squares in middle panel. Right: photomicrographs depicting HAPLN1 and aggrecan deposition on dopamine neuron surfaces, and the peri-dopamine neuron region of interest used to estimate protein abundances at neuronal membranes. Experiments were performed once on a single cohort of mice. **b** Boxplots depicting field of view coverage of HAPLN1 and aggrecan in the VTA of young-adult (black; $n = 8$ mice), aging-unimpaired (AU; green; $n = 6$ mice), and aging-impaired (AI; yellow; $n = 4$ mice) mice (n-Way ANOVA; post-hoc Tukey-Kramer; HAPLN: $p = 0.022$; Aggrecan: $p = 0.25$). **c** Boxplots depicting HAPLN1-DA neuron, aggrecan-DA neuron, and HAPLN1-aggrecan-DA neuron puncta densities in young-adult ($n = 8$ mice), aging-unimpaired (AU; $n = 6$ mice), and aging-impaired (AI; $n = 4$ mice) mice (n-Way ANOVA; two-sided; HAPLN1-DA:

$p = 0.029$; Aggrecan-DA: $p = 0.62$; HAPLN-Aggrecan-DA: $p = 0.031$). **d** Schematic of the analysis strategy used to examine hyaluronan and Homer2 on dopamine neurons (Homer-DA neuron). Experiments were performed once on a single cohort of mice. **e** Boxplots depicting homer-DA neuron puncta densities in young-adult (black) and late-middle-aged (gray) sedentary (squares) and behavior-trained (triangles) mice (n-way ANOVA; two-sided; Behavior: $p = 0.65$; $n = 8$ young-sedentary; 12 young-behavior; 8 middle-aged-sedentary; 12 middle-aged-behavior mice). **f** Relationship between homer-DA neuron puncta densities and performance on foraging task (robust regression; two-sided; Young: $p = 0.91$; Middle-aged: $p = 0.039$). **g** Scatter plots of hyaluronan-DA neuron puncta densities plotted against Homer2-DA neuron puncta densities in sedentary and behavior-trained mice (robust regression; two-sided; Young Sedentary: $p = 0.34$; Middle-aged Sedentary: $p = 0.14$; Young Behavior: $p = 0.0036$; Middle-aged Behavior: $p = 0.0068$). In all boxplots, boxes represent interquartile ranges (IQR; 25–75 percentiles), middle lines represent medians, and whiskers extend +/− 1.5*IQR. Source data are provided in the file Source Data - Fig. 7 and all statistics are provided in Supplementary Data 6. * represents $p < 0.05$.

vulnerabilities to circuit function, others may arise as adaptive responses that maintain neuronal network activity. As novel technologies emerge that allow for in vivo monitoring of specific ECM components, microglia-ECM interactions, and ECM-synapse interactions, such a framework (recognizing both detrimental and adaptive/beneficial aging-induced changes) will be critical in linking regional specializations in microglia-ECM-synapse dynamics with cognitive aging outcomes.

## Methods
All experimental protocols in this study adhered to protocols approved by the Animal Care and Use Committee and UCLA and were performed under protocol number: ARC-2018-103.

### Mice
This study uses C57Bl6 WT mice (*mus musculus*), CX3CR1[EGFP/+] and CX3CR1[EGFP/EGFP] mice, and Csf1r[+/+] and Csf1r[ΔFIRE/ΔFIRE] mice[46]. C57Bl6 WT mice used for behavioral experiments and histochemical experiments were purchased from the National Institutes on Aging (NIA) colony (Bethesda, Maryland), and the mice used for regional proteomic mapping (Figs. 1, 2) were purchased from Jackson Laboratory (Bar Harbor, ME; stock #000664). CX3CR1[EGFP/EGFP] breeders on a C57Bl6 background were originally purchased from the Jackson Laboratory (stock #005582) and crossed with C57Bl6 WT mice to obtain heterozygous (CX3CR1[EGFP/+]) mice and CX3CR1[EGFP/EGFP] mice to obtain homozygous mice (CX3CR1[EGFP/EGFP]). In these mice, EGFP is knocked into the fractalkine receptor (CX3CR1) locus, which is a receptor expressed specifically by most myeloid-lineage cells, including microglia[93]. Previous work has demonstrated that EGFP expression in these mice is specific to microglial cells in the basal ganglia[18]. Csf1r[ΔFIRE/ΔFIRE] breeders on a B6CBAF1/J background were originally purchased from Jackson Laboratory (stock # 032783) and crossed with C57BL/6 mice after which their offspring were interbred. These mice carry CRISPR/Cas9-generated deletion of the fms-intronic regulatory element (FIRE) of the Csf1r gene[46].

For immunohistochemical and histochemical experiments in sedentary WT mice, up to 12 young-adult (4 months; 6 male, 6 female) and 12 late-middle-aged mice (18 months; 6 male, 6 female) were used. Histochemical quantifications of synapse and ECM abundance in Cx3Cr1-deficient and knockout mice utilized 4 young-adult Cx3Cr1[EGFP/EGFP] (3-4 months; 2 male, 2 female) and 4 middle-aged Cx3Cr1[EGFP/EGFP] mice (12–15 months; 2 male, 2 female). Histological examinations comparing microglia-deficient mice with controls utilized 3 young-adult Csf1r[ΔFIRE/ΔFIRE] mice (3-4 months, 2 females, 1 male) and 3 young-adult Csf1r[+/+] mice (3-4 months, 1 female, 2 males). Quantitative proteomic experiments comparing ECM enrichment protocols utilized 8 young-adult WT mice (3–4 months). Midbrain

tissue from 4 mice (2 male, 2 female) underwent chaotropic extraction and digestion protocol and tissue from 4 mice (2 male, 2 female) underwent the tissue fractionation protocol. Quantitative proteomic experiments of the midbrain and striatum of non-behaviorally characterized aging mice included 4 young-adult (3–4 months; 2 male, 2 female) and 4 aged (22–24 months; 2 male, 2 female) WT mice. Quantitative proteomic experiments of the midbrain of behaviorally-characterized mice included 6 young-adult mice (4 months; 3 male, 3 female), 5 middle-aged unimpaired (18 months; 2 male, 3 female), and 7 middle-aged impaired mice (18 months; 4 male, 3 female). Histological experiments of behaviorally characterized mice included 8 young-adult mice (4 months; 4 male, 4 female); 6 middle-aged unimpaired (18 months; 5 male, 1 female), four middle-aged impaired mice (18 months; 0 male, 4 female). Foraging-based behavioral testing and histological examinations utilized 18 young-adult (4 months; 9 male, 9 female) and 18 late-middle-aged (18 months; 9 male, 9 female) WT mice. All mice within a given experiment were housed in the same vivarium with a normal light/dark cycle and were provided *ad libitum* access to food and water.

### Transcardial perfusion, immunohistochemistry, and histochemistry
Mice were euthanized by isofluorane overdose in a covered beaker and perfused transcardially with 1 M phosphate buffered saline (PBS; pH 7.4) followed by ice-cold 4% paraformaldehyde (PFA) in 1 M PBS. All perfusions were performed between 8:00 am and 12:00 pm to minimize the contribution that circadian changes may have on ECM, synapse, and microglial properties[94]. Brains were extracted immediately following perfusions and were allowed to post-fix for ~4 h in 4% PFA and then stored in 1 M PBS with 0.1% sodium azide until tissue sectioning.

Coronal brain sections were prepared using a vibratome in chilled 1 M PBS solution and stored in 1 M PBS with 0.1% sodium azide until histochemical labeling. Sections containing the ventral tegmental area (VTA) and nucleus accumbens (NAc) were selected using well-defined anatomical landmarks at 3 distinct anterior-posterior locations per mouse. Free-floating brain sections were briefly rinsed in 1 M PBS (5 min) and then permeabilized and blocked in a solution of 1% bovine serum albumin (BSA) and 0.1% saponin for 1 h. For hyaluronan labeling, sections were then incubated with a biotinylated HABP lectin (1:250; Sigma-Aldrich cat: 385911) in the 1% BSA and 0.1% saponin block solution overnight at 4 °C with mild agitation. Sections were washed in 1 M PBS (4 × 10 min) and then incubated with streptavidin-conjugated AlexaFluor-647 in the 1% BSA and 0.1% saponin block solution for 2 h. The tissue underwent another 4 × 10 min wash in 1 M PBS, and then was blocked again in 5% normal donkey serum with no additional permeabilization agent for 1 h and was then incubated overnight with mild

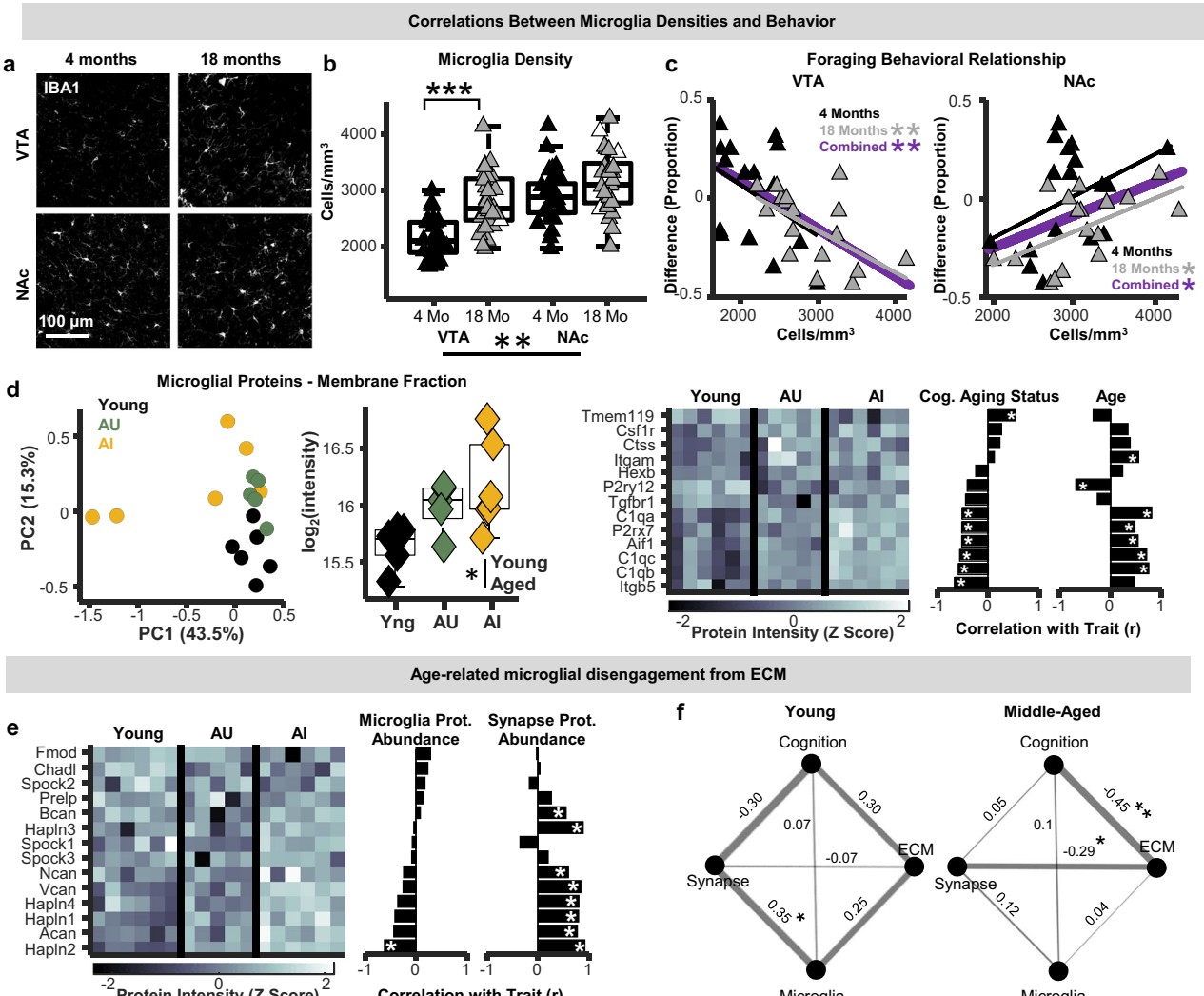

**Fig. 8 | Microglia-behavior relationships in aging mice. a** IBA1-positive microglia in the ventral tegmental area (VTA) and nucleus accumbens (NAc) of young-adult (4 months) and late-middle-aged (18 months) mice. **b** Boxplots of VTA and NAc microglia densities in young-adult (black; n = 28 mice) and late-middle-aged (gray; n = 29 mice) mice (n-way ANOVA; post-hoc Tukey-Kramer; two-sided; Age: p = 0; Region: p = 0; VTA post-hoc: p = 0.004). **c** Relationship between VTA and NAc microglia densities and performance on foraging task (robust regression; two-sided; Young VTA: p = 0.12; p = 0.004; Young NAc: p = 0.13; Middle-aged NAc: p = 0.038). **d** PCA plot of select microglia-enriched proteins from the membrane fraction (left). Average protein intensities of microglia-enriched proteins separated by age and cognitive status (middle; n-Way ANOVA; two-sided; *p* = 0.014; n = 6 young-adult, *n* = 5 middle-aged-unimpaired, n = 7 middle-aged-impaired mice). Heat plot of microglia-enriched protein abundances and bar plots of their

relationship with cognitive status and age (*p < 0.05; linear probability model). **e** Heat plot of ECM proteoglycan abundances from the membrane fraction and bar plots of their relationship with total microglia-enriched protein and synapse receptor abundances (*p < 0.05; linear probability model). **f** Correlation network plots of relationships between cognition and ECM, synapse, and microglia abundances from both behavioral experiments in this study shown separately for young-aged and middle-aged mice. Numbers are r values and the width of connections between variables scales with the strength of their correlation (linear regression). In all boxplots, boxes represent interquartile ranges (IQR; 25–75 percentiles), middle lines represent medians, and whiskers extend +/− 1.5*IQR. Source data are provided in the file Source Data - Fig. 8 and all statistics are provided in Supplementary Data 6. * represents p < 0.05, ** represents p < 0.01, *** represents p < 0.001.

agitation in a solution containing different combinations of goat anti-GFP (1:1000; Frontier Institute cat: GFP-go-Af1480), rabbit-anti-IBA1 (1:500; Wako cat: 019-19741), chicken anti-TH (1:500; Aves cat: TYH), rabbit anti-Homer2 (1:2000; Synaptic Systems cat: 160 003), mouse anti-Vglut1 (1:2000; DSBH cat: N28/9), biotinylated WFA (1:100; Vector Labs, B-1355), rabbit anti-aggrecan (1:200; Sigma Aldrich AB1031), goat-anti HAPLN (1:200; R&D Systems cat: AF2608) and the biotinylated HABP lectin in 5% NDS solution. Sections were again washed in 1 M PBS (4 × 10 min). Prior to secondary antibody incubation, all sections were treated with TrueBlack lipofuscin autofluorescence quencher (5%; Biotium cat: 23007) for 90 s followed by a 3 × 5 min rinse in 1 M PBS. Sections were then incubated in a secondary antibody solution containing combinations of chicken AlexaFluor-405 or chicken

AlexaFluor-594, goat AlexaFluor-488 or goat AlexaFluor-647, mouse AlexaFluor-647, rabbit AlexaFluor-488 or rabbit AlexaFluor594, and streptavidin-conjugated AlexaFluor-647 in 5% NDS for 2 h at room temperature with mild agitation. Experiments in which hyaluronan labeling was not performed were done in 1% BSA and 0.1% saponin throughout. The sections were again washed in 1 M PBS (4 × 10 min), coverslipped (#1.5 thickness), and mounted using Aqua-Poly/Mount (Polysciences cat: 18606).

### Image acquisition and analysis

Fixed-tissue was imaged using a Zeiss LSM-700 confocal microscope using a 63x objective at a z interval of 0.3 μm or a Leica STELLARIS 5 confocal microscope using 20x (z = 1.5) and 63x objectives (z = 0.3).

For quantification of hyaluronan and WFA fibril deposition patterns, 63x images were imported into Fiji image analysis software and underwent a background subtraction (rolling-ball radius = 15) and despeckling. The processed images were then binarized using the 'Max-Entropy' setting and the proportion of a FOV covered by hyaluronan or WFA was calculated. For the size-based analysis of WFA (Fig. S3), WFA puncta from thresholded images were filtered into two distinct groups based on pixel size: a putative interstitial WFA group (<150 pixels), or a putative perineuronal/perivascular WFA accumulation group (>150 pixels). Field-of-view coverage measures were then calculated for each group independently. Hyaluronan fibril densities and sizes were assessed using the 3D Object Counter plugin in Fiji using a minimum pixel cutoff of 10. Hyaluronan distribution regularity was quantified in the same images using the Fractal Dimension plugin in Fiji. HAPLN1 and aggrecan FOV coverage was calculated by importing images into Fiji image analysis software, applying a gamma correction and then a background subtraction (rolling-ball radius = 15). The processed images were then binarized using the 'MaxEntropy' setting and the proportion of a FOV covered by HAPLN1 or aggrecan was calculated. Densities of Homer2, VGlut1, putative Vglut1-Homer2 colocalization, putative hyaluronan-Homer2 interactions, and hyaluronan, HAPLN1, aggrecan, and Homer2 near TH-positive dopamine neurons were examined in 3-dimensions using custom-written MATLAB (Natick, MA) scripts. This custom code uses binarized images of neurons, microglia, and ECM/synapses and creates a peri-neuron and peri-microglia ROI by dilating the signal from each respective channel by 4 pixels and then subtracting the original thresholded signal from this dilated image. The resulting ROI is then used to index ECM molecules that are within 4 pixels of the cell surface. This code is available on Code Ocean under the capsule titled *ECM and microglial regulation of cognitive aging*; Capsule ID: f16ff2af-17b9-4b42-bd78-f7d79a08a10a. Densities of ECM or synaptic components were normalized by the total surface area of TH signal within the FOV.

Microglia densities were calculated by manually counting microglial cell bodies within a z-stack using the Cell Counter plugin and then normalizing these counts by the volume of the image. Microglial morphological complexity was examined with 2-dimensional Sholl analysis using the SNT and Sholl Analysis plugin in Fiji[95]. Images containing IBA1 or GFP signal were first maximum intensity projected in the z dimension. A mark was placed in the center of each microglial soma and concentric radii were drawn from that point. The initial radius size was 7.5 μm and the step size of each radius after was 2.5 μm. The standard output values given by this plugin were used for analysis. Sholl intersection plots were made using MATLAB code. Manual reconstructions of microglia and the hyaluronan matrix were done by importing high-magnification (63x) confocal z-stack images into Imaris (Bitplane; Belfast, UK) software for reconstruction using the surfaces module. For each channel, a threshold that most accurately represented the signal was manually set and surfaces smaller than 1 μm³ were filtered out. Hyaluronan-microglia contacts were captured by filtering the hyaluronan surface using the GFP fluorescence histograms. The number of GFP-filtered hyaluronan aggregates was normalized by the total GFP within the FOV.

## Quantitative proteomics: sample preparation

Young and aged WT mice were euthanized by isofluorane overdose in a closed chamber and perfused with 1 M PBS. Brains were extracted and the midbrain and striatum were dissected using a scalpel, minced, triturated, washed in 1 M PBS, and stored at −80 °C until further processing. For solubility-based fractionation experiments, the tissue was processed using protocols from Naba et al. 2015[32] that use a compartment protein extraction kit (Millipore cat: 2145). Triturated brain tissue was thawed on ice for ~30 min before being homogenized in 2 ml/g of Buffer C (HEPES (pH 7.9), MgCl2, KCl, EDTA, Sucrose, Glycerol, Sodium OrthoVanadate) with protease inhibitors. This mixture

was rotated on ice for 20 min and then spun at 20,000xg at 4 °C for 20 min. The supernatant was removed and stored at −80 °C until proteomic analysis. This fraction was considered the cytoplasmic fraction. The pellet was washed in 4 ml/g of the same buffer (HEPES (pH 7.9), MgCl2, KCl, EDTA, Sucrose, Glycerol, Sodium OrthoVanadate) with protease inhibitors, rotated for 5 min at 4 °C, and spun at 20,000 x g at 4 °C for 20 min. The supernatant was removed and discarded. The pellet was then incubated in 1 ml/g of Buffer N (HEPES (pH 7.9), MgCl2, NAcl, EDTA, Glycerol, Sodium OrthoVanadate), rotated at 4 °C for 20 min, spun at 20,000 × g for 20 min, and the supernatant was removed and stored at −80 °C until proteomic analysis. This fraction was considered the nuclear fraction. The pellet was then washed in 1 ml/g of Buffer M (HEPES (pH 7.9), MgCl2, KCl, EDTA, Sucrose, Glycerol, Sodium deoxycholate, NP-40, Sodium OrthoVanadate), rotated at 4 °C for 20 min, spun at 20,000 × g for 20 min, and the supernatant was removed and stored at −80 °C until proteomic analysis. This fraction was considered the membrane fraction. The pellet was washed in 0.5 ml/g of buffer CS (Pipes (pH6.8), MgCl2, NAcl, EDTA, Sucrose, SDS, Sodium OrthoVanadate), rotated at 4 °C for 20 min, spun at 20,000 x g for 20 min, and the supernatant was removed and stored at −80 °C until proteomic analysis. This fraction was considered the cytoskeletal fraction. The final remaining insoluble pellet was also stored at −80 °C for proteomic analysis. This fraction was considered the insoluble fraction.

Chaotropic extraction and digestion experiments followed protocols from McCabe et al. 2023[33], with the following modifications: Frozen samples were milled into powder in liquid nitrogen using a mortar and pestle. Approximately 5 mg of tissue was resuspended in a fresh prepared high-salt buffer (50 mM Tris-HCL, 3 M NaCl, 25 mM EDTA, 0.25% w/v CHAPS, pH 7.5) containing 1x protease inhibitor (Halt Protease Inhibitor, Thermo Scientific) at a concentration of 10 mg/ml. The samples were vortexed for 5 min and rotated at 4 °C for 1 h. The samples were then centrifuged for 20 min at 18,000 × g at 4 °C. The supernatants were discarded, and the pellets were resuspended in a high-salt buffer, vortexed for 5 min, and rotated again at 4 °C for 1 h. The samples were then recentrifuged for 15 min at 18,000 × g at 4 °C and the final pellets were used for proteomic analysis.

50–100 ug protein was processed for mass-spectrometry analysis. Samples were resuspended in an equal volume of 100 mM Tris-Cl (pH 8) and 8 M urea. Proteins were then reduced with TCEP, alkylated using IAA, and cleaned using SP3 beads. Proteolytic digestion was performed overnight using lysC enzymes and trypsin. Peptides were subsequently cleaned using the SP3 protocol and eluted in 2% DMSO. Following elution, samples were dried in a speed vacuum and the resulting peptides were resuspended in 5% formic acid prior to liquid chromatography−tandem mass spectrometry (LC-MS/MS) analysis.

## LC-MS/MS parameters and database search

For quantitative proteomic experiments comparing ECM enrichment strategies (Figure S1; Supplementary Data 1) and experiments comparing midbrain and striatum proteomes of non-behaviorally characterized aging mice (Fig. 1; Supplementary Data 2), LC-MS/MS was carried out as detailed in ref. 96. Approximately 200–500 ng peptide amounts were subjected to LC-MS/MS analysis. Briefly, peptide separation was carried out using reversed-phase chromatography on a 75 μm inner diameter fritted fused silica capillary column, which was packed in-house to a length of 25 cm with 1.9 μm ReproSil-Pur C18-AQ beads (120 Å pore size). An increasing gradient of acetonitrile was delivered using a Dionex Ultimate 3000 nano-LC system (Thermo Scientific) at a constant flow rate of 200 nL/min. Tandem mass spectra (MS/MS) were acquired in data-dependent acquisition (DDA) mode on an Orbitrap Fusion Lumos Tribrid mass spectrometer (Thermo Fisher Scientific). Full MS1 scans were acquired at a resolution of 120,000, followed by MS2 scans at a resolution of 15,000. The raw LC-MS/MS data were analyzed using the MaxQuant computational platform[97]. The

Andromeda search engine, integrated within MaxQuant, was employed for peptide identification against the UniProt reference proteome for *Mus musculus* (UP000000589). Search parameters allowed a maximum of two missed tryptic cleavages and included cysteine carbamidomethylation as a fixed modification and N-termination acetylation and methionine oxidation as variable modifications. A false discovery rate (FDR) threshold of 1% was applied at both peptide and protein levels. Label-free quantification (LFQ) was enabled, with a minimum LFQ ratio count set to one. Precursor ion and fragment ion mass tolerances were set at 20 and 4.5 ppm, respectively. The resulting MaxQuant output files were subsequently used for statistical analysis to identify differentially enriched proteins. Raw data files have been deposited in the MassIVE proteomics repository under accession number MSV000096508 and the ProteomeXchange repository under the accession number PXD070000. MS metrics and metadata information can be found in Supplementary Data 3.

For quantitative proteomic experiments of the midbrain of behaviorally-characterized mice (Fig. 6; Supplementary Data 4), the cytosolic (CS), membrane (M), and insoluble (Insol) fractions were subjected to sequential reduction and alkylation steps using 5 mM tris(2-carboxyethyl)phosphine (TCEP) and 10 mM iodoacetamide, respectively. Following this, the protein aggregation capture (PAC) protocol, as described by Batth et al. (2019)[98] was employed to purify the reduced and alkylated proteins. Proteins were then enzymatically digested overnight at 37 °C using Lys-C and trypsin proteases. The resulting peptide mixtures were dried completely and prepared for subsequent LC-MS/MS analysis. Dried tryptic peptides were resuspended in 5% formic acid and subjected to liquid chromatography-tandem mass spectrometry (LC-MS/MS) analysis using a Vanquish Neo ultra-high-performance liquid chromatography (UHPLC) system coupled to an Orbitrap Astral mass spectrometer (Thermo Fisher Scientific, Bremen, Germany). Briefly, peptide samples were introduced into a PepSep C18 reverse-phase analytical column (150 mm × 150 μm, 1.7 μm particle size), maintained at 59 °C, and separated using a trap-and-elute workflow. The UHPLC system employed mobile phase A consisting of water with 0.1% formic acid and mobile phase B consisting of acetonitrile with 0.1% formic acid. Peptide separation was achieved using a 15-min chromatographic gradient with the following composition: 5% B from 0–1 min at a flow rate of 2.45 μL/min; a linear increase from 5% to 15% B from 1–5 min at 1.75 μL/min; 15% to 25% B from 5–12.6 min at 1.75 μL/min; 25% to 38% B from 12.6–13.6 min at 1.75 μL/min; and a rapid gradient from 38% to 80% B between 13.6–13.7 min at 2.45 μL/min, followed by a hold at 80% B until 15 min, maintaining the flow at 2.45 μL/min.

Mass spectrometric acquisition was carried out in data-independent acquisition (DIA) mode on the Orbitrap Astral instrument operating in positive electrospray ionization mode. MS1 survey scans were acquired across an m/z range of 380–980 with a resolution of 240,000, using a normalized AGC (automatic gain control) target of 500% and a maximum injection time of 3 ms. For DIA, sequential isolation windows of 4 m/z were used to comprehensively cover the 380–980 m/z range. Fragment ion (MS2) spectra were acquired at a resolution of 80,000, with a normalized higher-energy collisional dissociation (HCD) energy of 25%, an AGC target of 500%, and a maximum injection time of 7 ms. The raw Thermo.RAW files were analyzed using DIA-NN software, searching against an in silico predicted spectral library generated from the Mus musculus reference proteome (UniProt ID: UP000000589), as described by Demichev et al., (2020)[99]. The resulting DIA-NN outputs were processed using FragPipe Analyst to identify differentially expressed proteins, following the workflow outlined by Hsiao et al., (2024)[100]. All raw data files have been deposited in the MassIVE proteomics repository under accession number MSV000096508 and the ProteomeXchange repository under the identifier PXD070000. MS metrics and metadata information for these data can be found in Supplementary Data 5.

## Quantitative proteomics: analysis

Protein intensity values for all samples were batch corrected via quantile normalization and median centering. For regional mapping experiments of ECM protein abundance across subcellular/solubility fractions (Fig. 1c–e), age-related shifts in protein solubility were examined by calculating the fold-change of individual ECM protein abundance (log2-normalized protein intensities) that were detected across at least 2 distinct fractions (fold change relative to the more insoluble fraction), and comparing these fold-changes between young-adult and aged mice. For regional mapping experiments of total ECM protein abundance, and WGCNA analysis (see below), data for each identified protein was summed across solubility fractions for each mouse and log2 normalized for analysis to estimate total protein intensity. For proteomic mapping of cognitive phenotypes, log2-normalized protein intensities detected in membrane, cytoskeletal, and insoluble fractions were examined independently. This proteomic mapping was only performed on midbrain tissue. To be included in the analysis, a protein needed to be detected in 75% or more of the samples within a brain region. Fold changes were calculated as the ratio of the aged signal relative to the young signal. Statistical significance was assessed using unpaired t-tests with an alpha-level of 0.05. Annotations for different protein classes were downloaded from publicly available online databases. ECM protein annotations were downloaded from the Matrisome project database[101], innate immune system protein annotations from the InnateDB database[102], and synapse protein annotations from the SynGo database[103].

## Quantitative proteomics: weighted gene correlation network analysis (WGCNA)

To examine relationships between age, cognitive status, synaptic, matrisome, and innate immune proteins, proteomic data underwent unsupervised clustering using Weighted Gene Correlation Network Analysis (WGCNA) in R[34]. For the regional mapping experiment, processed log2(intensity) data from the midbrain and striatum were used for network construction. One striatal sample from an aged female mouse was identified as an outlier and removed from this analysis. WGCNA provides modules of covarying proteins that are agnostic to traits of the samples (i.e., age, region, sex). The distribution of synaptic, matrisome, and immune proteins across the protein modules was used to identify modules of interest for further analysis. Relationships between modules and external traits or protein abundance were examined using the module-trait relationship code provided in the WGCNA R package. Process enrichment analysis of the protein modules was conducted in Metascape[104] using default parameters. For the proteomic mapping of cognitive phenotypes experiment, only proteomic data from the midbrain was used for WGCNA analysis. Using age and cognitive status (see below) as traits, protein-trait relationships were derived using the module-trait relationship code provided in the WGCNA R package.

## Mining of published RNAseq data from WT and Cx3Cr1-deficient microglia

Publicly available bulk RNA-seq datasets of isolated microglia from Cx3cr1 knockout (Cx3cr1$^{-/-}$), Cx3Cr1 heterozygous (Cx3cr1$^{+/-}$), and WT mice were retrieved from Gyoneva et al., 2019[43]. Analysis was restricted to a curated list of matrisome genes, defined according to the MatrisomeDB[101], encompassing core ECM proteins and ECM regulators. Fold changes of aging-induced changes in matrisome genes were extracted from datasets comparing 2-month and 12-month-old Cx3Cr1-knockout and WT mice. Additionally, fold changes of matrisome gene expression between Cx3Cr1 heterozygous and Cx3Cr1 homozygous knockout (Cx3cr1$^{-/-}$) mice (2 months old) were examined. This targeted approach enabled the identification of Cx3cr1-dependent alterations in microglial ECM-associated gene expression.

## Foraging-based behavioral testing procedures

For the foraging-based behavioral experiment, behavioral testing was conducted in a large square open field (120 × 120 cm) with 35 cm high walls made from transparent Plexiglas. On one wall, there was a 20 × 20 cm opening where a start box equipped with a guillotine door was placed. This start location did not change across testing sessions. Mouse bedding was placed on the floor of the arena, and 2 intra-arena reference cues were placed at two fixed locations for the entire experiment. To also provide fixed distal landmarks to the mice, black corrugated plastic was placed on two adjacent walls in the arena, and white corrugated plastic was placed at the other two. Plexiglas dishes were filled with Sandtastik play sand (Sandtastik Products LLC, Port Colborne Ontario) and were used to plug 5TUL purified rodent tablets into (TestDiet; cat: 1811142; Quakertown, PA) to engage foraging behavior. Identical feeders without reward pellets were used as distractor feeders to test goal-directed memory. The feeders were placed at 1 of 4 locations within the arena. All feeders were sham baited with powdered reward pellets (5% by weight). Mouse behavior was recorded using a 1.3 megapixel, low-illumination overhead camera (ELP; Shenzhen, Guangdong, China) and Bioserve Viewer software (Behavioral Instruments; Hillsborough, NJ).

Young-adult (3–4 months) and late-middle-aged (16–18 months) mice were ordered in 3 cohorts of 20. Twelve mice from each cohort then underwent behavioral testing and the other 8 lived with the behavior-trained mice but did not undergo testing (sedentary controls). Thus, in total this experiment used 24 sedentary mice (12 young, 12 middle-aged; 6 male and 6 female in each group) and 36 behavior-trained mice (18 young, 18 middle-aged; 9 male and 9 female in each group). Sample sizes were determined based on variability in pilot data and from published data on a similar task from which this one was derived[105].

Behavior-trained mice were handled daily for 1 week prior to testing. During week 1 of testing (habituation phase) mice were placed within the start box for ~2–3 min prior to the guillotine door opening to allow entry into the arena. A single sand well with 4 reward pellets was placed at one of the reward locations and the mice were allowed 5 min to explore the arena and retrieve the reward. This procedure was repeated daily for 5 sessions for each mouse and the location of the rewarded feeder was changed daily and was counterbalanced within a day across mice. During weeks 2 and 3 (training phase), the mice again entered the arena via the start box and foraged for reward in a single sand well at one location in the arena for 5 min (encoding). Following a 30 min delay period, mice reentered the arena with the rewarded feeder in the same location and 3 unrewarded feeders in the other locations for 2 min (retrieval). Note that this is a win-stay strategy. Again, the location of the rewarded feeder was changed daily for each mouse. Each mouse underwent ten training sessions across 2 weeks of testing. In the final 2 weeks of testing (test phase), mice repeated the encoding procedures described above. Following either 2- or 24 h delays, mice underwent a probe test in which 4 unrewarded feeders were placed within the arena, 1 at the previously rewarded location. After 2 min the mice exited the arena through the guillotine door and the experimenter placed reward pellets in the previously rewarded location. The mice then reentered the arena and consumed the food reward to minimize extinction. Again, rewarded locations changed across sessions, and the 2- and 24 h delays were interleaved to eliminate practice effects between the two conditions.

## Foraging-based behavioral testing data analysis

Mouse behavioral tracking was collected using Bioserve Viewer software in 1-s timestamps and was used to determine the time of entry and exit to various zones of interest within the arena. These zones included the reward feeder locations, and a box that separated the arena's perimeter from the center. Data were exported from the Bioserve Viewer software and loaded into Matlab for behavioral analysis using custom-written scripts. From this data the following output measures were derived: average running speed, start-box exit latency, proportion of time on the arena's perimeter, number of feeder visits, proportion of feeder visits to the correct feeder, the relative proportion of time in correct and incorrect feeders, and the number of errors prior to correct feeder visits. Note that because there were 4 foraging locations, chance performance during probe trials is 25% for proportional measures and 1.5 for error measures. Probe trial data were used as the estimate of goal-directed memory function in these mice.

The temporal progression of the emergence of foraging behavior was tracked across sessions for each animal with a state-space modeling approach using Bernoulli observation models[106]. Mice were considered to have foraged if they consumed at least 1 of the 4 treats within the arena during the encoding sessions. Using this binary assessment (1 = consumed, 0 = non consumed), a learning curve, its 90% confidence bounds and the trial at which consistent foraging was observed were determined. This trial was defined as the point in which the lower bound of the 90% confidence interval exceeded chance (50%) and stayed above chance for the remainder of the experiment.

## Behavioral battery for proteomic mapping of cognitive phenotypes

For behavioral characterization experiments using spontaneous rodent behaviors, 12 young-adult (4 months; six male and six female) and 24 middle-aged (18 months; 12 male and 12 female) were used. First, all mice underwent an open field test to assess general locomotor activity and anxiety-like behavior using a square arena (60 × 60 cm with opaque walls 30 cm high). Mice were placed in the center of the arena and allowed to explore freely for 10 min. Their behavior was tracked with an overhead camera using Bioserve Viewer software, and the floor of the arena was digitally divided into central and peripheral zone to calculate the proportion of time spent in the center and periphery of the arena. The arena was cleaned with 70% ethanol and water between mice. Following open-field testing, all mice then underwent a NOR task in the same behavioral apparatus. Because testing was done in the same apparatus, the open field test also served as habituation for the NOR test. Thus, NOR testing began with familiarization of two identical objects (either two LEGO stacks or two PVC pipes of equal height and width, counterbalanced across subjects) that were placed symmetrically near the corners of the arena. Mice again were allowed to explore for 10 min. Using a separate group of aging mice, we confirmed that mice did not inherently prefer one object over the other by placing one of each object within the arena and allowing mice to explore for 10 min. After a retention interval of 2 h, NOR mice underwent the test phase of the task where one familiar object was replaced with a novel object. The position of the novel object (left/right) was counterbalanced across subjects to control for side preference. Again mice explored for 10 min. Object exploration was defined as the mouse directing its nose, within 2 cm from the object. Using this data, a discrimination index was calculated as follows: (time with novel – time with familiar) / (total time with both objects). Objects were cleaned with 70% ethanol and water between trials. Following NOR testing, mice underwent a T-maze spontaneous alternation test. This testing was done in a T-shaped maze made of white corrugated plastic, comprising a start arm and two goal arms. Each arm was 30 cm long, 10 cm wide, and 25 cm tall. During the first trial, mice were forced into either a right or a left decision using a removable barrier. This forced choice was counterbalanced between mice. During all subsequent trials, mice were allowed to freely choose between the two goal arms. After entering one of the goal arms, mice were returned to the start arm for the subsequent trial. The procedure was repeated for ten consecutive trials, with an inter-trial interval of ~30 s. An alternation was recorded when the mouse chose the goal arm opposite to the one it entered on the previous trial, and an alternation index was calculated as follows: number of alternations / total number of trials.

## Classification of middle-aged unimpaired and impaired mice

To behaviorally classify middle-aged mice as impaired or unimpaired, behavioral data from open field, NOR, and the T-maze were used for unsupervised hierarchical clustering analysis. For each mouse, a behavioral performance vector was created using the proportion of time on the arena perimeter from the open field test, the discrimination index from the NOR test, and the alternation index from the T-maze. All data was z-scored prior to constructing these vectors to ensure equal weighting across features. Hierarchical clustering was performed in MATLAB using the 'linkage' function with Ward's method (minimizing total within-cluster variance) and Euclidean distance as the similarity metric. A dendrogram was generated using the 'dendrogram' function to visualize hierarchical relationships between subjects. Clusters were identified by applying a threshold to the dendrogram (via cluster function with $k = 2$), resulting in two distinct clusters. Middle-aged mice that fell within the cluster with the majority of young mice (75%) were considered unimpaired and the middle-aged mice that fell in the other cluster were considered impaired. This data-driven behavioral classification was then used to examine group-level differences in proteomic and histological data.

## Statistical analysis and reproducibility

Statistical analyses were done using either MATLAB or R. Statistical significance of proteomic protein-intensity data was evaluated using unpaired t-tests. Immunohistochemical data were statistically analyzed using multi-way ANOVAs with Tukey's post-hoc tests where applicable, and foraging behavior data were assessed using repeated-measures ANOVAs and unpaired t-tests. Relationships between anatomical variables and behavioral variables, or between different anatomical variables, were assessed using robust regression analyses. Relationships between proteomic data and cognitive status were examined using point-serial correlation analysis (regression with one binary variable). To visualize relationships between all behavioral, anatomical, and proteomic variables created in both behavioral experiments, a correlation-based network analysis was performed in MATLAB. Here, a pairwise Pearson correlation matrix was computed across all behavioral measures using the 'corrcoef' function with pairwise deletion for missing data. Self-correlations on the diagonal ($r = 1$) were set to zero to exclude self-links in the resulting network. This network was then visualized as an undirected graph using the 'graph' function. For all experiments, a significance cutoff of $p < 0.05$ was used. Potential sex differences in ECM, microglia, and excitatory synapse status were evaluated in all cases (Supplementary Data 6).

All histological, and proteomic experiments were performed using biologically independent mice (per group for young and aged cohorts across behavioral conditions). For histology, three brain sections per mouse were processed and imaged using the same antibody set; data represent the average of technical replicates (sections) for each biological replicate (mouse). For proteomics, tissue from each mouse was fractionated into five samples, which were analyzed independently; results were either aggregated per mouse to represent one biological replicate or analyzed independently as biological replicates. All experiments were performed once on a single cohort and yielded consistent results across all samples.

## Reporting summary

Further information on research design is available in the Nature Portfolio Reporting Summary linked to this article.

## Data availability

Minimal datasets and source data are provided with this paper in Source Data file. The raw quantitative proteomics datasets generated in this study and the necessary metadata for replication and verification have been deposited in the MassIVE online database under accession code MSV000096508 (FTP download link: ftp://massive-ftp. ucsd.edu/v07/MSV000096508/) and the ProteomeXchange online database under the identifier PXD070000. Raw microscopy images are available upon request. Source data are provided with this paper.

## Code availability

Custom code used for analyses is available on CodeOcean (capsule number: 18-559067-2) and is accessible here: https://doi.org/10.24433/CO.0457638.v1.

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

## Acknowledgements

The authors would like to thank Keionna Newton and Fanny Etienne for providing tissue used in this study. Additionally, we thank Mia Donato and Claribel Charway for important contributions to mouse behavior

pipelines used in this study. This research was supported in part by the Intramural Research Program of the National Institutes of Health (NIH). The contributions of the NIH author are considered Works of the United States Government. The findings and conclusions presented in this paper are those of the authors and do not necessarily reflect the views of the NIH or the U.S. Department of Health and Human Services. This work was also supported by the American Federation on Aging Research grant number: MCKNIGHT21003 and NIH/NIA grant R01AG075909-01 awarded to L.M.D., by the Simons foundation Collaboration on Plasticity in the Aging Brain grant number: SFI-AN-NC-AB-Independence Postdoctoral-00007445 awarded to D.T.G., NIH grants NIH/NINDS R01NS109025, NIH/NICHD P50HD103557, NIH/NIDA R01DA059873 awarded to Y.Z.

## Author contributions

D.T.G. conceived the study, designed methodology, performed experiments, analyzed data, and wrote the manuscript. A.G. performed behavioral experiments and microglial analysis. Y.J.A. performed proteomic experiments and proteomic data analysis. V.P. performed proteomic experiments and proteomic data analysis. L.P. performed proteomic experiments comparing ECM enrichment strategies. Y.Z. performed proteomic experiments comparing ECM enrichment strategies and provided critical support in manuscript preparation. J.A.W. conceptualized proteomic experiments and analysis. R.A.M. helped conceptualize behavioral experiments and provided critical support in manuscript preparation. L.M.D. conceived the study, designed methodology, and wrote the manuscript.

## Competing interests

The authors declare no competing interests.
