## [Transparent Peer Review file · Nature Communications]

Midbrain extracellular matrix and microglia are associated with cognition in aging mice

Corresponding Author: Dr Daniel Gray

Version 0:

Reviewer comments:

Reviewer #1

(Remarks to the Author)

The noteworthy results:

Grey et al. used proteomics to reveal substantial regional differences in ECM composition between VTA and NAc and aging-induced ECM remodeling. Next, the authors relied on a combination of behavioral and histochemical analyses to present interesting positive correlations between the expression of hyaluronic acid, microglia and excitatory postsynapses in VTA, and the latter showed a negative correlation with learning. In a mouse model of microglia ablation, ECM (hyaluronic acid) expression was elevated, but no postsynaptic changes were detected. ECM was not changed in aged Cx3Cr1 heterozygous mice but was downregulated in Cx3Cr1 knockout mice. However, the expression of the postsynaptic marker was similarly reduced in both genotypes, so clear relationship between ECM and synapses emerged from analysis of these mutants.

Will the work be of significance to the field and related fields?

It is of interest and can be significantly improved.

Does the work support the conclusions and claims, or is additional evidence needed?

Indeed, additional analyses are needed.

Unfortunately, in the current version of the manuscript, the proteomics results seem to be insufficiently exploited and not integrated with the follow-up analyses. Proteomics revealed changes in numerous perivascular ECM molecules. However, there is no attempt to show vascular changes or discuss their impact despite the remodeling of perivascular ECM is known to be linked to glial activation and remodeling of neuronal ECM (e.g., DOI: 10.1016/j.matbio.2024.02.007). Although alterations in the expression of lecticans were revealed by proteomics, they were not studied by immunohistochemistry. Moreover, analysis was focused on excitatory postsynapses, while most prominently, lecticans are aggregating in WFA-positive lectican-enriched perineuronal nets, which are known to have critical behavioral functions in the studied regions (DOI: 10.1016/j.biopsych.2024.02.003). The study would also benefit from the analysis of presynaptic changes (e.g., VGLUT1) in excitatory synapses, which are related to microglia activity and perisynaptic ECM expression according to previous microglia depletion studies (e.g., see DOI: 10.3390/cells10081862 and refs herein).

This study used proteomics of synaptosomes to study age-dependent changes in excitatory synapses: DOI: 10.1074/mcp.M113.032086. Such measures (although done in VTA and NAc) would be more directly linked to changes in ECM of excitatory synapses as the main focus of the present work. It is worth acknowledging/discussing this approach.

Understanding microglia relationships with ECM and synapses requires a more substantial characterization of microglia in the studied regions. E.g., the size of microglial soma is an indicator of its activation, and this measure can be easily added to the analysis. Also, relevant information on the expression of microglial proteins can be extracted from the proteomic data. The relationship between microglia, the expression of C1q and ECM could be studied with higher resolution by measuring C1q expression as a function of ECM near or far from synapses.

Analysis of microglial mutants shows that the expression of hyaluronic acid is not proportional to the number of synapses. A more comprehensive analysis of ECM markers would possibly help resolve this issue.

Lectican degradation was shown to be driven through ADAMTS4/5 release downstream of D1/D5 receptor activation. There is somatodendritic dopamine release in VTA and VTA projections to NAc, so increased excitatory drive to VTA dopaminergic neurons may result in ECM remodeling in these regions. A discussion of the link from dopamine to ECM would be relevant, particularly if it may explain why “the accumulation of VTA hyaluronan observed in aging sedentary animals was no longer apparent in aging behavior-trained mice”.

Is the methodology sound? Does the work meet the expected standards in your field?
Mostly, yes, but a more solid analysis of ECM and microglial markers is expected.

Is there enough detail provided in the methods for the work to be reproduced?
Yes.

(Remarks on code availability)

Reviewer #2

(Remarks to the Author)

In this manuscript, Gray and colleagues explore the basal ganglia extracellular matrix (ECM) during aging. They identified regional-specific changes in ECM composition during aging with proteomic mapping approaches. Using histology and confocal imaging combined with a reward-based memory test, they found that deposition of HA (a key ECM component that acts as a substrate for many other ECM molecules to bind), specifically in the VTA, aligns with cognitive performance, synapses, and microglial phenotypes. They further showed that HA abundance is altered in mice lacking microglia (Csf1r KO) and in mice lacking Cx3cr1. The authors conclude that glia-ECM interactions influence synapse and cognitive status in aging. The strength of the study lies in the overall topic of investigation (the ECM is an understudied area of neuroscience), as well as in the comprehensive analysis of the ECM molecules in the aged brain. There were also many sophisticated and consistent correlation analyses aligning these various measures (i.e., HA, synapses, behavioral performance, and microglia) to each other in aging that are quite strong. However, the major concern is the lack of causal role with any one specific measure to another. Some evidence is provided for microglia as being the underlying effector that promotes ECM deposition, yet it is not strong enough to support the conclusions. A clearer mechanistic understanding of how the ECM negatively impacts cognitive function/synapses in aging and how microglia may contribute to this process is needed to give depth to the study.

1) On a correlational level, the data suggests that HA abundance negatively impacts cognition/synaptic refinement in the aged brain. Yet, the act of performing cognitive tests itself affects HA abundance and synapses, making it challenging to establish clear directionality. The current data do not provide direct causal evidence. A more direct approach would be to explore whether reducing HA has positive impacts on cognition during healthy aging, or whether greater HA limits “synaptic refinement” in brain aging.

2) There were several flaws with the microglial manipulation experiments. For example, the authors use constitutive knockouts of Csf1r and Cx3cr1 as their manipulations, which do not rule out the possibility that the observed effects arise from developmental compensations or deficits rather than aging-specific roles. Because both microglia and the ECM play important roles during circuit formation during development, it seems particularly important to delineate as these effects could simply be related to development. Experiments using conditional knockouts during adulthood/aging would be more informative.

3) Another important note is the lack of microglial mechanism. Microglial regulation of the ECM has been shown in several papers now, including a report using Cx3cr1 deficiency (PMID: 29017970). Yet, the mechanistic role microglia play in regulating the ECM during healthy aging is unclear from the data. The authors suggest that it may be due to reduced ECM engulfment, but there is a lack of evidence to support the claim. Without a mechanistic view or a cleaner manipulation of microglia in the aged brain, the conceptual novelty and depth of the manuscript are limited.

4) Furthermore, none of the microglia manipulations are linked to cognitive function / ECM deposition around synapses during brain aging. It remains unclear how these various measures align.

5) More information is needed on the regional-specific roles of microglia. The data show different associations of microglia with the ECM in the VTA vs the NAc, with the NAc showing prominent associations with age that are similar to the associations shown in the VTA throughout life. Yet, none of the microglial manipulations examined the NAc. It is also unclear whether aged microglia are functionally different in their regulation of the ECM in VTA vs NAc.

6) The Cx3cr1 hets do not recapitulate the WT aging data. It is also unclear from the data whether the young Cx3cr1 hets/ko are similar to the young WT, or whether this is a result of brain aging. It would be helpful to compare these data to WTs.

7) There are many references to “synaptic function” or “synapse status” yet the synaptic findings reported are quite minimal. No attempt to look at synaptic physiology or other correlates of synaptic function.

Other minor comments:

1) Throughout the paper, there is a distinguishment made between basal ganglia nuclei overall and then in response to

aging. This often makes it difficult to follow what is being compared. It would be helpful to clarify (in the figures particularly) which significant differences correspond to specific comparisons and the statistical tests used. It would also be helpful to include more statistical information in the figure legend for the individual panels. Along these lines, providing R values for the correlations and including the key for the lines would be helpful.

2) Fig 1e, the dotted purple line is not at 0 like in the other figures panels. This should be explained. More information is needed for how the fold change was calculated for the panels in Fig 1.

3) On a similar note, line 95 of page 5 indicates that Tenascin C is upregulated during aging, yet the graph shows a decrease (Fig 1e). This should be clarified.

(Remarks on code availability)

Reviewer #3

(Remarks to the Author)

In the manuscript entitled "Extracellular matrix remodeling during aging aligns with synapse microglia and cognitive status," Gray and colleagues performed a multi-modal analysis of the extracellular matrix of different brain regions in young and old mice to identify the possible molecular bases leading to the cognitive changes that occur during aging. The significance of the premise of this study is undeniable. The extent of the effort put into this study is also undeniable, as demonstrated by the vast amount of data (8 main multi-paneled figures, 9 extended multi-paneled data figures, and 8 multi-paneled supplemental figures) spanning different scales, from molecular characterization by proteomics to behavioral testing. However, significant flaws in the initial proteomic analysis and the interpretation of the changes in the ECM observed undermine the results derived from the analysis. In addition, the lack of clear connections between the different sections of the manuscript (from proteomics to HA – a non-protein component- patterns, from HA patterns to the involvement of the microglia, etc.) makes the manuscript quite difficult to read and digest.

My specific comments focus on aspects of the first part of the study pertaining to the proteomic experiments and the analysis of the ECM, my primary domains of expertise. I hope they help strengthen the study design and data interpretation.

> Major comments regarding the proteomic analysis:

1) Insufficient information is provided in the Material and Method to interpret, let alone attempt to reproduce, the proteomic experiments performed:

- The authors employed two experimental approaches to enrich ECM proteins. However, the authors state that only 4 animals were used per age group (2 males and 2 females; lines 457-458). Is this per experimental approach, or are only 1 male+1 female per age group used for each approach? This should be clarified. The very small cohort size is also of concern.
- What amount of protein (ug? mg?) was digested?
- A section describing the LC-MS/MS acquisition modality is missing. Specifically, what amount of peptide (ug? ng?) was separated via LC (what parameters were used for the LC separation), and how was the MS/MS acquisition performed (DIA? DDA?). I have tried looking for this information in the MassIVE repository created to disseminate the dataset but could not find it.
- A section describing the parameters used for the database search is also missing: The MassIVE repository indicates that no post-translational modifications were included during the database search. If true, this is a critical limitation of the experimental design that undoubtedly resulted in the underestimation of the number of proteins identified and all the quantitative analyses performed (extensive studies have been published on the topic that the authors could refer to).

2) Insufficient information is provided to assess the rigor of the quantitative analysis:

- Are all proteins identified included in the analysis, or did the authors apply thresholds to exclude low-probability IDs (what were the FDRs applied? At what level? PSM? Protein?).
- It is customary for a study reporting proteomic data to provide the details of the MS metrics collected for each sample, i.e., the non-normalized and normalized precursor ion intensity for each replicate, and obtain via pipeline in a supplementary table. Here, the authors only report the "log2-normalized intensity values" (note the typo in the word "intensity" in the supplementary table). The headers of the columns ("OF1", "OF2", etc.) should be spelled out.
- Since the authors have performed the proteomic analysis on ECM-enriched samples obtained using two different experimental approaches, it is surprising that no comparative analysis is presented.
- Unless I missed that, the tables are not cited in line with the text and no table legends are provided.
- While I appreciated that the authors deposited their raw MS files to MassIVE, the lack of meta-data (for example, key linking .raw files to sample type; LC_MS/MS acquisition parameters) prevents proper evaluation, re-analysis, and re-use. The authors should provide additional meta-data information. I suggest adopting the Sample and Data Relationship Format (SDRF), now broadly used in proteomics (PMID: 37875519).

> Minor comment regarding the proteomic analysis:

- I would encourage the authors to refrain from using the word "novel" to describe the ECM-optimized proteomic workflows published several years ago and already broadly adopted by the scientific community.
- Figure 1: could the authors add the gene symbols of the proteins found in statistically significant differential abundance in

the volcano plot panels (c)? It would simplify Figure 1 and allow the elimination of panels d- f.

> Major comments regarding the interpretation of the ECM phenotype:

- Since the proteomic analysis identified compositional differences across brain regions and ages, it would have been interesting to validate some of these differences (including those relevant to HA metabolism and functions) using orthogonal approaches, such as immuno-staining.
- The transition from the proteomic analysis to the focus on HA is unclear.
- Since the authors interpret their observation as resulting from “ECM remodeling”, it would have been interesting to stain for brain sections where HA patterns are shown to differ for total collagen content as well (using anti-collagen antibodies or Masson’s trichrome or Picrosirius red stains). This would be a more definitive proof of ECM remodeling.
- The authors indicate having summed the intensity values of each identified protein across all solubility fractions (lines 570 – 571) but do not provide a rationale for doing so. This is a missed opportunity since their study otherwise could have provided valuable insights not only on the compositional changes that occur in the brain ECM during aging but also changes in ECM protein solubility that may have occurred during aging, which could inform on the state of remodeling of the ECM (e.g., increase insolubility of certain ECM components could point to a role for certain cross-linking enzymes, that would provide an actionable point to revert the changes and perhaps attain a therapeutic benefit).
- Since a large part of the manuscript focuses on the HA phenotype and HA is used as a proxy to evaluate ECM content, are the authors able to provide data on the abundance and nature of the enzymes involved in HA synthesis (which isoenzyme(s) of HAS is/are expressed in the different brain regions studied and how do their expression change with aging?).
- As a reader, I am left wondering how the proteomic analysis described in the first part of the paper relates to the second half of the paper. It would have been more obvious to manipulate the expression of the genes encoding some of the ECM proteins found in differential abundance in the aging brain to establish their role in cognitive behavior.

> Minor comment regarding the interpretation of the ECM phenotype:

- Overall, I would encourage the authors to be more rigorous with their wording, and instead of referring to “ECM deposition”, refer to HA deposition, unless they can demonstrate that other components are deposited or accumulate together with HA.

(Remarks on code availability)

Version 1:

Reviewer comments:

Reviewer #1

(Remarks to the Author)

The authors made significant efforts to address all criticism and extended their study accordingly. The presented manuscript version contains a lot of interesting data and provides new insights into the relationships between ECM, microglia, synapses and behavior for the studied brain regions. Despite the lack of direct ECM-manipulating experiments, the study is comprehensive enough to give a conclusive picture for the chosen angle of research. It provides probably the best combination of proteomics and histochemical analyses to date.

(Remarks on code availability)

Custom code used for analyses are available on CodeOcean (capsule 1074 number: 18-559067-0).
The code is well-described and a test data set is provided.

Reviewer #2

(Remarks to the Author)

This revised manuscript has significantly improved and many of the main concerns were addressed. This revision now includes new proteomic analysis of the midbrain in cognitively impaired and unimpaired aged mice, several additional histological markers of ECM, and more thorough analyses of the relationships between ECM, microglia, synapses, and cognitive function.

There are a few additional considerations:

- 1) The authors mention that the Cx3cr1 het mice are quite different than WT mice and differentially express many ECM-related genes, yet they are still included for the microglial contacts with the ECM in Fig 4d. Consider doing this analysis with WT mice as it is difficult to know what conclusions can be made from the data.
- 2) The proteomic analyses examining cognitively unimpaired vs impaired are exciting and provide potential ECM targets for cognitive status. However, there are a few concerns with the approach to classify cognitive status. Firstly, the novel object data in Fig 6a show a few mice with very negative DRs. This suggests that these mice may be cognitively unimpaired as they recognize novel from familiar objects but instead prefer familiar. More information is needed for how these types of data were incorporated into the unsupervised clustering analysis. Secondly, it is also unclear why the open field test values were used to classify cognitive function as this is not a test that assesses cognitive function.
- 3) It is unclear if WFA was analyzed only in the interstitial matrix and excluded the PNN portion. Since PNNs are much more densely labeled with WFA, including the PNN would likely increase the coverage area of WFA. Since the images included

in Fig 3a look like they might not be devoid of PNN, additional clarification is needed.

4) Many of the correlation graphs only show the asterisks but not a legend. It would be helpful to provide the legend on all the graphs for clarity like in Fig 4b.

5) Fig 5l and Fig 5m, it would be helpful to include the title on the graphs to the left (sedentary).

6) Some of the wording (including the Fig 4 subtitle) refers to WFA as a marker of proteoglycans. While WFA does stain a portion of proteoglycans (an element of the GAG chain), it is not a marker of proteoglycans and often changes independently of a change in the protein.

7) While all the data is included in extended Fig 4, it would be helpful to show the same corresponding data in the main figure (either HA or WFA) instead of HA for the Csf1r KO mice and WFA for the Cx3cr1 KO mice.

8) Along these lines, in Fig 3, homer and vGlut1 puncta are shown together in the images, yet the graph included in the main figure is homer only with the co-localized puncta shown in the extended data fig.

(Remarks on code availability)

Reviewer #3

(Remarks to the Author)

The authors have added a substantial amount of new data to the revised manuscript and, in doing so, have satisfactorily addressed most of my comments (staining for aggrecan, Link-1, global labeling of glycoproteins with WFA lectin). The

I have one remaining concern, I do not think the addition of the following sentence: "A key future direction will be to implement proteomic pipelines and enrichment strategies that incorporate ECM-relevant post-translational modifications in order to capture the full complexity of the aging 772 brain matrisome and its relationship with cognition" (page 28, sentence starting line 636 of the revised manuscript) sufficiently cautions future readers of the limitations of the method employed for proteomic data search, likely significantly underestimating collagens and collagen-related protein content in the datasets.

I would strongly encourage the authors to consult with a proteomic expert to clarify how database searching works. Allowing additional variable PTMs during the database search phase of the analysis does not jeopardize, unlike stated in their response, "[the] confident identification of [...] ECM regulatory proteins, synaptic proteins, microglial proteins, etc.". Neither does it increase "false positives" and nor prevents "robust protein-level quantification". This is a missed opportunity and, while the review appreciates that it may not be feasible, due to the time it would take, to re-run the analysis, this needs to be more clearly stated in the result and discussion, rather than being listed as a future direction.

(Remarks on code availability)

Version 2:

Reviewer comments:

Reviewer #2

(Remarks to the Author)

The authors have addressed all of my concerns through reanalysis and clarification of text. The paper is comprehensive and of high scientific value.

(Remarks on code availability)

Reviewer #3

(Remarks to the Author)

I thank the authors for their responsiveness to my comment, which they have now adequately addressed.

(Remarks on code availability)

REVIEWER COMMENTS

Reviewer #1 (Remarks to the Author):

The noteworthy results:

Grey et al. used proteomics to reveal substantial regional differences in ECM composition between VTA and NAc and aging-induced ECM remodeling. Next, the authors relied on a combination of behavioral and histochemical analyses to present interesting positive correlations between the expression of hyaluronic acid, microglia and excitatory postsynapses in VTA, and the latter showed a negative correlation with learning. In a mouse model of microglia ablation, ECM (hyaluronic acid) expression was elevated, but no postsynaptic changes were detected. ECM was not changed in aged Cx3Cr1 heterozygous mice but was downregulated in Cx3Cr1 knockout mice. However, the expression of the postsynaptic marker was similarly reduced in both genotypes, so clear relationship between ECM and synapses emerged from analysis of these mutants.

Will the work be of significance to the field and related fields?

It is of interest and can be significantly improved.

Does the work support the conclusions and claims, or is additional evidence needed?

Indeed, additional analyses are needed.

We appreciate the reviewer for recognizing the interest of this work for the field and for the deep insights into glial-matrix biology apparent throughout the critique. While this revision contains much of the core data presented in the originally submitted manuscript, this a major revision that has incorporated numerous new datasets that help strengthen and refine our previous conclusions, as well as somewhat change the overarching goals of the study. In its current form, this manuscript presents a comprehensive mapping of relationships between basal ganglia ECM, synapse, and microglial aging phenotypes and cognitive aging profiles in mice.

The most significant change is the incorporation of a brand-new tissue proteomic dataset of ECM-enriched midbrain samples obtained from mice that underwent a cognitive battery that enabled us to classify them as impaired or unimpaired relative to young-adult mice using unbiased clustering algorithms. In addition, we now incorporate analysis of several other histological markers of the ECM including the WFA lectin and antibodies for aggrecan and HAPLN1, deeper analysis of synaptic and microglial phenotypes, and statistical analyses of aging-related shifts in relationships between ECM, synapses, microglia, and cognition.

* Please note that all page and line numbers refer to the clean (not track changes) version of the revised manuscript.

1) Unfortunately, in the current version of the manuscript, the proteomics results seem to be insufficiently exploited and not integrated with the follow-up analyses. Proteomics revealed changes in numerous perivascular ECM molecules. However, there is no attempt to show vascular changes or discuss their impact despite the remodeling of perivascular ECM is known to be linked to glial activation and remodeling of neuronal ECM (e.g., DOI: 10.1016/j.matbio.2024.02.007).

We agree with the reviewer that the previous version of this manuscript did not sufficiently integrate proteomic analysis with the follow-up experiments conducted.

⇒ To address this concern, we have carried out histological examination of WFA within our brain regions of interest (Figure 3), to obtain a more general measure of ECM proteoglycan deposition patterns, as well as immunohistochemical analysis of aggrecan and HAPLN1 in tissue from cognitively characterized mice (Figure 7).

⇒ We have generated a second proteomic dataset of midbrain tissue from cognitively characterized mice (Figure 6) that directly links proteomic and cognitive measures.

As the reviewer correctly points out, our proteomic mapping did uncover aging-associated changes in numerous collagens and laminins and other ECM proteins that are largely found in the basement membrane surrounding vasculature in the brain. In the present manuscript, we decided to focus on relationships between ECM proteoglycans, age, and cognitive function for two reasons. First, our initial regional proteomic mapping (Figure 1) indicated that ECM proteoglycans show robust aging-related changes in the midbrain, the brain regions where we previously found signs of microglial aging (proliferation, decreased morphological complexity, inflammatory factor production) months before aging-related microglial changes emerge elsewhere in the brain (doi: 10.1523/JNEUROSCI.1922-21.2022). Second, our novel proteomic dataset from cognitively characterized mice (Figure 6) showed clear relationships between the abundance of numerous lecticans and cognitive phenotypes in late-middle-aged mice (18 months). Thus, we felt that further examination of ECM proteoglycans should be a priority in efforts to link microglial and ECM aging phenotypes with cognition.

Nevertheless, we do appreciate the importance that vascular changes have on cognitive function in the aging brain, as well as the many potential roles that the ECM may play in glial activation and other changes that could contribute to this dysfunction.

⇒ To address the reviewer's concern we have added the citation the reviewer provides within the following comment in the discussion section of the revised manuscript: 'Our regional proteomic mapping experiments revealed aging-related changes in numerous perivascular ECM molecules (e.g., collagens and laminins) that have been linked to microglial activation patterns and remodeling of the neuronal ECM. Given the important links between neurovascular health and cognitive function during aging, it will be critical for future studies to focus on aging-related changes in the basement membrane-associated ECM in the context of cognitive aging.' (p. 27, ln. 617-623)

2) Although alterations in the expression of lecticans were revealed by proteomics, they were not studied by immunohistochemistry. Moreover, analysis was focused on excitatory postsynapses, while most prominently, lecticans are aggregating in WFA-positive lectican-enriched perineuronal nets, which are known to have critical behavioral functions in the studied regions (DOI: 10.1016/j.biopsych.2024.02.003). The study would also benefit from the analysis of presynaptic changes (e.g., VGLUT1) in excitatory synapses, which are related to microglia activity and perisynaptic ECM expression according to previous microglia depletion studies (e.g., see DOI: 10.3390/cells10081862 and refs herein).

In the initial submission, we chose to focus on the prominent ECM scaffold hyaluronan, as numerous lecticans can be anchored to this core ECM component.

⇒ As described in reviewer concern #1, we have expanded on this analysis via histological analysis of WFA abundance in the ventral tegmental area (midbrain) and nucleus accumbens (striatum) (Figure 3b, 3c). This analysis

revealed that while perineuronal nets are found within both regions, they are far less abundant and elaborate compared to those observed in the cerebral cortex and hippocampus, which aligns with Figure 1a of the citation provided by the reviewer (DOI: 10.1016/j.biopsych.2024.02.003). Instead, our analysis, which uses much higher magnification confocal imaging, indicates that WFA label is also distributed relatively evenly across the parenchyma (particularly in the VTA), revealing the presence of a prominent WFA-positive interstitial matrix in these subcortical circuits (Figure 3b). Like hyaluronan, WFA deposition increased in the VTA with advanced age, providing critical validation of the aging-related increases in lectican abundance uncovered by proteomics.

Because there are comparatively fewer perineuronal net structures in the basal ganglia regions we analyzed (particularly in the VTA), we felt that a comprehensive characterization of ECM-synapse interactions using a presynaptic marker as in (DOI: 10.3390/cells10081862) was somewhat tangential to the central goal of this study. However, a prominent observation in our study was that ECM and microglia aging phenotypes occur in the absence of synapse loss.

⇒ **To reinforce histological analyses of synapses included in the initial submission, we carried out follow-up analysis of the pre-synaptic marker VGlut1 (colabelled with the post-synaptic protein homer2).** Similar to homer2 analysis in the initial submission, this new immunohistochemical data showed that the abundance of VGlut1 puncta, as well as adjacent/colocalized Vglut1-Homer2 puncta, did not change with age in either the VTA or NAc (Figure 3f, 3g; Ext. Data. Fig. 3f).

3) This study used proteomics of synaptosomes to study age-dependent changes in excitatory synapses: DOI: 10.1074/mcp.M113.032086. Such measures (although done in VTA and NAc) would be more directly linked to changes in ECM of excitatory synapses as the main focus of the present work. It is worth acknowledging/discussing this approach.

This is an excellent point that we agree needs to be incorporated into the discussion of our data. An additional approach that would provide more synapse specificity in proteomic analysis is through the use of proximity biotinylation studies based on specific excitatory / inhibitory pre- and post-synaptic compartments. We have added the following sentence and the citation provided by the reviewer to the revised discussion section:

'An important future direction for this research will be to examine relationships between basal ganglia ECM, immune, and synapse protein status using isolated synaptosomes, which allow for a more targeted quantification of synapse-associated proteomes' (p. 23, ln. 527-530)

Minor point 1) Understanding microglia relationships with ECM and synapses requires a more substantial characterization of microglia in the studied regions. E.g., the size of microglial soma is an indicator of its activation, and this measure can be easily added to the analysis. Also, relevant information on the expression of microglial proteins can be extracted from the proteomic data. The relationship between microglia, the expression of C1q and ECM could be studied with higher resolution by measuring C1q expression as a function of ECM near or far from synapses.

We agree that more information on microglial aging phenotypes would help further elucidate the relationship that these cells have with the ECM during aging. To address this concern, we have:

⇒ **Carried out Sholl analysis of microglia.** This showed that VTA microglia from aging mice have reduced morphological complexity (Figure 4c; Extended Data Figure 4a), a phenotypic change that we showed previously arises alongside increases in microglial proliferation and inflammatory factor production (doi: 10.1523/JNEUROSCI.1922-21.2022). Note that this analysis was previously included in the extended data figures of the originally submitted manuscript.

⇒ **In the same fields of view, we examined hyaluronan abundance and performed regression analyses between hyaluronan abundance and microglial morphological complexity and microglial cell density.** This analysis indicated that greater hyaluronan abundance in the VTA was significantly associated with greater microglia morphological complexity and density only in young-adult mice, whereas this relationship was lost in late-middle-aged mice (Figure 4b,4c).

⇒ **To provide further insight into the relationship between microglial morphology and ECM abundance, we used Imaris software to reconstruct microglia and hyaluronan in the VTA and identify sites of putative microglia-ECM contacts.** This analysis indicated that young-adult microglia make relatively regular contacts with the hyaluronan matrix, and that aging VTA microglia make far fewer hyaluronan contacts (Figure 4d).

⇒ **We now include abundances of microglia-enriched proteins in our original proteomic dataset (Figure 4a), as well as in the newly generated proteomic dataset in cognitively characterized aging mice (Figure 8d).** In both datasets we see that numerous microglial proteins, including C1qA, C1qB, and C1qC are significantly more abundant with age. This aligns with the observation that there is increased microglial density in this brain region during aging (doi: 10.1523/JNEUROSCI.1922-21.2022).

Together, these observations argue that increased abundance of VTA/midbrain microglia during normative aging are not accompanied by increased physical interaction with the ECM. Indeed, our data suggest the opposite, that aging microglia exhibit a relative disengagement from the ECM. This is an important hypothesis that we now integrate throughout the revised manuscript.

⇒ **In support of this hypothesis, our statistical analyses indicate that increases in the abundance of most microglia-enriched proteins during aging are not strongly aligned with aging-related changes in the ECM.**

- WGCNA analysis in Figure 2d (included in the original submission) indicates that most detected microglia-enriched proteins are not strongly aligned with modules containing the majority of ECM proteoglycans.
- Regression analyses between individual ECM protein abundances and combined total microglia protein abundance within the tissue show that most ECM proteoglycans are not significantly correlated with microglial protein abundance in young-adult and late-middle-aged mice (Figure 8e).
- **Finally, we created correlation network plots that integrate cognitive performance (from both behavioral paradigms) with microglial, ECM, and synapse abundances (using all histological and proteomic data) (Figure 8f).** In these plots, nodes represent individual features, and line thicknesses represent r values of pairwise correlations between traits. This analysis revealed that, in young-adult mice, microglia abundances were strongly correlated to both ECM and synapse abundances, both of which showed relatively pronounced

correlations with cognition (Figure 8f). In late-middle-aged mice, however, microglial relationships with both ECM and synapse abundances were drastically reduced, and ECM-synapse and ECM-cognition relationships were substantially higher (Figure 8f).

With respect to C1q specifically, we draw the reviewer's attention to a recent study showing that increases in C1q during brain aging are related to incorporation of C1q into neuronal ribonucleoprotein complexes (doi: 10.1016/j.cell.2024.05.058). This suggests that microglial derived C1q is unlikely to have a primary role in tagging and phagocytic elimination of ECM in the context of healthy aging. However, doi: 10.1002/glia.24588 suggests that there may be complement-ECM interactions in some brain regions.

⇒ **We have added new discussion regarding ECM-related and ECM-independent roles for the complement system in cognitive aging.** (p. 25-26, ln. 576 - 589)

Minor Point 2) Analysis of microglial mutants shows that the expression of hyaluronic acid is not proportional to the number of synapses. A more comprehensive analysis of ECM markers would possibly help resolve this issue.

We have provided two new analyses of the ECM in the context of Cx3Cr1 deficiency that we believe strengthen the contribution of this data to the overall conclusions of the manuscript.

⇒ **To glean insights into how Cx3Cr1 deficiency alters ECM-relevant aspects of microglial gene expression during aging, we mined a published RNAseq dataset of microglia from young-adult (2 months) and middle-aged (12 months) wild-type (WT) and Cx3Cr1^{GFP/GFP} homozygous (KO) mice (doi: 10.26508/lsa.201900453).** We found that over 50 matrisome-related genes were differentially expressed in microglia from young adult KO compared to WT mice, indicating that this manipulation is poised to alter ECM composition. Moreover, aging had a more prominent impact on microglial expression of ECM-relevant genes in WT mice compared to KO mice, indicating that Cx3Cr1-deficiency alters ECM-related aspects of microglial aging. In particular, microglia from WT mice showed prominent upregulation of genes associated with peptidase and hydrolase activity during aging. This gene expression change was absent in microglia from KO mice, suggesting a potential mechanism for increased abundance of ECM components that we observed in these mice compared to WT (see below). We have added this data and the context it provides to the revised manuscript. (p. 12, ln. 253-264)

⇒ **We have expanded our analysis of ECM in Cx3Cr1^{GFP/GFP} (KO) mice via WFA staining to obtain a more comprehensive analysis of ECM proteoglycans.** We observed a robust increase in WFA deposition in the VTA of KO relative to WT mice (Figure 4i), which is congruent with one study that observed greater proteoglycan deposition in the hippocampus of Cx3Cr1 KO compared to WT (doi: 10.1016/j.bbi.2017.10.002). This, in combination with observations from microglia-deficient FIRE mice indicating that loss of microglia results in greater ECM deposition, leads to the intriguing hypothesis that wild-type microglia have an attenuating effect on ECM deposition relative to contexts where microglia are absent or functionally altered, and that during normative aging this microglial function is lost to some degree. We now integrate this hypothesis into both the results and discussion sections of the manuscript. (p. 13, ln. 283-286). *Note that the hyaluronan data from the original manuscript has been moved to Extended Data Figure 5 and is now also compared to wild-type rather than Cx3Cr1-heterozygous mice. Our data mining of Cx3Cr1-heterozygous microglia (which are also included in the doi: 10.26508/lsa.201900453 study) indicated that

numerous ECM-relevant genes are also differentially expressed in Cx3Cr1-heterozygous mice compared to both wild-type and Cx3Cr1-knockouts (Ext. Data Figure 4). Moreover, multiple recent studies also indicate that Cx3Cr1-heterozygous mice have complex phenotypes that are not simply “intermediate” between those of WT and KO mice (e.g., doi: 10.1186/1742-2094-11-26; doi: 10.1523/JNEUROSCI.3667-11.2011). Hence, we feel that comparing wild-type and Cx3Cr1-knockout mice is a more straightforward approach for gaining insights into how altered microglial aging trajectory may shape ECM status during aging.

⇒ **Incorporation of WFA and focusing on WT vs KO comparisons in this analysis also added important context to our investigation of synapses in these mice.** We observed maintenance of synapse numbers in aging WT mice and *reduced* synapse numbers in KO mice despite these mice also having greater ECM (WFA) abundance during aging. These observations suggest that excess ECM deposition during aging may actually limit synapse formation or plasticity in the aging brain that is needed to maintain stable synapse numbers. We now include this hypothesis in the revised discussion section (p. 25, ln. 566-573).

Minor point 3) Lectican degradation was shown to be driven through ADAMTS4/5 release downstream of D1/D5 receptor activation. There is somatodendritic dopamine release in VTA and VTA projections to NAc, so increased excitatory drive to VTA dopaminergic neurons may result in ECM remodeling in these regions. A discussion of the link from dopamine to ECM would be relevant, particularly if it may explain why “the accumulation of VTA hyaluronan observed in aging sedentary animals was no longer apparent in aging behavior-trained mice”.

This is an excellent point. We have added the following citation (doi: 10.3390/cells9020260) and sentence to the discussion section after our discussion of these data:

“This ECM remodeling may reflect the engagement of somatodendritic dopamine release within the VTA in mice engaged in this task as previous work has shown that ECM degradation is driven through protease release downstream of D1/D5 receptor activation.” (p. 26, ln. 598 - 600)

Is the methodology sound? Does the work meet the expected standards in your field?
Mostly, yes, but a more solid analysis of ECM and microglial markers is expected.

Is there enough detail provided in the methods for the work to be reproduced?
Yes.

Reviewer #2 (Remarks to the Author):

In this manuscript, Gray and colleagues explore the basal ganglia extracellular matrix (ECM) during aging. They identified regional-specific changes in ECM composition during aging with proteomic mapping approaches. Using histology and confocal imaging combined with a reward-based memory test, they found that deposition of HA (a key ECM component that acts as a substrate for many other ECM molecules to bind), specifically in the VTA, aligns with cognitive performance, synapses, and microglial phenotypes. They further showed that HA abundance is altered in mice lacking microglia (Csf1r KO) and in mice lacking Cx3cr1. The authors conclude that glia-ECM interactions influence synapse and cognitive status in aging. The strength of the

study lies in the overall topic of investigation (the ECM is an understudied area of neuroscience), as well as in the comprehensive analysis of the ECM molecules in the aged brain. There were also many sophisticated and consistent correlation analyses aligning these various measures (i.e., HA, synapses, behavioral performance, and microglia) to each other in aging that are quite strong. However, the major concern is the lack of causal role with any one specific measure to another. Some evidence is provided for microglia as being the underlying effector that promotes ECM deposition, yet it is not strong enough to support the conclusions. A clearer mechanistic understanding of how the ECM negatively impacts cognitive function/synapses in aging and how microglia may contribute to this process is needed to give depth to the study.

We thank the reviewer for their thorough review of this manuscript and for their appreciation of the timeliness of the topic for the fields of neuroscience and cognitive aging. We appreciate - and share - the desire for identification of causal links between microglia- ECM- synapses- and cognition. Our primary goal in this study was to provide a sophisticated and comprehensive mapping of basal ganglia ECM, synapse, and microglial aging phenotypes in the context of normative cognitive aging. Such a mapping is currently unavailable and will enable future mechanistic studies centered on understanding glial-matrix contributions to the preservation or loss of cognitive function across the healthy adult lifespan.

Moreover, effective tools for brain ECM manipulation and causal experiments are lacking (See further discussion in point 1 below), and optimal strategies for observing, measuring, and manipulating the brain ECM are only beginning to emerge. We feel that generating and employing such tools is an essential future direction for the field, and one that should be guided by detailed mapping and identification of the ECM-relevant molecules that are most impacted by aging, brain region, and cognitive status.

To that end, we have generated a novel tissue proteomic dataset of ECM-enriched midbrain samples obtained from mice that underwent a cognitive battery that enabled us to classify them as impaired or unimpaired relative to young-adult mice. In addition, we now complement our hyaluronan histology with analysis of several other histological markers of the ECM including the WFA lectin and antibodies for aggrecan and HAPLN1. We also provide deeper analysis of synaptic abundance using a presynaptic marker (VGlut1), as well as microglial aging phenotypes through characterization of microglia morphology and putative contact with the ECM. Furthermore, we provide additional statistical analyses of aging-related shifts in relationships between ECM, synapses, microglia, and cognition. These nuanced approaches and data elucidate numerous entirely novel insights as well as specific candidate proteins and pathways for future mechanistic studies of microglia/ECM/synapse-based mechanisms of cognitive resilience vs decline. We hope the reviewer will agree that these new data greatly expand and strengthen the study and that inclusion of ECM-targeted viral / transgenic manipulations is beyond the scope of this manuscript.

* Please note that all page and line numbers refer to the clean (not track changes) version of the revised manuscript.

1) On a correlational level, the data suggests that HA abundance negatively impacts cognition/synaptic refinement in the aged brain. Yet, the act of performing cognitive tests itself affects HA abundance and synapses, making it challenging to establish clear directionality. The current data do not provide direct causal evidence. A more direct approach would be to explore whether reducing HA has positive impacts on cognition during healthy aging, or whether greater HA limits “synaptic refinement” in brain aging.

We agree with the reviewer that direct manipulation of hyaluronan (or other ECM proteoglycans) in the context of cognitive aging is necessary for the field. However, effective tools for ECM manipulation are lacking. One strategy that various labs have taken is to stereotactically inject ECM-degrading enzymes such as hyaluronidase or chondroitinase ABC. A limitation of these approaches is that they induce mechanical injuries to brain tissue and elicit injury responses of microglia and other glial cells. They are also difficult to titrate and generally result in complete local digestion of tissue hyaluronan or proteoglycans. A minimally-invasive approach that only a few labs have successfully implemented is pharmacological depletion of hyaluronan using the compound 4-methylumbelliferone (4-MU) delivered through chow or drinking water. 4-MU interferes with function of uridine diphosphate (UDP)-glucuronyltransferase, an enzyme involved in hyaluronan synthesis (doi: ./10.1074/jbc.M405918200). Unfortunately, 4MU is also highly unpalatable. We attempted to knock down hyaluronan levels in two distinct cohorts of mice using 4-MU, first through administration in their chow (which they refused to eat) and second by feeding them peanut butter containing the compound (which carries suboptimal changes to diet that can also influence cognitive function). Even after 2 mo of treatment, we did not observe a substantial depletion of hyaluronan in 4MU treated mice (reviewer Figure 1).

Figure 1: Hyaluronan field of view coverage in middle-aged (18 months) control mice and mice administered 4-MU.

⇒ **To address the reviewer’s concern and provide more mechanistic insights into the ECM’s roles in the cognitive aging process, we now provide a novel proteomic dataset from aging mice that were cognitively characterized (Figure 6).** This data not only provided independent verification that having lower ECM and synapse abundances in the midbrain is cognitively beneficial for middle-aged mice, but it identified individual ECM proteins that are potentially key modulators of cognitive aging phenotypes in mice. For example, our data clearly show that the hyalactans, which are ECM proteoglycans that either directly or indirectly interact with hyaluronan scaffolds (HAPLN1-4; aggrecan, brevican, etc) are a promising target for future mechanistic studies of cognitive aging using viral manipulation strategies that are deliverable via retro orbital injection or other minimally invasive strategies.

Additionally, because we are making this dataset available to the broader community, this study is now poised to generate novel hypotheses and enable mechanistic studies of the cognitive aging process in multiple fields.

2) There were several flaws with the microglial manipulation experiments. For example, the authors use constitutive knockouts of Csf1r and Cx3cr1 as their manipulations, which do not rule out the possibility that the observed effects arise from developmental compensations or deficits rather than aging-specific roles. Because both microglia and the ECM play important roles during circuit formation during development, it seems particularly important to delineate as these effects could simply be related to development. Experiments using conditional knockouts during adulthood/aging would be more informative.

We agree with the reviewer that conditional manipulations are an essential future direction for study of microglial-ECM-synapse interactions and that limitations of constitutive manipulations need to be acknowledged.

⇒ **To address the reviewer's concern, we added the following sentence to the results section:** "It will be critical to replicate these observations using conditional manipulations as some findings may be influenced by developmental compensations that arise in constitutive models." (p. 13, ln. 285-288).

Despite these limitations, we feel that the use of two distinct constitutive manipulations (Csf1r and Cx3cr1 knockout) provide valuable insights to this study. As the reviewer will see in our response to concerns #3 and #6, we have carried out additional analysis in these mutant mice that we believe strengthen their contribution to the study.

3) Another important note is the lack of microglial mechanism. Microglial regulation of the ECM has been shown in several papers now, including a report using Cx3cr1 deficiency (PMID: 29017970). Yet, the mechanistic role microglia play in regulating the ECM during healthy aging is unclear from the data. The authors suggest that it may be due to reduced ECM engulfment, but there is a lack of evidence to support the claim. Without a mechanistic view or a cleaner manipulation of microglia in the aged brain, the conceptual novelty and depth of the manuscript are limited.

We agree that mechanistic insight is an essential goal for this field. However, we are sorry that the reviewer does not feel that the comprehensive proteomic mapping of the ECM in multiple brain regions during aging gives ample depth to our study. We are similarly sorry that we have not effectively communicated how 1) regional differences in the impact of aging on the ECM and 2) reduced ECM abundance and synapse number being associated with better cognition constitute highly novel concepts for the field of cognitive aging.

⇒ **To address the reviewer's concern and provide greater mechanistic insight into effects of Cx3cr1 knockout on microglial-ECM interactions during aging, we have:**

1) Mined a published RNAseq dataset of microglia from young-adult (2 months) and middle-aged (12 months) wild-type (WT) and Cx3Cr1^{GFP/GFP} homozygous (KO) mice (doi: 10.26508/lsa.201900453). We found that over 50 matrisome-related genes were differentially expressed in microglia from young adult KO compared to WT mice, *indicating that this manipulation is poised to alter ECM composition via its impact on microglia-ECM interactions*. Moreover, aging had a more prominent impact on microglial expression of ECM-relevant genes in WT mice compared to KO mice, indicating that Cx3Cr1-deficiency alters ECM-related aspects of microglial aging, *beyond any changes in these cells that arose due to developmental compensation for lack of Cx3Cr1*. In particular, microglia from WT mice showed prominent upregulation of genes associated with peptidase and hydrolase activity during aging. This gene expression change was absent in microglia from KO mice, *suggesting a potential mechanism for increased abundance of ECM components that we observed in these mice* (see below). We have added this data and the context it provides to the revised manuscript. (p. 12, ln. 253-264)

2) We have expanded our analysis of ECM in Cx3Cr1^{GFP/GFP} (KO) mice via WFA staining to obtain a more comprehensive analysis of ECM proteoglycans. We observed a robust increase in WFA deposition in the VTA of KO relative to WT mice (Figure 4i), which is congruent with one study that observed greater proteoglycan deposition in the hippocampus of Cx3Cr1 KO compared to wild-type (doi:

10.1016/j.bbi.2017.10.002). This, in combination with observations from microglia-deficient FIRE mice indicating that loss of microglia results in greater ECM deposition, leads to the intriguing hypothesis that wild-type microglia have an attenuating effect on ECM deposition relative to contexts where microglia are absent or functionally altered, and that during normative aging this microglial function is lost to some degree. We now integrate this hypothesis into both the results and discussion sections of the manuscript. (p. 13, ln. 283-286). *Note that the hyaluronan data from the original manuscript has been moved to Extended Data Figure 5 and is now also compared to wild-type rather than Cx3Cr1-heterozygous mice. Our data mining of Cx3Cr1-heterozygous microglia (which are also included in the doi: 10.26508/lsa.201900453 study) indicated that numerous ECM-relevant genes are also differentially expressed in Cx3Cr1-heterozygous mice compared to both wild-type and Cx3Cr1-knockouts (Ext. Data Figure 4). Moreover, multiple recent studies also indicate that Cx3Cr1-heterozygous mice have complex phenotypes that are not simply “intermediate” between those of WT and KO mice (e.g., doi: 10.1186/1742-2094-11-26; doi: 10.1523/JNEUROSCI.3667-11.2011). Hence, we feel that comparing wild-type and Cx3Cr1-knockout mice is a more straightforward approach for gaining insights into how altered microglial aging trajectory may shape ECM status during aging.

3) Incorporation of WFA and focusing on WT vs KO comparisons in this analysis also added important context to our investigation of synapses in these mice. We observed maintenance of synapse numbers in aging WT mice and reduced synapse numbers in KO mice despite these mice also having greater ECM (WFA) abundance during aging. These observations suggest that excess ECM deposition during aging may actually limit synapse formation or plasticity in the aging brain that is needed to maintain stable synapse numbers. We now include this hypothesis in Discussion section (p. 25, ln. 566-573).

4) Furthermore, none of the microglia manipulations are linked to cognitive function / ECM deposition around synapses during brain aging. It remains unclear how these various measures align.

⇒ **To address this concern, we tested late-middle-aged Cx3Cr1^{GFP/GFP} (KO) mice on the same behavioral battery used to characterize wild-type mice in our novel proteomic experiments (Figure 6). This battery included an open field test of anxiety-like behavior, a novel object recognition test, and a T-Maze test of spontaneous alternation.** KO mice did not exhibit deficits in cognitive function beyond those observed in the late-middle-aged WT mice on the open field and NOR tasks.

However, late-middle-aged KO mice showed greater impairment relative to age-matched WT mice on the T-Maze task (**Reviewer Figure 2**). While these data cannot unequivocally link the excess ECM deposition we observe in these mice to their cognitive impairments, they do suggest that the wild-type microglia-ECM phenotype is preferential for some aspects of behavior. **We have added this new**

Figure 2: Performance of young-adult wild-type (4mo; black), middle-aged wild-type (18 mo; grey), and middle-aged Cx3Cr1 knockout (18 mo; white) mice on open field, novel object recognition, and T-Maze behavioral tasks.

behavioral data into Ext. Data Figure 5).

5) More information is needed on the regional-specific roles of microglia. The data show different associations of microglia with the ECM in the VTA vs the NAc, with the NAc showing prominent associations with age that are similar to the associations shown in the VTA throughout life. Yet, none of the microglial manipulations examined the NAc. It is also unclear whether aged microglia are functionally different in their regulation of the ECM in VTA vs NAc.

Analysis of ECM (hyaluronan) and synapses (homer2) in the NAc of Cx3Cr1^{GFP/GFP} (KO) mice was carried out in the original submission (Supplemental Figure 4) of the original manuscript; However, we did not refer to this data in text and we thank the reviewer for catching this oversight.

⇒ **In this revision, this data is now found in Extended Data Figure 5, and we refer to it in the text (p. 12, ln. 270-272).** Because investigating and understanding regional differences in the impact of Cx3Cr1-deficiency on ECM-synapse abundances is not a central goal of the study, we have not elaborated further on these observations in the present manuscript.

6) The Cx3cr1 hets do not recapitulate the WT aging data. It is also unclear from the data whether the young Cx3cr1 hets/ko are similar to the young WT, or whether this is a result of brain aging. It would be helpful to compare these data to WTs.

We agree with this important point. As we highlight in our response to reviewer concern #3, ⇒ **we have now included a direct comparison of ECM abundance (using the WFA lectin) in wild-type (WT) and Cx3Cr1^{GFP/GFP} (KO) mice, and have elected to remove the Cx3Cr1-heterozygous mice from this analysis.** This decision was made on the basis of our data mining, which indicated that Cx3Cr1-heterozygous microglia are not equivalent to WT in ECM-relevant aspects of their gene expression profiles. This analysis showed that KO mice have greater proteoglycan deposition within the VTA at all points of the adult lifespan, which aligns with a previous report showing greater ECM abundance in the hippocampus of young-adult KO mice compared to WT mice (doi: 10.1016/j.bbi.2017.10.002). Thus, this suggests that aging-related ECM accumulation is not observed in KO mice because ECM levels are already elevated in these mice at the start of their adult lifespan. This framework is a valuable addition to the manuscript as it supports the hypothesis that WT microglia have a net antagonistic effect on ECM deposition in the VTA compared to contexts where microglia are altered (Cx3cr1 KO), and that microglial antagonism of the ECM diminishes to some degree during aging.

7) There are many references to “synaptic function” or “synapse status” yet the synaptic findings reported are quite minimal. No attempt to look at synaptic physiology or other correlates of synaptic function.

Unfortunately, slice physiology in aging tissue faces numerous technical hurdles, including poor slice viability and poor reproducibility, that go well beyond the scope of the present study.

⇒ **To address this concern, we have:**

1) Changed all references to ‘synapse function’ and ‘synapse status’ to ‘synapse abundance,’ or similar. Our references to synapse function or status now only come in the abstract and introduction where we discuss the rationale for the present study.

2) Added the following sentence to the discussion section, following description and interpretation of ECM-synapse relationships in our proteomic and histological data: “Additionally, it will be critical to directly measure synaptic activity and capacity for synaptic plasticity and relate such measures to aging-related changes in ECM abundance and composition” (p. 23-24, ln. 530-532).

3) Generated a new immunohistochemical dataset in the VTA and NAc of young-adult and aging mice using both the postsynaptic marker (homer2) and the presynaptic marker (VGlut1). Analysis of adjacent pre- and post-synaptic markers confirmed our previous proteomic and histological data (homer2 only) showing no change in synapse abundance with age, as well as the regional differences we observed between the VTA and NAc (Figure 3f; Ext. Data. Fig. 3h).

Other minor comments:

1) Throughout the paper, there is a distinguishment made between basal ganglia nuclei overall and then in response to aging. This often makes it difficult to follow what is being compared. It would be helpful to clarify (in the figures particularly) which significant differences correspond to specific comparisons and the statistical tests used. It would also be helpful to include more statistical information in the figure legend for the individual panels. Along these lines, providing R values for the correlations and including the key for the lines would be helpful.

Thank you for this feedback. We have made an attempt to more clearly distinguish when we are referring to regional comparisons between basal ganglia nuclei compared to when we are making age comparisons within a region. We have also reformatted our denotation of significance in our figures to help disambiguate aging comparisons from regional comparisons. Additionally, we now include statistics for significant findings in figure legends and have added keys that denote the meaning of different colors in figures containing regression analyses.

2) Fig 1e, the dotted purple line is not at 0 like in the other figures panels. This should be explained. More information is needed for how the fold change was calculated for the panels in Fig 1.

We have reformatted the presentation of the proteomic data to heatmaps with fold-changes adjacent to them. Thus, there are no longer purple lines representing a fold change of 0. We hope the reviewer agrees that this presentation format is more effective and clear.

3) On a similar note, line 95 of page 5 indicates that Tenascin C is upregulated during aging, yet the graph shows a decrease (Fig 1e). This should be clarified.

We thank the reviewer for catching this error. We no longer refer to Tenascin C in parentheses when discussing midbrain glycoproteins. Because the focus of the manuscript does not relate to either the tenascins or glycoproteins in general, we have decided not to try and incorporate this into the main text as it may unnecessarily cause confusion. (p. 6, ln. 110).

Reviewer #3 (Remarks to the Author):

In the manuscript entitled “Extracellular matrix remodeling during aging aligns with synapse microglia and cognitive status,” Gray and colleagues performed a multi-modal analysis of the

extracellular matrix of different brain regions in young and old mice to identify the possible molecular bases leading to the cognitive changes that occur during aging. The significance of the premise of this study is undeniable. The extent of the effort put into this study is also undeniable, as demonstrated by the vast amount of data (8 main multi-paneled figures, 9 extended multi-paneled data figures, and 8 multi-paneled supplemental figures) spanning different scales, from molecular characterization by proteomics to behavioral testing. However, significant flaws in the initial proteomic analysis and the interpretation of the changes in the ECM observed undermine the results derived from the analysis. In addition, the lack of clear connections between the different sections of the manuscript (from proteomics to HA – a non-protein component- patterns, from HA patterns to the involvement of the microglia, etc.) makes the manuscript quite difficult to read and digest.

My specific comments focus on aspects of the first part of the study pertaining to the proteomic experiments and the analysis of the ECM, my primary domains of expertise. I hope they help strengthen the study design and data interpretation.

We thank the reviewer for recognizing the significance of the premise of this study, as well as the effort that has gone into it. To address the valid concerns regarding reproducibility, analysis, and integration of the proteomic dataset with the different sections of the manuscript, we have carried out numerous additional experiments and analyses, have re-written large portions of the text, and have incorporated all the missing information that the reviewer identifies regarding the methodology of our proteomic workflow. While this revision contains the majority of the core data presented in the originally submitted manuscript, this a major revision that has incorporated numerous new datasets that help strengthen and refine our previous conclusions, as well as more thoroughly integrate the different components of the study. In its current form, this manuscript presents a comprehensive mapping of relationships between basal ganglia ECM, synapse, and microglial aging phenotypes and their links to cognitive aging profiles in mice.

The most significant change is the incorporation of a brand new tissue proteomic dataset of ECM-enriched midbrain samples obtained from mice that underwent a cognitive battery that enabled us to classify them as impaired or unimpaired relative to young-adult mice using unbiased clustering algorithms. This dataset included more aging mice compared to the previous proteomic dataset, and its results are consistent with those observed in the original data, providing key validation of our reported findings regarding the aging matrisome. In addition, we now incorporate analysis of several other histological markers of the ECM including the WFA lectin and antibodies for aggrecan and HAPLN1, deeper analysis of synaptic and microglial phenotypes, and further statistical analyses of aging-related shifts in relationships between ECM, synapses, microglia, and cognition.

* Please note that all page and line numbers refer to the clean (not track changes) version of the revised manuscript.

> Major comments regarding the proteomic analysis:

1) Insufficient information is provided in the Material and Method to interpret, let alone attempt to reproduce, the proteomic experiments performed:

- The authors employed two experimental approaches to enrich ECM proteins. However, the authors state that only 4 animals were used per age group (2 males and 2 females; lines 457-458). Is this per experimental approach, or are only 1 male+1 female per age group used for each approach? This should be clarified. The very small cohort size is also of concern.

We appreciate the feedback that our presentation of the sample sizes used in our different proteomic experiments was not clear. The sample sizes for the different proteomic experiments were as follows:

1. **Comparison of experimental approaches (tissue fractionation vs chaotropic extraction) used 8 young-adult (~3 months) mice total - 4 for tissue fractionation and 4 for chaotropic extraction.** In reviewing this section of the Materials and Methods, we realized that we did not provide information regarding the sample sizes for this comparison of experimental approaches. We have now included this information in the revised manuscript. (p. 30-31, ln. 688-691).
 - a. Note: we now provide supplementary tables containing the proteomic data from the comparison of experimental approaches, as this was not provided in the original submission. (Supplementary Table 1).
2. **Regional proteomic mapping of the midbrain and striatum of aging mice also utilized 8 total mice (4 young-adult and 4 aged, equal male and female in each group).** This tissue underwent solubility-based fractionation and all samples underwent proteomic analysis. This information is found (p. 31, ln. 691-694). Furthermore, as the reviewer will see in response to a later concern, we have now provided additional analyses of ECM proteins across solubility fractions in Figure 1.
3. **For the new proteomic dataset incorporated into this revision, midbrain tissue from 6 young-adult mice (4 months; 3 male, 3 female), 5 middle-aged unimpaired (18 months; 2 male, 3 female), and 7 middle-aged impaired mice (18 months; 4 male, 3 female) was used (18 total mice).** This information is found (p. 31, ln. 694-697).

- What amount of protein (ug? mg?) was digested?

Approximately 50-100ug protein was processed for mass-spectrometry analysis. This information is now included in the materials and methods (p. 36, ln. 814).

- A section describing the LC-MS/MS acquisition modality is missing. Specifically, what amount of peptide (ug? ng?) was separated via LC (what parameters were used for the LC separation), and how was the MS/MS acquisition performed (DIA? DDA?). I have tried looking for this information in the MASSIVE repository created to disseminate the dataset but could not find it.

We have now added these details to the Materials and Methods section under the heading LC-MS/MS parameters and database search. Note that the new proteomic dataset in cognitively-characterized aging mice (Figure 6) was generated using a different instrument and MS acquisition scheme than the 2 proteomic experiments in the original submission (Figure 1; Ext. Data Figure 1). **The following information for both MS acquisitions is now provided in the Materials and Methods (p. 36-38, ln. 823-882):**

“LC-MS/MS parameters and database search

For quantitative proteomic experiments comparing ECM enrichment strategies (Ext. Data Figure 1; Supplementary Table 1) and experiments comparing midbrain and striatum proteomes of non-behaviorally characterized aging mice (Figure 1; Supplementary Table 2), LC-MS/MS was carried out as detailed in 93. Approximately 200-500ng peptide amounts were

subjected to LC-MS/MS analysis. Briefly, peptide separation was carried out using reversed-phase chromatography on a 75 μm inner diameter fritted fused silica capillary column, which was packed in-house to a length of 25 cm with 1.9 μm ReproSil-Pur C18-AQ beads (120 \AA pore size). An increasing gradient of acetonitrile was delivered using a Dionex Ultimate 3000 nano-LC system (Thermo Scientific) at a constant flow rate of 200 nL/min. Tandem mass spectra (MS/MS) were acquired in data-dependent acquisition (DDA) mode on an Orbitrap Fusion Lumos Tribrid mass spectrometer (Thermo Fisher Scientific). Full MS1 scans were acquired at a resolution of 120,000, followed by MS2 scans at a resolution of 15,000. The raw LC-MS/MS data were analyzed using the MaxQuant computational platform⁹⁴. The Andromeda search engine, integrated within MaxQuant, was employed for peptide identification against the UniProt reference proteome for *Mus musculus* (UP000000589). Search parameters allowed a maximum of two missed tryptic cleavages and included cysteine carbamidomethylation as a fixed modification and N-termination acetylation and methionine oxidation as variable modifications. A false discovery rate (FDR) threshold of 1% was applied at both peptide and protein levels. Label-free quantification (LFQ) was enabled, with a minimum LFQ ratio count set to one. Precursor ion and fragment ion mass tolerances were set at 20 ppm and 4.5 ppm, respectively. The resulting MaxQuant output files were subsequently used for statistical analysis to identify differentially enriched proteins. Raw data files have been deposited in the MassIVE proteomics repository under accession number MSV000096508. MS metrics and metadata information can be found in Supplementary Table 3.

For quantitative proteomic experiments of the midbrain of behaviorally-characterized mice (Figure 6; Supplementary Table 4), the cytosolic (CS), membrane (M), and insoluble (Insol) fractions were subjected to sequential reduction and alkylation steps using 5 mM tris(2-carboxyethyl)phosphine (TCEP) and 10 mM iodoacetamide, respectively. Following this, the protein aggregation capture (PAC) protocol, as described by Batth et al., (2019)⁹⁵ was employed to purify the reduced and alkylated proteins. Proteins were then enzymatically digested overnight at 37 $^{\circ}\text{C}$ using Lys-C and trypsin proteases. The resulting peptide mixtures were dried completely and prepared for subsequent LC-MS/MS analysis. Dried tryptic peptides were resuspended in 5% formic acid and subjected to liquid chromatography-tandem mass spectrometry (LC-MS/MS) analysis using a Vanquish Neo ultra-high-performance liquid chromatography (UHPLC) system coupled to an Orbitrap Astral mass spectrometer (Thermo Fisher Scientific, Bremen, Germany). Briefly, peptide samples were introduced into a PepSep C18 reverse-phase analytical column (150 mm \times 150 μm , 1.7 μm particle size), maintained at 59 $^{\circ}\text{C}$, and separated using a trap-and-elute workflow. The UHPLC system employed mobile phase A consisting of water with 0.1% formic acid and mobile phase B consisting of acetonitrile with 0.1% formic acid. Peptide separation was achieved using a 15-minute chromatographic gradient with the following composition: 5% B from 0–1 min at a flow rate of 2.45 $\mu\text{L}/\text{min}$; a linear increase from 5% to 15% B from 1–5 min at 1.75 $\mu\text{L}/\text{min}$; 15% to 25% B from 5–12.6 min at 1.75 $\mu\text{L}/\text{min}$; 25% to 38% B from 12.6–13.6 min at 1.75 $\mu\text{L}/\text{min}$; and a rapid gradient from 38% to 80% B between 13.6–13.7 min at 2.45 $\mu\text{L}/\text{min}$, followed by a hold at 80% B until 15 min, maintaining the flow at 2.45 $\mu\text{L}/\text{min}$.

Mass spectrometric acquisition was carried out in data-independent acquisition (DIA) mode on the Orbitrap Astral instrument operating in positive electrospray ionization mode. MS1 survey scans were acquired across an m/z range of 380–980 with a resolution of 240,000, using a normalized AGC (automatic gain control) target of 500% and a maximum injection time of 3 ms. For DIA, sequential isolation windows of 4 m/z were used to comprehensively cover the 380–980 m/z range. Fragment ion (MS2) spectra were acquired at a resolution of 80,000, with a normalized higher-energy collisional dissociation (HCD) energy of 25%, an AGC target of 500%, and a maximum injection time of 7 ms. The raw Thermo .RAW files were analyzed using DIA-NN software, searching against an *in silico* predicted spectral library generated from the *Mus musculus* reference proteome (UniProt ID: UP000000589), as described by Demichev et al.,

(2020)96. The resulting DIA-NN outputs were processed using FragPipe Analyst to identify differentially expressed proteins, following the workflow outlined by Hsiao et al., (2024)97. All raw data files have been deposited in the MassIVE proteomics repository under accession number MSV000096508. MS metrics and metadata information for these data can be found in Supplementary Table 5.

References:

1. Jami-Alahmadi Y, Pandey V, Mayank AK, Wohlschlegel JA. A Robust Method for Packing High Resolution C18 RP-nano-HPLC Columns. J Vis Exp. 2021 May 14;(171). doi: 10.3791/62380. PMID: 34057454.
2. Batth, Tanveer S. et al. (2019). Protein Aggregation Capture on Microparticles Enables Multipurpose Proteomics Sample Preparation. Molecular & Cellular Proteomics, 18 (5), 1027 – 1035.
3. Demichev, V., Messner, C.B., Vernardis, S.I., Lilley, K.S., and Ralser, M. (2020). DIA-NN: neural networks and interference correction enable deep proteome coverage in high throughput. Nat Methods 17, 41-44.
4. Hsiao Y, Zhang H, Li GX, Deng Y, Yu F, Valipour Kahrood H, Steele JR, Schittenhelm RB, Nesvizhskii AI. Analysis and Visualization of Quantitative Proteomics Data Using FragPipe-Analyst. J Proteome Res. 2024 Oct 4;23(10):4303-4315. doi: 10.1021/acs.jproteome.4c00294. Epub 2024 Sep 10. PMID: 39254081.

- A section describing the parameters used for the database search is also missing: The MassIVE repository indicates that no post-translational modifications were included during the database search. If true, this is a critical limitation of the experimental design that undoubtedly resulted in the underestimation of the number of proteins identified and all the quantitative analyses performed (extensive studies have been published on the topic that the authors could refer to).

Cysteine carbamidomethylation was included as a fixed modification and N-termination acetylation and methionine oxidation as variable modifications in the search parameters for MaxQuant. **We have now included this information in the additional details added to the methods sections under the heading LC-MS/MS parameters and database search (see response to previous reviewer concern).**

We appreciate the reviewer's insightful comment regarding the omission of more ECM-relevant PTMs in our database search parameters, and we acknowledge that the exclusion of variable PTMs - particularly hydroxylation, sulfation, and glycosylation, which are prevalent in brain ECM proteins - can contribute to an underestimation of ECM protein identifications. This is indeed a limitation of our current datasets and an important consideration for interpreting quantitative results.

Our rationale for initially omitting variable PTMs was to prioritize confident identification of not only core matrisome proteins, but also ECM regulatory proteins, synaptic proteins, microglial proteins, etc. using a conservative and computationally efficient search strategy, particularly given the large size of both of our proteomic datasets and the complexity introduced by highly modified peptide forms. At the time of analysis, we aimed to minimize false positives and ensure robust protein-level quantification, though we recognize this may have come at the cost of reduced sensitivity for PTM-rich ECM constituents. Importantly, our primary findings are based on relative comparisons within a consistent experimental and analytical framework. As such, although the absolute number of identified proteins may be underestimated, the comparative

patterns and conclusions regarding age-related changes remain valid within the defined search space.

Nevertheless, this is a critical point and we have added the following text describing this technical limitation to the revised discussion section:

“In this regard, there is evidence that unique post-translational modifications on ECM proteoglycans (i.e., hydroxylation, sulfation, and glycosylation) can alter key aspects of ECM physiology and its regulation of neuronal function during aging. While our primary findings are based on relative comparisons within consistent experimental and analytical frameworks, our proteomic database searches did not include assessment of these ECM-relevant post-translational modifications, meaning that we likely underestimated matrisome protein abundance and cannot provide insights into more subtle changes in matrisome composition. A key future direction will be to implement proteomic pipelines and enrichment strategies that incorporate ECM-relevant post-translational modifications in order to capture the full complexity of the aging brain matrisome and its relationship with cognition.” (p. 28, ln. 629-639).

Citations:

1. Fawcett, J. W. & Kwok, J. C. F. Proteoglycan Sulphation in the Function of the Mature Central Nervous System. *Frontiers in Integrative Neuroscience* **16**, (2022).
2. Foscarin, S., Raha-Chowdhury, R., Fawcett, J. W. & Kwok, J. C. F. Brain ageing changes proteoglycan sulfation, rendering perineuronal nets more inhibitory. *Aging (Albany NY)* **9**, 1607–1622 (2017).

2) Insufficient information is provided to assess the rigor of the quantitative analysis:

- Are all proteins identified included in the analysis, or did the authors apply thresholds to exclude low-probability IDs (what were the FDRs applied? At what level? PSM? Protein?).

An FDR threshold of 1% was applied at both peptide as well as protein levels during analysis on the MaxQuant pipeline. **We have now included this information in the additional details added to the methods sections under the heading *LC-MS/MS parameters and database search*. (p. 37, ln. 839-840)**

- It is customary for a study reporting proteomic data to provide the details of the MS metrics collected for each sample, i.e., the non-normalized and normalized precursor ion intensity for each replicate, and obtain via pipeline in a supplementary table. Here, the authors only report the “log2-normalized intensity values” (note the typo in the word “intensity” in the supplementary table). The headers of the columns (“OF1”, “OF2”, etc.) should be spelled out.

We have now provided supplementary tables containing the MS metrics collected for each sample for both proteomic datasets in the revised manuscript. (Supplementary Table 3; Supplementary Table 5)

Additionally, we have provided new supplementary tables with log2-normalized intensity values for each solubility fraction examined, rather than the combined tables provided in the original version of this manuscript. (Supplementary Table 2; Supplementary Table 5).

Finally, we have corrected the typo of the word 'intensity' and have spelled out column headers in supplementary tables.

- Since the authors have performed the proteomic analysis on ECM-enriched samples obtained using two different experimental approaches, it is surprising that no comparative analysis is presented.

While we agree with the reviewer that a comprehensive comparison of the two approaches would be interesting for a number of different fields, such an analysis is quite tangential to the overall goal of the manuscript, which is to provide a comprehensive and sophisticated mapping of ECM, synapse, and microglial phenotypes associated with better or worse cognition during normative brain aging. Because very few studies have examined different ECM enrichment strategies in brain tissue, our motivation for comparing the two ECM enrichment approaches was to evaluate what method would provide the most useful data for our purposes. For this reason, we constrained our analysis of the two approaches only to core matrisome proteins. The results were insightful in that they revealed that while the two approaches identified similar numbers of structural ECM proteins (core matrisome) in brain tissue, solubility-based fractionation on average was more efficient at enriching ECM proteoglycans while chaotropic digestion better enriched more fibrous ECM proteins like collagens in brain tissue. **To better illustrate this, we have reformatted Extended Data Figure 1 to more clearly illustrate this point.** In its current version, we now use heatmaps of the z-scored protein intensities of each individual protein for the two experimental conditions, with the fold changes depicted as a bar plot adjacent to them. Additionally, we have separated the different subclasses of proteins (glycoproteins, proteoglycans, collagens) for more clarity.

While we do not believe that a deeper analysis is needed for the purposes of this manuscript, we now provide supplementary tables containing the data from each approach for the broader community to pursue their own analyses. (Supplementary Table 1).

- Unless I missed that, the tables are not cited in line with the text and no table legends are provided.

Thank you for pointing out this omission – **we now cite supplemental tables containing all proteomics data generated in this study throughout the Materials and Methods section. (p. 35-38, ln. 823-882)**

- While I appreciated that the authors deposited their raw MS files to MassIVE, the lack of meta-data (for example, key linking .raw files to sample type; LC_MS/MS acquisition parameters) prevents proper evaluation, re-analysis, and re-use. The authors should provide additional meta-data information. I suggest adopting the Sample and Data Relationship Format (SDRF), now broadly used in proteomics (PMID: 37875519).

Supplementary tables containing all necessary meta-data related to both proteomics experiments are now provided. (Supplementary Table 3; Supplementary Table 5).

> Minor comment regarding the proteomic analysis:

- I would encourage the authors to refrain from using the word "novel" to describe the ECM-

optimized proteomic workflows published several years ago and already broadly adopted by the scientific community.

This is a fair point. We have refrained from asserting that the ECM proteomic workflows are novel in the revised manuscript.

- Figure 1: could the authors add the gene symbols of the proteins found in statistically significant differential abundance in the volcano plot panels (c)? It would simplify Figure 1 and allow the elimination of panels d- f.

Because brain ECM research is a vastly understudied area of neuroscience, we do believe there is value in presenting the full mapping of the aging basal ganglia matrisome, separated by categories (rather than only labeling significantly altered proteins in the volcano plot). However, **to improve and simplify the presentation of these data, we have changed the format of the data in Figure 1 from bar plots with fold-changes overlaid via a line plot (as in original manuscript) to heatmaps of the z-scored protein intensities** of each individual protein for young and aged mice, with the fold changes adjacent to them in the form of a bar plot. This data presentation highlights the inter-subject variability in the data as well as the extent of the aging-related changes that we observed for each individual ECM protein.

> Major comments regarding the interpretation of the ECM phenotype:

- Since the proteomic analysis identified compositional differences across brain regions and ages, it would have been interesting to validate some of these differences (including those relevant to HA metabolism and functions) using orthogonal approaches, such as immunostaining.

We agree that validation efforts are important for this study.

⇒ **To address this concern, we carried out additional experiments with a focus on the lectican family of ECM proteoglycans**, due to their relative enrichment in brain ECM and indications from our original proteomic analysis of midbrain and striatum tissue (Figure 1) showing aging-related increases in the abundance of numerous lecticans and hyaluronan and proteoglycan link proteins (hapln1; hapln2), with which they interact. Moreover, our novel proteomic dataset from cognitively characterized mice (Figure 6) revealed clear relationships between lectican abundance and cognitive phenotypes in late-middle-aged mice (18 months).

- 1) **We now provide additional histological data using Wisteria floribunda agglutinin (WFA), a lectin that preferentially labels N-acetylgalactosamine residues on glycosylated ECM proteins.** WFA is easily the most widely used ECM marker in neuroscience due to the enrichment of proteoglycans in the brain parenchyma. Our analysis of WFA in the ventral tegmental area (midbrain) and nucleus accumbens (striatum) further validated our previous proteomics and hyaluronan histology, indicating that aging-related ECM accumulation preferentially impacts the midbrain (Figure 3).
- 2) **We now include immunohistochemical analysis of aggrecan and HAPLN1 in tissue from cognitively characterized mice (Figure 7).** These analyses were consistent with proteomic analysis in that we observed lower HAPLN1 deposition in cognitively unimpaired aging mice compared to impaired mice (Figure 7b). Although not statistically significant we observed the same trend with aggrecan, which is also consistent with proteomic experiments.

- The transition from the proteomic analysis to the focus on HA is unclear.

We appreciate this feedback. Our original rationale for focusing on hyaluronan was twofold. First, there is vast heterogeneity across brain regions in which specific ECM proteins are expressed. Hence, we reasoned that a more ubiquitous ECM component like hyaluronan was a more reliable proxy for brain region differences in aging-induced ECM remodeling. Second, our original regional proteomic mapping (Figure 1) indicated that hyaluronan and proteoglycan link protein 2 (HAPLN2) was the most upregulated ECM proteoglycan with age. In the manuscript's current form, we now present the WFA data before the hyaluronan data, which makes this transition more natural.

- Since the authors interpret their observation as resulting from “ECM remodeling”, it would have been interesting to stain for brain sections where HA patterns are shown to differ for total collagen content as well (using anti-collagen antibodies or Masson's trichrome or Picrosirius red stains). This would be a more definitive proof of ECM remodeling.

This is a good suggestion. As we highlight above, we have elected to provide follow-up analysis using histology for various ECM proteoglycans using the WFA lectin and antibodies for HAPLN1 and aggrecan. However, in our discussion of the ‘remodeling’ of the hyaluronan matrix in response to behavioral training (Figure 5, Figure 7), we now use the phrase ‘hyaluronan matrix remodeling,’ or similar, rather than the broader phrase of ‘ECM remodeling.’ Additionally, we have added text to the discussion section highlighting that staining for additional ECM components, including collagens, is a critical future direction for the field. (p. 27, ln. 617-623)

- The authors indicate having summed the intensity values of each identified protein across all solubility fractions (lines 570 – 571) but do not provide a rationale for doing so. This is a missed opportunity since their study otherwise could have provided valuable insights not only on the compositional changes that occur in the brain ECM during aging but also changes in ECM protein solubility that may have occurred during aging, which could inform on the state of remodeling of the ECM (e.g., increase insolubility of certain ECM components could point to a role for certain cross-linking enzymes, that would provide an actionable point to revert the changes and perhaps attain a therapeutic benefit).

We thank the reviewer for this important suggestion. Our rationale for summing protein intensity values across solubility fractions was to best represent the total abundance of protein in the tissue, regardless of solubility status, for downstream WGCNA analysis. **We now add this justification to the revised Materials and Methods (p. 39, ln. 891-893).**

As the reviewer correctly points out, this approach does not allow any aging-associated shifts in ECM protein solubility to be examined.

⇒ **To address this concern, we have carried out additional analyses of the midbrain and striatum proteomic data (Figure 1) to map ECM protein abundance across solubility fractions and the relationship of solubility with age in both regions.** This analysis provided several important insights that are now incorporated into the discussion of these data. First, it suggested that midbrain ECM protein abundances were highest in insoluble and membrane fractions, and that age-related increases in ECM protein abundance were relatively uniform across solubility fractions (Figure 1c-1e). In the striatum, ECM proteins were relatively enriched in membrane fractions, and ECM protein abundance disproportionately increased with age within insoluble fractions (Figure 1c-1e).

⇒ **To probe this data further, we determined the number of ECM proteins showing significant differences in solubility with age by making pairwise comparisons of ECM protein abundance using proteins that were found in at least two different**

solubility fractions. This analysis revealed that just 6 percent of midbrain ECM proteins (found in multiple fractions) became more soluble in older animals and none became more insoluble. In contrast, no striatal ECM protein became more soluble and 18 percent became more insoluble (Ext. Data Fig 1). These results indicate there are important regional differences in the subcellular localization and/or solubility of ECM proteins at different points of the lifespan. We have added these data and a discussion of them to the revised manuscript (p. 5-6, ln. 93-107).

⇒ **Additionally, in our analysis of the newly generated proteomic dataset from young-adult and late-middle-aged mice that were cognitively characterized, we kept the membrane, cytoskeletal, and insoluble fractions separate and present the data accordingly (Figure 6).** This approach revealed that ECM proteoglycan abundances in the membrane and cytoskeletal fractions show robust relationships with cognitive status, whereas ECM glycoproteins in insoluble fractions most strongly map onto cognitive status (Figure 6; Extended Data Figure 7). *This suggests that both the abundance and subcellular localization/solubility of proteoglycans play roles in maintaining function with advanced age,* which is a critical added layer of depth to these proteomic datasets.

- Since a large part of the manuscript focuses on the HA phenotype and HA is used as a proxy to evaluate ECM content, are the authors able to provide data on the abundance and nature of the enzymes involved in HA synthesis (which isoenzyme(s) of HAS is/are expressed in the different brain regions studied and how do their expression change with aging?).

The relative contribution of the hyaluronan phenotype is much smaller in the revised manuscript due to our generation of new histological data using the WFA lectin and antibodies specific for aggrecan and HAPLN, as well as our generation of the new proteomic dataset in cognitively characterized aging mice. Thus, we feel that this information is less essential in this revision than it was in the original submission.

- **Regardless, we have probed both of our proteomic datasets for both hyaluronan synthase proteins (HAS1, HAS2, and HAS3) and hyaluronidases (HYAL1-4). Interestingly, we did not detect any of these proteins in our tissue proteomic dataset, indicating that different tissue processing approaches are needed.**

Because the overall goal of this study is not to elucidate ECM regulatory mechanisms that change with advanced age (an important future direction), but rather to understand what aging-related ECM changes most prominently align with cognitive processing, we elected to direct our follow-up efforts towards generating new data that is more central to the premise of this study. **However, to provide more information on the changing ECM regulatory environment, we now present and discuss protein abundances of all ECM regulatory proteins detected in both of our proteomic datasets (Figure 1d; Ext. Data Figure 8).**

- As a reader, I am left wondering how the proteomic analysis described in the first part of the paper relates to the second half of the paper. It would have been more obvious to manipulate the expression of the genes encoding some of the ECM proteins found in differential abundance in the aging brain to establish their role in cognitive behavior.

We are sorry to hear that the reviewer did not feel that the introduction clearly and cohesively set up the rationale for the study; however, we welcome and appreciate this important feedback. As we highlight above, this manuscript presents a comprehensive mapping of relationships

between basal ganglia ECM, synapse, and microglial aging phenotypes and the links that these phenotypes have to cognitive aging in mice. We have edited and expanded the abstract, introduction, and portions of the results sections to try and more clearly delineate this overarching goal.

We agree that targeted manipulation of specific ECM proteins (or regulatory proteins that impact ECM proteins) is an essential future direction to reveal causal relationships to cognition. Indeed, we believe that the data presented in this manuscript will enable those sorts of experiments within the field in an unprecedented fashion. However, including such experiments is well beyond the scope of the present study.

> Minor comment regarding the interpretation of the ECM phenotype:

- Overall, I would encourage the authors to be more rigorous with their wording, and instead of referring to “ECM deposition”, refer to HA deposition, unless they can demonstrate that other components are deposited or accumulate together with HA.

We now more rigorously discuss our ECM data by minimizing our use of phrases like ECM deposition and indicating the particular ECM components under discussion.

REVIEWER COMMENTS

Reviewer #1 (Remarks to the Author):

The authors made significant efforts to address all criticism and extended their study accordingly. The presented manuscript version contains a lot of interesting data and provides new insights into the relationships between ECM, microglia, synapses and behavior for the studied brain regions. Despite the lack of direct ECM-manipulating experiments, the study is comprehensive enough to give a conclusive picture for the chosen angle of research. It provides probably the best combination of proteomics and histochemical analyses to date.

We are very grateful for the reviewer's generous and encouraging assessment of our work. We appreciate the recognition of our efforts to revise and extend the manuscript, as well as the positive evaluation of the integrative approach combining proteomic and histochemical analyses. We also acknowledge the important point raised regarding the absence of direct ECM-manipulating experiments and agree that future studies incorporating such approaches will be essential to further build on our findings

Reviewer #1 (Remarks on code availability):

Custom code used for analyses are available on CodeOcean (capsule 1074 number: 18-559067-0).

The code is well-described and a test data set is provided.

Reviewer #2 (Remarks to the Author):

This revised manuscript has significantly improved and many of the main concerns were addressed. This revision now includes new proteomic analysis of the midbrain in cognitively impaired and unimpaired aged mice, several additional histological markers of ECM, and more thorough analyses of the relationships between ECM, microglia, synapses, and cognitive function.

We thank the reviewer for their encouraging comments and thoughtful assessment of our work. We are pleased that the reviewer agrees that our additional proteomic analyses, expanded histological characterization of the ECM, and more detailed evaluations of the relationships between ECM, microglia, synapses, and cognitive function have strengthened the manuscript.

Based on the reviewer's most recent feedback, we have made several additional changes throughout the manuscript to further clarify our rationale, highlight important caveats, and emphasize future directions, which we highlight below.

There are a few additional considerations:

1) The authors mention that the Cx3cr1 het mice are quite different than WT mice and differentially express many ECM-related genes, yet they are still included for the microglial contacts with the ECM in Fig 4d. Consider doing this analysis with WT mice as it is difficult to know what conclusions can be made from the data.

We agree that analysis of microglial-ECM contact during aging in WT mice should be an important future direction. Our rationale for using Cx3cr1 heterozygous mice in this analysis (Fig. 4d) balanced the potential caveats of using Cx3cr1 hets with the limitations of immunostaining-based analyses of microglial morphology and putative contact with surrounding structures in aging mice. The GFP reporter mice allow clear visualization of fine distal processes that are not always fully captured by IBA1 immunostaining. Immunostaining with P2RY12, which does typically reveal distal processes, carries its own caveats as expression levels of P2RY12 vary (usually decline) with aging. Optimal approaches for carrying out this analysis in aging WT mice would ideally include staining with multiple microglial markers and leveraging of super-resolution STED based approaches. Use of membrane-anchored fluorescent reporters and distinct microglial cell lines would also provide more definitive insights into microglial-ECM interactions in young adult and aging mice. However, we feel that these additional analyses would require significant further validation that is beyond the scope of this study, particularly at this late stage.

However, as the reviewer notes, Cx3cr1 het mice exhibit transcriptome differences from WT mice. To be transparent about this caveat and to highlight critical next steps warranted by this analysis, we have added the following sentences to the revised manuscript following discussion of these data in the results section:

“Importantly, an emerging body of evidence indicates that Cx3Cr1^{EGFP/+} microglia exhibit some phenotypic differences from wild-type (WT) microglia, including expression of some ECM-relevant genes (Ext. Data Fig. 5). Thus, future work using distinct microglia reporter lines, immunostaining approaches in WT mice that faithfully label microglial fine distal processes, and super-resolution microscopy will be needed to expand these analyses of microglial-ECM contact”
(p 12; ln 249-254)

2) The proteomic analyses examining cognitively unimpaired vs impaired are exciting and provide potential ECM targets for cognitive status. However, there are a few concerns with the approach to classify cognitive status. Firstly, the novel object data in Fig 6a show a few mice with very negative DRs. This suggests that these mice may be cognitively unimpaired as they recognize novel from familiar objects but instead prefer familiar. More information is needed for how these types of data were incorporated into the unsupervised clustering

analysis. Secondly, it is also unclear why the open field test values were used to classify cognitive function as this is not a test that assesses cognitive function.

These are thoughtful considerations that highlight the inherent challenges of using spontaneous behavioral assays in rodents (particularly mice) since performance can be shaped not only by individual differences in cognitive/memory function, but also by different exploration strategies, anxiety levels, motivation, etc. For example, while we agree that negative discrimination ratios on the NOR test suggest that some late-middle-aged mice *do* discriminate the novel and familiar objects, their preference for the familiar object is a distinct behavioral phenotype from what we observed in the young-adult mice that may reflect divergent exploratory strategies or anxiety states. Ultimately, challenges with interpretation such as this are what drove us to implement unsupervised clustering approaches across multiple paradigms to capture a wider spectrum of behaviors while minimizing the impact that a single behavioral discrepancy might have on the classification of individual mice.

We also agree that the open field test is not a direct measure of cognitive function, but rather a test of locomotor activity, exploratory strategy, and anxiety-like behavior. Critically, only estimates of anxiety-like behavior (% time in center are arena vs. perimeter) from the open field test were included in our clustering analysis. Because heightened anxiety can reduce exploratory drive and confound both NOR and T-maze performance (PMID: 17429409; PMID: 38541632; PMID: 22760949), we reasoned that incorporating this measure would enable the clustering framework to better distinguish animals with lower recognition or spatial task performance due to anxiety-related suppression of exploration from those with performance deficits more consistent with memory decline. In this way, the anxiety metric functions as a behavioral covariate that increases the specificity of our classification, rather than as a cognitive readout itself.

Finally, we feel it is important to highlight that our analysis represents just one approach to link behavioral variability with underlying molecular states and should be considered as a starting point for the field rather than a definitive framework. By providing all behavioral and proteomic source data with this publication, we offer the field a resource to explore alternative classification strategies, examine domain-specific cognitive functions, and assess how different analytical approaches influence the interpretation of behavior-molecular relationships. Future work will be critical to refine these strategies, incorporate additional behavioral or physiological measures, and further clarify the molecular signatures associated with behavioral variability in aging.

Given these important points, we have revised the manuscript text to clarify our approach, rationale, and important future directions in the following ways:

1. Addition of the following text to the results section describing this analysis:
“Leveraging this variability, we fed estimates of anxiety-like behavior from the open field test, discrimination indices from the NOR test, and alternation

indices from the T-maze into an unbiased hierarchical clustering algorithm to cognitively classify all mice tested in this pipeline. Importantly, while anxiety-like behavior does not directly reflect cognitive abilities per se, these measures were included to account for potential confounding effects of anxiety on exploration-based cognitive tasks such as the NOR and T-maze tests.” (p 17; ln. 379 - 384).

2. Addition of the following text to the discussion section: **“Importantly, while our analysis presents one strategy for behavioral classification of aging mice, different classification strategies may highlight distinct aspects of behavioral heterogeneity across the lifespan. It will be important for the field to apply alternative approaches to evaluate how different behavioral analysis strategies influence the relationships between behavior and distinct molecular states.”** (p. 28; ln. 637-641)

3) It is unclear if WFA was analyzed only in the interstitial matrix and excluded the PNN portion. Since PNNs are much more densely labeled with WFA, including the PNN would likely increase the coverage area of WFA. Since the images included in Fig 3a look like they might not be devoid of PNN, additional clarification is needed.

The reviewer raises an important point. In the previous revision, the data presented in Figure 3c included all WFA signal in the field of view, regardless of whether it was associated with a PNN accumulation or not. We elected to analyze the data this way for several reasons. First, one of the more important purposes of this histological figure is to provide orthogonal validation of the proteomic analyses in Figure 1 showing that ECM protein abundance increases with age. Because the ECM proteins contributing to these proteomic samples came from both the interstitial matrix and ECM accumulations around neurons (i.e., PNNs) or vasculature (i.e., basement membrane), we felt it was appropriate to include all WFA signal in this analysis. Second, a mapping of aging-associated changes in WFA abundance in the VTA does not currently exist in the literature, thus we felt that providing this most baseline assessment was most relevant for the field.

To provide more clarity regarding this analysis, we have revised the results section describing these data to explicitly state that all WFA was included in this analysis regardless of its association with the interstitial matrix or perineuronal nets (p. 9, ln. 182-183).

The question of whether increases in WFA signal with age arise due to changes in the interstitial matrix, WFA accumulations around neurons and/or vasculature, or both is both interesting and important in the context of this manuscript.

→ To evaluate whether the aging-related increase in WFA within the VTA was driven primarily by increases in the abundance of the interstitial matrix or larger accumulations around neurons and/or vasculature, we implemented a size filter of 150

pixels to separate smaller WFA puncta from larger accumulations. WFA puncta smaller than 150 pixels were classified as putative interstitial matrix and those larger than 150 pixels were classified as putative PNN / perivascular accumulations. This analysis indicated that increases in WFA within the VTA of older mice arises from both increases in the abundance of the putative interstitial ECM and increases in the size or abundance of putative PNN / perivascular accumulations. We have included this new analysis as an extended data figure in this revision (Ext. Data Fig. 3), and have described these findings in the text (p. 9 line 185-191).

Additionally, we have added the following information to the revised methods section: **“For the size-based analysis of WFA (Ext. Data Fig. 3), WFA puncta from thresholded images were filtered into two distinct groups based on pixel size: a putative interstitial WFA group (< 150 pixels), or a putative PNN / perivascular WFA accumulation group (>150 pixels). Field-of-view coverage measures were then calculated for each group independently.”** (p. 34, ln. 776-780).

4) Many of the correlation graphs only show the asterisks but not a legend. It would be helpful to provide the legend on all the graphs for clarity like in Fig 4b.

We now provide legends with asterisks next to groups exhibiting statistical significance in all correlation graphs throughout the main figures and the extended data figures.

5) Fig 5l and Fig 5m, it would be helpful to include the title on the graphs to the left (sedentary).

The omission of ‘Sedentary’ in Figures 5l and 5m in the previous revision was unintentional. We have added the title ‘Sedentary’ to the appropriate panels in Figure 5l and 5m. Additionally, we have changed the titles in Figure 7g to ‘Sedentary’ and ‘Behavior’ from ‘Sedentary Mice’ and ‘Behaving Mice’ to match the titles in Figure 5.

6) Some of the wording (including the Fig 4 subtitle) refers to WFA as a marker of proteoglycans. While WFA does stain a portion of proteoglycans (an element of the GAG chain), it is not a marker of proteoglycans and often changes independently of a change in the protein.

We thank the reviewer for pointing this out and agree that our wording was somewhat imprecise. WFA labels glycan structures rather than proteoglycan core proteins, and as the reviewer notes, these signals can vary independently. We have revised the Fig. 4 subtitles to read “Increased WFA deposition with microglia ablation” and “Increased WFA deposition with microglia Cx3Cr1 deficiency” to be more precise.

We have also updated relevant text to remove references to WFA as a direct “marker of proteoglycans” and now more accurately describe it as labeling glycan components. (p. 9, ln. 177).

7) While all the data is included in extended Fig 4, it would be helpful to show the same corresponding data in the main figure (either HA or WFA) instead of HA for the Csfr1 KO mice and WFA for the Cx3cr1 KO mice.

We agree with the reviewer that it would be helpful to show the same data for the Csfr1 KO and Cx3Cr1 KO. The reason the data was not matching in the previous revision was that we had not examined WFA in the Csfr1 KO mice. Because this was the only mouse line in this manuscript we had not collected complementary WFA and hyaluronan histological data for, we performed additional ECM analyses in the Csfr1 KO mice using WFA to round out our dataset. This analysis indicated that the same accumulation pattern observed with hyaluronan in the Csfr1 knockouts was also present when using WFA. Thus, in this revision we have replaced the Csfr1 hyaluronan data with WFA data (Figure 4) and have moved the hyaluronan data to Extended Data Figure 6.

8) Along these lines, in Fig 3, homer and vGlut1 puncta are shown together in the images, yet the graph included in the main figure is homer only with the co-localized puncta shown in the extended data fig.

The reason we had kept the Homer2-only data in the main figure in the previous revision was because these data came from the same images used in the ECM-synapse proximity analysis in Figures 3h-3i. However, we do agree with the reviewer that the co-localized puncta data (Vglut1-Homer2) present a more rigorous quantification of putative synapse abundance that is more relevant for this manuscript. Therefore, in this revision we have replaced Figure 3g with the colocalized VGlut1-Homer2 data and moved the Homer2 data to Extended Data Figure 4.

Reviewer #3 (Remarks to the Author):

The authors have added a substantial amount of new data to the revised manuscript and, in doing so, have satisfactorily addressed most of my comments (staining for aggrecan, Link-1, global labeling of glycoproteins with WFA lectin). The

I have one remaining concern, I do not think the addition of the following sentence: "A key future direction will be to implement proteomic pipelines and enrichment strategies that incorporate ECM-relevant postn -translational modifications in order to capture the full complexity of the aging 772 brain matrisome and its relationship with cognition" (page 28, sentence starting line 636 of the revised manuscript) sufficiently cautions future readers of the limitations of the method employed for proteomic data search, likely significantly underestimating collagens and collagen-related protein content in the datasets.

I would strongly encourage the authors to consult with a proteomic expert to clarify how database searching works. Allowing additional variable PTMs during the database search phase of the analysis does not jeopardize, unlike stated in their response, "[the] confident identification of [...] ECM regulatory proteins, synaptic proteins, microglial proteins, etc.". Neither does it increase "false positives" and nor prevents "robust protein-level quantification". This is a missed opportunity and, while the review appreciates that it may not be feasible, due to the time it would take to re-run the analysis, this needs to be more clearly stated in the result and discussion, rather than being listed as a future direction.

We thank the reviewer for this thoughtful follow-up and for further clarifying the methodological considerations related to database searching strategies and the limitations they pose in this study. We agree that the limitations in our proteomic database search - particularly with respect to collagens and collagen-related proteins - should be made more explicit in the manuscript rather than only noted as a future direction. In the revised manuscript, we now directly state in the results section immediately following discussion of these data that our database search strategy likely underestimates the abundance of some ECM proteins, including collagens, with the addition of the following sentences:

“An important cautionary note is that the proteomic data search strategy used did not include ECM-relevant post-translational modifications, which very likely resulted in an underestimation of certain highly cross-linked and modified ECM proteins such as collagens. It will be critical for future studies to determine the extent to which this limitation affects the detection and quantification of all classes of brain ECM proteins.” (p. 6-7, ln. 120 - 125)

We believe this addition now adequately cautions readers about the scope of our proteomic results while also highlighting important opportunities for methodological refinement in subsequent studies. As the reviewer notes, this is an important addition that increases the transparency and rigor of this study.

We also note that we will continue to be in active consultation with proteomic experts and a key future direction for our group will be to implement multiple analysis pipelines that incorporate ECM-relevant post-translational modifications, thereby enabling a more complete representation of the aging brain matrisome. More broadly, we wish to highlight the challenges that arise when bridging scientific disciplines (i.e., cognitive aging research, systems neuroscience, glial-matrix biology, proteomics, etc). This exchange with the reviewer is a valuable example of how interdisciplinary dialogue enhances both the rigor and interpretability of complex datasets, and underscores the importance of such feedback for advancing new fields of study.